# Impact of Dataset Properties on Membership Inference Vulnerability of Deep Transfer Learning

**Marlon Tobaben**[1*]  **Hibiki Ito**[2*†]  **Joonas Jälkö**[1*]  **Yuan He**[1]  **Antti Honkela**[1]

[1]Department of Computer Science, University of Helsinki, Finland
[2]School of Informatics, Kyoto University, Japan
{marlon.tobaben,joonas.jalko,yuan.he,antti.honkela}@helsinki.fi
ito.hibiki.77n@st.kyoto-u.ac.jp

## Abstract

Membership inference attacks (MIAs) are used to test practical privacy of machine learning models. MIAs complement formal guarantees from differential privacy (DP) under a more realistic adversary model. We analyze MIA vulnerability of fine-tuned neural networks both empirically and theoretically, the latter using a simplified model of fine-tuning. We show that the vulnerability of non-DP models when measured as the attacker advantage at a fixed false positive rate reduces according to a simple power law as the number of examples per class increases. A similar power-law applies even for the most vulnerable points, but the dataset size needed for adequate protection of the most vulnerable points is very large.

## 1 Introduction

Membership inference attacks (MIAs; Shokri et al., 2017; Carlini et al., 2022) and differential privacy (DP; Dwork et al., 2006) provide complementary means of deriving lower and upper bounds for the privacy loss of a machine learning algorithm. Yet, the two operate under slightly different threat models. DP implicitly assumes a very powerful adversary with access to all training data except the target point and provides guarantees against every target point.

MIAs assume an often more realistic adversary model with access to just the data distribution and the unknown training data becoming latent variables that introduce stochasticity into the attack. However, the practical evaluation is statistical and cannot provide universal guarantees.

In this paper, we seek to explore MIA vulnerability to extrapolate this gap. Inspired by an empirical finding that average MIA vulnerability of neural network fine-tuning strongly reduces

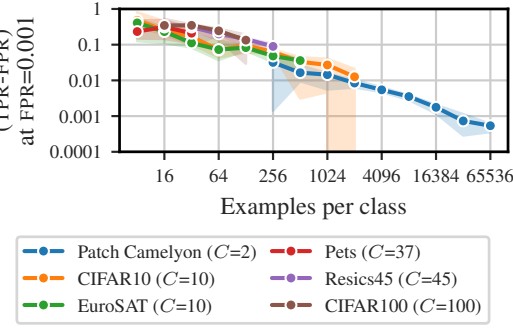

Figure 1: We observe a power-law relation between MIA vulnerability and examples per class (denoted as $S$ or shots) when attacking a fine-tuned ViT-B Head using LiRA. Each colored line denotes a different fine-tuning dataset where $C$ specifies the number of classes. The solid line is median and the error bars the min/max bounds for the Clopper-Pearson CIs over six seeds.

as the number of samples in the target class increases (see Figure 1), we develop theory of optimal MIA against a simple model of neural network fine-tuning and reproduce the decrease in vulnerability.

---

[*]These authors contributed equally.

[†]Work performed in part while at the University of Helsinki.

39th Conference on Neural Information Processing Systems (NeurIPS 2025).

Furthermore, the theoretical model predicts that the vulnerability of all individual samples should reduce as the number of samples increases, which we are able to verify empirically.

To achieve our goal, we theoretically analyze and systematically apply two state-of-the-art black-box MIAs, LiRA (Carlini et al., 2022) and RMIA (Zarifzadeh et al., 2024), to help understand practical privacy risks when fine-tuning deep-learning-based classifiers without DP protections. For the theoretical model that we analyze, LiRA is the optimal attack by the Neyman–Pearson lemma (Neyman and Pearson, 1933). Under the black-box threat model, in which the adversary does not have access to the model parameters, LiRA and RMIA have been shown to empirically outperform other attacks, especially when the number of shadow models is sufficiently large.

We focus on transfer learning using fine-tuning because this is increasingly used for all practical applications of deep learning and especially important when labeled examples are limited, which is often the case in privacy-sensitive applications. Our case study focuses on understanding and quantifying factors that influence the vulnerability of non-DP deep transfer learning models to MIA. In particular, we theoretically study the relationship between the number of examples per class, which we denote as shots ($S$), and MIA vulnerability (true positive rate TPR at fixed false positive rate FPR) for a simplified model of fine-tuning, and derive a power-law relationship (Figure 1) in the form

$$\log(\text{TPR} - \text{FPR}) = -\beta_S \log(S) - \beta_0. \tag{1}$$

We complement the theoretical analysis with extensive experiments over many datasets with varying sizes, in the transfer learning setting for image classification tasks, and observe the same power-law. Based on extrapolation from our results, the number of examples per each class that are needed for adequate protection of the most vulnerable samples appears very high.

**Related work** There has been evidence that classification models with more classes are more vulnerable to MIA (Shokri et al., 2017), models trained on fewer samples can be more vulnerable (Chen et al., 2020; Németh et al., 2025), and classes with less examples tend to be more vulnerable (Chang and Shokri, 2021; Kulynych et al., 2022; Tonni et al., 2020). A larger generalization error, which is related to dataset size, has also been shown to be sufficient for MIA success (Song and Mittal, 2021), though not necessary (Yeom et al., 2018). Similarly, minority subgroups tend to be more affected by DP (Suriyakumar et al., 2021; Bagdasaryan et al., 2019). Feldman and Zhang (2020) showed that neural networks trained from scratch are required to memorize a significant fraction of their training data to obtain high utility, while the memorization is greatly reduced for fine-tuning. Additionally, Tobaben et al. (2023) reported how the MIA vulnerability of few-shot image classification is affected by the number of shots. Yu et al. (2023) studied the relationship between the MIA vulnerability and individual privacy parameters for different classes. Recently, worst-case MIA vulnerability has gained more attention (Guépin et al., 2024; Meeus et al., 2024; Azize and Basu, 2025). Nonetheless, the prior works do not consider the rate of change in the vulnerability evaluated at a low FPR, as dataset properties change. Our work significantly expands on these works by a) explicitly identifying a quantitative relationship between dataset properties and MIA vulnerability, i.e., the power-law in Equation (1), and b) focusing on the worst-case vulnerability, both evaluated at a low FPR. This in turn allows us to extrapolate MIA vulnerability to DP guarantee.

**List of contributions** We analyze the MIA vulnerability of deep transfer learning using two state-of-the-art score-based MIAs, LiRA (Carlini et al., 2022) and RMIA (Zarifzadeh et al., 2024), which are a strong realistic threat model. We first analytically derive the power-law relationship in Equation (1) for both MIAs by introducing a simplified model of the optimal membership inference (Section 3). We support our theoretical findings by an extensive empirical study on the MIA vulnerability of deep learning models by focusing on a transfer-learning setting for image classification task, where a large pre-trained neural network is fine-tuned on a sensitive dataset.

1. *Power-law in a simplified model of the optimal MIA:* We formulate a simplified model of MIA to quantitatively relate dataset properties and MIA vulnerability. In this model LiRA is the optimal attack. For this model, we prove a power-law relationship for both average and worst-case between the LiRA as well as RMIA vulnerability and the number of examples per class (See Section 3.4).

2. *MIA experiments on the average case vulnerability:* We conduct a comprehensive study of MIA vulnerability (TPR at a fixed low FPR) in the transfer learning setting for image classification tasks with target models trained using many different datasets with varying sizes and confirm the theoretical power-law between the number of examples per class and the vulnerability to MIA (see Figure 1 and Section 4.2). We fit a regression model which follows the functional form of the

theoretically derived power-law. We show both a very good fit on the training data as well as a good prediction quality on unseen data from a different feature extractor and when fine-tuning other parameterizations (see Section 4.3).

3. *MIA experiments on the worst-case vulnerability:* We extend the experiments to worst-case individual sample vulnerabilities and observe a similar decrease in vulnerability for quantiles of vulnerable data points and a slower decrease for the maximum individual vulnerability (Section 4.4). By extrapolation we find that an adequate protection of the most vulnerable samples would require an extremely large dataset (Section 4.5).

## 2 Background

**Notation** for the properties of the training dataset $\mathcal{D}$: (i) $C$ for the number of classes (ii) $S$ for shots (examples per class) (iii) $|\mathcal{D}|$ for training dataset size ($|\mathcal{D}| = CS$). We denote the number of MIA shadow models with $M$.

**Membership inference attacks (MIAs)** aim to infer whether a particular sample was part of the training set of the targeted model (Shokri et al., 2017). Thus, they can be used to determine lower bounds on the privacy leakage of models to complement the theoretical upper bounds obtained through DP.

**Likelihood Ratio attack** (**LiRA**; Carlini et al., 2022) While many different MIAs have been proposed (Hu et al., 2022), in this work we consider the Likelihood Ratio Attack (LiRA). LiRA is a strong attack that assumes an attacker that has black-box access to the attacked model, knows the training data distribution, the training set size, the model architecture, hyperparameters and training algorithm. Based on this information, the attacker can train so-called shadow models (Shokri et al., 2017) which imitate the model under attack but for which the attacker knows the training dataset.

LiRA exploits the observation that the loss function value used to train a model is often lower for the examples that were part of the training set compared to those that were not. For a target tuple $(\boldsymbol{x}, y)$, where $y$ is a label, LiRA trains the shadow models: (i) with $(\boldsymbol{x}, y)$ as a part of the training set ($(\boldsymbol{x}, y) \in \mathcal{D}$) and (ii) without $\boldsymbol{x}$ in the training set ($(\boldsymbol{x}, y) \notin \mathcal{D}$). After training the shadow models, $(\boldsymbol{x}, y)$ is passed through the shadow models, and based on the losses (or predictions) two Gaussian distributions are formed: one for the losses of $(\boldsymbol{x}, y) \in \mathcal{D}$ shadow models, and one for the $(\boldsymbol{x}, y) \notin \mathcal{D}$. Finally, the attacker computes the loss for the point $\boldsymbol{x}$ using the model under attack and determines using a likelihood ratio test on the distributions built from the shadow models whether it is more likely that $(\boldsymbol{x}, y) \in \mathcal{D}$ or $(\boldsymbol{x}, y) \notin \mathcal{D}$. When the true distributions of the shadow models are Gaussians, LiRA is the optimal attack provided by the Neyman–Pearson lemma (Neyman and Pearson, 1933). We use an optimization by Carlini et al. (2022) for performing LiRA for multiple models and points without training a computationally infeasible number of shadow models. It relies on sampling the shadow datasets in a way that each sample is in expectation half of the time included in the training dataset of a shadow model and half of the time not. At attack time each model will be attacked once using all other models as shadow models.

**Robust Membership Inference Attack** (**RMIA**; Zarifzadeh et al., 2024) RMIA is a new MIA algorithm, which aims to improve performance when the number of shadow models is limited. We show both theoretically and empirically that the power-law also holds for RMIA.

**Measuring MIA vulnerability** Using the chosen MIA score of our attack, we can build a binary classifier to predict whether a sample belongs to the training data or not. The accuracy profile of such classifier can be used to measure the success of the MIA. More specifically, throughout the rest of the paper, we will use the true positive rate (TPR) at a specific false positive rate (FPR) as a measure for the vulnerability. Identifying even a small number of examples with high confidence is considered harmful (Carlini et al., 2022) and thus we focus on the regions of small FPR.

**Measuring the uncertainty for TPR** The TPR values from the LiRA-based classifier can be seen as maximum likelihood-estimators for the probability of producing true positives among the positive samples. Since we have a finite number of samples for our estimation, it is important to estimate the uncertainty in these estimators. Therefore, when we report the TPR values for a single repeat of the learning algorithm, we estimate the stochasticity of the TPR estimate by using Clopper-Pearson intervals (Clopper and Pearson, 1934). Given TP true positives among P positives, the $1-\alpha$ confidence

Clopper-Pearson interval for the TPR is given as

$$B(\alpha/2; \text{TP}, \text{P} - \text{TP} + 1) < \text{TPR} < B(1 - \alpha/2; \text{TP} + 1, \text{P} - \text{TP}), \tag{2}$$

where $B(q; a, b)$ is the $q$th-quantile of Beta$(a, b)$ distribution.

## 3 Theoretical analysis

In this section, we seek to theoretically understand the impact of the dataset properties on the MIA vulnerability. It is known that different data points exhibit different levels of MIA vulnerability depending on the underlying distribution (e.g. Aerni et al., 2024; Leemann et al., 2024). Therefore, we start with analysing *per-example* MIA vulnerabilities. In order to quantitatively relate dataset properties to these vulnerabilities, a simplified model is formulated. Within this model, we prove a power-law between the per-example vulnerability and the number $S$ of examples per class. Finally, the per-example power-law is analytically extended to *average-case* MIA vulnerability, for which we provide empirical evidence in Section 4. We primarily focus on the analysis of (online) LiRA, since it is the optimal attack in our simplified model. We show that similar theoretical results also hold for RMIA in Appendix B and offline LiRA Appendix A.4.

### 3.1 Preliminaries

First, let us restate the MIA score from (online) LiRA as defined by Carlini et al. (2022). Denoting the logit of a target model $\mathcal{M}$ applied on a target data point $(\boldsymbol{x}, y)$ as $\ell(\mathcal{M}(\boldsymbol{x}), y)$, LiRA computes the MIA score as the likelihood ratio

$$\text{LR}(\boldsymbol{x}) = \frac{p(\ell(\mathcal{M}(\boldsymbol{x}), y) \mid \mathbb{Q}_{\text{in}}(\boldsymbol{x}, y))}{p(\ell(\mathcal{M}(\boldsymbol{x}), y) \mid \mathbb{Q}_{\text{out}}(\boldsymbol{x}, y))}, \tag{3}$$

where the $\mathbb{Q}_{\text{in/out}}$ denote the hypotheses that $(\boldsymbol{x}, y)$ was or was not in the training set of $\mathcal{M}$. Carlini et al. (2022) approximate the IN/OUT hypotheses as normal distributions. Denoting $t_{\boldsymbol{x}} = \ell(\mathcal{M}(\boldsymbol{x}), y)$, the score becomes

$$\text{LR}(\boldsymbol{x}) = \frac{\mathcal{N}(t_{\boldsymbol{x}}; \hat{\mu}_{\text{in}}(\boldsymbol{x}), \hat{\sigma}_{\text{in}}(\boldsymbol{x})^2)}{\mathcal{N}(t_{\boldsymbol{x}}; \hat{\mu}_{\text{out}}(\boldsymbol{x}), \hat{\sigma}_{\text{out}}(\boldsymbol{x})^2)}, \tag{4}$$

where the $\hat{\mu}_{\text{in/out}}(\boldsymbol{x})$ and $\hat{\sigma}_{\text{in/out}}(\boldsymbol{x})$ are the means and standard deviations for the IN/OUT shadow model losses for $(\boldsymbol{x}, y)$. Larger values of LR$(\boldsymbol{x})$ suggest that $(\boldsymbol{x}, y)$ is more likely in the training set and vice versa. Now, to build a classifier from this score, the LiRA tests if LR$(\boldsymbol{x}) > \tau$ for some threshold $\tau$. Note that the attacker only has a finite set of shadow models to estimate the IN/OUT parameters. Therefore, the MIA scores become random variables over the true population level IN/OUT distributions.

### 3.2 Computing the TPR for LiRA

Using the LiRA formulation of Equation (4), the TPR for the target point $(\boldsymbol{x}, y)$ for LiRA is defined as

$$\text{TPR}_{\text{LiRA}}(\boldsymbol{x}) = \Pr_{\mathcal{D}_{\text{target}} \sim \mathbb{D}^{|\mathcal{D}|}, \phi^M} \left(\text{LR}(\boldsymbol{x}) \geq \tau \mid (\boldsymbol{x}, y) \in \mathcal{D}_{\text{target}}\right), \tag{5}$$

where $\tau$ is a threshold that defines a rejection region of the likelihood ratio test, and $\phi^M$ denotes the randomness in shadow set sampling and shadow model training (see Appendix A.1 for derivation).

We define the average-case TPR for LiRA by taking the expectation over the data distribution:

$$\overline{\text{TPR}}_{\text{LiRA}} = \mathbb{E}_{(\boldsymbol{x}, y) \sim \mathbb{D}}[\text{TPR}_{\text{LiRA}}(\boldsymbol{x})] \tag{6}$$

### 3.3 Per-example MIA vulnerability

Although LiRA models $t_{\boldsymbol{x}}$ by a normal distribution, we consider a more general case where the true distribution of $t_{\boldsymbol{x}}$ is of the location-scale family. That is,

$$t_{\boldsymbol{x}} = \begin{cases} \mu_{\text{in}}(\boldsymbol{x}) + \sigma_{\text{in}}(\boldsymbol{x})Z & \text{if } (\boldsymbol{x}, y) \in \mathcal{D}_{\text{target}} \\ \mu_{\text{out}}(\boldsymbol{x}) + \sigma_{\text{out}}(\boldsymbol{x})Z & \text{if } (\boldsymbol{x}, y) \notin \mathcal{D}_{\text{target}}, \end{cases} \tag{7}$$

where $Z$ has the standard location and unit scale, and $\mu_{\text{in}}(\boldsymbol{x}), \mu_{\text{out}}(\boldsymbol{x})$ and $\sigma_{\text{in}}(\boldsymbol{x}), \sigma_{\text{out}}(\boldsymbol{x})$ are the locations and scales of IN/OUT distributions of $t_{\boldsymbol{x}}$. We assume that the target and shadow datasets have a sufficient number of examples. This allows us to also assume that $\hat{\sigma}(\boldsymbol{x}) = \hat{\sigma}_{\text{in}}(\boldsymbol{x}) = \hat{\sigma}_{\text{out}}(\boldsymbol{x})$ and $\sigma(\boldsymbol{x}) = \sigma_{\text{in}}(\boldsymbol{x}) = \sigma_{\text{out}}(\boldsymbol{x})$, where $\hat{\sigma}(\boldsymbol{x})$ is the standard deviation of $t_{\boldsymbol{x}}$ estimated from shadow models and $\sigma(\boldsymbol{x})$ is the true scale parameter of $t_{\boldsymbol{x}}$. (See Appendix A.2 for the validity of these assumptions). The following result reduces the LiRA vulnerability to the location and scale parameters of $t_{\boldsymbol{x}}$.

**Lemma 1** (Per-example LiRA vulnerability). *Suppose that the true distribution of $t_{\boldsymbol{x}}$ is of location-scale family with locations $\mu_{\text{in}}(\boldsymbol{x}), \mu_{\text{out}}(\boldsymbol{x})$ and scale $\sigma(\boldsymbol{x})$, and that LiRA models $t_{\boldsymbol{x}}$ by $\mathcal{N}(\hat{\mu}_{\text{in}}(\boldsymbol{x}), \hat{\sigma}(\boldsymbol{x})^2)$ and $\mathcal{N}(\hat{\mu}_{\text{out}}(\boldsymbol{x}), \hat{\sigma}(\boldsymbol{x})^2)$. Assume that an attacker has access to the underlying distribution $\mathbb{D}$. Then for a large enough number of examples per class and infinitely many shadow models, the LiRA vulnerability of a fixed target example is*

$$
\text{TPR}_{\text{LiRA}}(\boldsymbol{x}) = \begin{cases} 1 - F_Z\left(F_Z^{-1}(1 - \text{FPR}_{\text{LiRA}}(\boldsymbol{x})) - \frac{\mu_{\text{in}}(\boldsymbol{x}) - \mu_{\text{out}}(\boldsymbol{x})}{\sigma(\boldsymbol{x})}\right) & \text{if } \hat{\mu}_{\text{in}}(\boldsymbol{x}) > \hat{\mu}_{\text{out}}(\boldsymbol{x}) \\ F_Z\left(F_Z^{-1}(\text{FPR}_{\text{LiRA}}(\boldsymbol{x})) - \frac{\mu_{\text{in}}(\boldsymbol{x}) - \mu_{\text{out}}(\boldsymbol{x})}{\sigma(\boldsymbol{x})}\right) & \text{if } \hat{\mu}_{\text{in}}(\boldsymbol{x}) < \hat{\mu}_{\text{out}}(\boldsymbol{x}), \end{cases} \tag{8}
$$

*where $F_Z$ is the cdf of $Z$ with the standard location and unit scale, assuming that the inverse of $F_Z$ exists.*

*Proof.* See Appendix A.3. $\square$

Here we assume that an attacker trains shadow models with the true underlying distribution. However, in real-world settings the precise underlying distribution may not be available for an attacker. We relax this assumption in Appendix A.5 so that the attacker only needs an approximated underlying distribution for the optimal LiRA as in Lemma 1.

### 3.4 A simplified model of the optimal membership inference

Now we construct a simplified model of membership inference that streamlines the data generation and shadow model training.

We sample vectors on a high-dimensional unit sphere and classify them based on inner product with estimated class mean. This model is easier to analyze theoretically than real-world deep learning examples. We generate the data and form the classifiers (which are our target models) as follows:

1. For each class, we first sample a true class mean $\boldsymbol{m}_c$ on a high dimensional unit sphere that is orthogonal to all other true class means ($\forall i, j \in \{1, \ldots, C\}, i \neq j : \boldsymbol{m}_i \perp \boldsymbol{m}_j$).

2. We sample $2S$ vectors $\boldsymbol{x}_c$ for each class. We assume that they are Gaussian distributed around the the true class mean $\boldsymbol{x}_c \sim \mathcal{N}(\boldsymbol{m}_c, \Sigma)$ where the $\Sigma$ is the in-class covariance.

3. For each "target model" we randomly choose a subset of size $CS$ from all generated vectors and compute per-class means $\hat{\boldsymbol{m}}_c$.

4. The computed mean is used to classify sample $\boldsymbol{x}$ by computing the inner product $\langle \boldsymbol{x}, \hat{\boldsymbol{m}}_c \rangle$ as a metric of similarity.

The attacker has to infer which vectors have been used for training the classifier. Instead of utilising the logits (like in many image classification tasks), the attacker can use the inner products of a point with the cluster means. Since the inner product score follows a normal distribution, LiRA with infinitely many shadow models is the optimal attack by the Neyman–Pearson lemma (Neyman and Pearson, 1933), which states that the likelihood ratio test is the most powerful test for a given FPR.

This simplified model resembles a linear (Head) classifier often used in transfer learning when adapting to a new dataset. We also focus on the linear (Head) classifier in our empirical evaluation in Section 4. In the linear classifier, we find a matrix $W$ and biases $b$, to optimize the cross-entropy between the labels and logits $Wv + b$, where $v$ denotes the feature space representation of the data. In the simplified model, the rows of $W$ are replaced by the cluster means and we do not include the bias term in the classification.

Now, applying Lemma 1 to the simplified model yields the following result.

**Theorem 2** (Per-example LiRA power-law). *Fix a target example $(\boldsymbol{x}, y)$. For the simplified model with arbitrary $C$ and infinitely many shadow models, the per-example LiRA vulnerability is given as*

$$\log(\text{TPR}_{\text{LiRA}}(\boldsymbol{x}) - \text{FPR}_{\text{LiRA}}(\boldsymbol{x}))$$
$$= -\frac{1}{2}\log S - \frac{1}{2}\Phi^{-1}(\text{FPR}_{\text{LiRA}}(\boldsymbol{x}))^2 + \log\frac{|\langle\boldsymbol{x}, \boldsymbol{x} - \boldsymbol{m_x}\rangle|}{\sqrt{\boldsymbol{x}^T\Sigma\boldsymbol{x}}\sqrt{2\pi}} + \log(1 + \xi(S)), \quad (9)$$

*where $\boldsymbol{m_x}$ is the true mean of class $y$ and $\xi(S) = O(1/\sqrt{S})$. For large $S$ we have*

$$\log(\text{TPR}_{\text{LiRA}}(\boldsymbol{x}) - \text{FPR}_{\text{LiRA}}(\boldsymbol{x})) \approx -\frac{1}{2}\log S - \frac{1}{2}\Phi^{-1}(\text{FPR}_{\text{LiRA}}(\boldsymbol{x}))^2 + \log\frac{|\langle\boldsymbol{x}, \boldsymbol{x} - \boldsymbol{m_x}\rangle|}{\sqrt{\boldsymbol{x}^T\Sigma\boldsymbol{x}}\sqrt{2\pi}}. \quad (10)$$

*Proof.* See Appendix A.6. □

An immediate upper bound is obtained from Theorem 2 by the Cauchy-Schwarz inequality:

$$\log(\text{TPR}_{\text{LiRA}}(\boldsymbol{x}) - \text{FPR}_{\text{LiRA}}(\boldsymbol{x})) \leq -\frac{1}{2}\log S - \frac{1}{2}\Phi^{-1}(\text{FPR}_{\text{LiRA}}(\boldsymbol{x}))^2 + \log\frac{||\boldsymbol{x} - \boldsymbol{m_x}||}{\sqrt{\boldsymbol{x}^T\Sigma\boldsymbol{x}}\sqrt{2\pi}}$$
$$+ \log(1 + \xi(S)). \quad (11)$$

This implies that if $||\boldsymbol{x} - \boldsymbol{m_x}||$ is bounded, then the worst-case vulnerability is also bounded. Hence we can significantly reduce the MIA vulnerability of all examples in this non-DP setting by simply increasing the number of examples per class.

*Remark* 3. By Lemma 1 and the proof of Theorem 2, a necessary condition for the power-law is that $(\mu_{\text{in}}(\mathbf{x}) - \mu_{\text{out}}(\mathbf{x}))/\sigma(\mathbf{x})$ converges to zero at rate $O(1/S^\alpha)$ with $\alpha > 0$. In our simplified model, this holds with $\alpha = 1/2$. However, $\mu_{\text{in}}(\mathbf{x}) - \mu_{\text{out}}(\mathbf{x}) \to \mathbf{0}$ might not always be the case for larger neural networks trained from scratch. Furthermore, even if $(\mu_{\text{in}}(\mathbf{x}) - \mu_{\text{out}}(\mathbf{x}))/\sigma(\mathbf{x})$ converges to zero, it is not clear what the convergence rate would be.

Now the following corollary extends the power-law to the average-case MIA vulnerability. We will also empirically validate this result in Section 4.

**Corollary 4** (Average-case LiRA power-law). *For the simplified model with arbitrary $C$, sufficiently large $S$ and infinitely many shadow models, we have*

$$\log(\overline{\text{TPR}}_{\text{LiRA}} - \overline{\text{FPR}}_{\text{LiRA}}) \approx -\frac{1}{2}\log S - \frac{1}{2}\Phi^{-1}(\overline{\text{FPR}}_{\text{LiRA}})^2 + \log\left(\mathbb{E}_{(\boldsymbol{x},y)\sim\mathbb{D}}\left[\frac{|\langle\boldsymbol{x}, \boldsymbol{x} - \boldsymbol{m_x}\rangle|}{\sqrt{\boldsymbol{x}^T\Sigma\boldsymbol{x}}\sqrt{2\pi}}\right]\right). \quad (12)$$

*Proof.* See Appendix A.7. □

## 4 Empirical evaluation of MIA vulnerability and dataset properties

In this section, we investigate how different properties of datasets (shots $S$ and number of classes $C$) affect the MIA vulnerability. Based on our observations, we propose a method to predict the vulnerability to MIA using these properties.

### 4.1 Experimental setup

We focus on a image classification setting where we fine-tune pre-trained models on sensitive downstream datasets and assess the MIA vulnerability using LiRA and RMIA with $M = 256$ shadow/reference models. We base our experiments on a subset of the few-shot benchmark VTAB (Zhai et al., 2019) that achieves a test classification accuracy $> 80\%$ (see Table A2).

We report results for fine-tuning a last layer classifier (Head) trained on top of a Vision Transformer ViT-Base-16 (ViT-B; Dosovitskiy et al., 2021), pre-trained on ImageNet-21k (Russakovsky et al., 2015). The results for using ResNet-50 (R-50; Kolesnikov et al., 2020) as a backbone can be found in Appendix D.1. We optimize the hyperparameters (batch size, learning rate and number of epochs) using the Optuna library (Akiba et al., 2019) with the Tree-structured Parzen Estimator (TPE; Bergstra et al., 2011) sampler with 20 iterations (more details in Appendix C.2). We provide the the code for reproducing the experiments in an open repository[3].

---

[3] https://github.com/DPBayes/impact-dataset-properties-MI-vulnerability-deep-TL

## 4.2 Experimental results

Using the setting described above, we study how the number of classes and the number of shots affect the vulnerability (TPR at FPR as described in Section 2) using LiRA. We make the following observations:

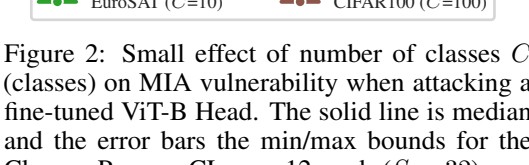

- A larger number of $S$ (**shots**) decrease the vulnerability in a power law relation as demonstrated in Figure 1. We provide tabular data and experiments using ResNet-50 in the Appendix (Figure A.1 and Tables A3 and A4).
- Contrary, a larger number of $C$ (**classes**) increases the vulnerability as demonstrated in Figure 2 with tabular data and experiments using ResNet-50 in the Appendix (Figure A.2 and Tables A5 and A6). However, the trend w.r.t. $C$ is not as clear as with $S$.

Figure 2: Small effect of number of classes $C$ (classes) on MIA vulnerability when attacking a fine-tuned ViT-B Head. The solid line is median and the error bars the min/max bounds for the Clopper-Pearson CIs over 12 seeds ($S = 32$).

**RMIA** In Figure 3 we compare the vulnerability of the models to LiRA and RMIA as a function of the number of $S$ (shots) at FPR $= 0.1$. We observe the power-law for both attacks, but the RMIA is more unstable than LiRA (especially for lower FPR). More results for RMIA are in Figures A.5 to A.7 in the Appendix.

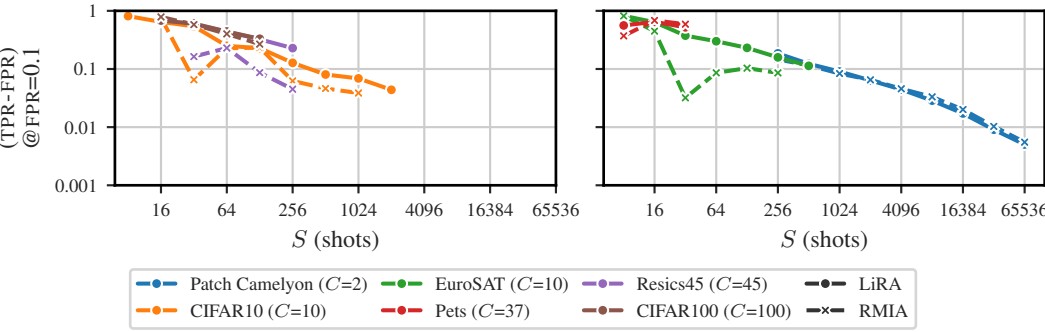

Figure 3: LiRA and RMIA vulnerability (($\text{TPR} - \text{FPR}$) at FPR $= 0.1$) as a function of shots ($S$) when attacking a ViT-B Head fine-tuned on different datasets. For better visibility, we split the datasets into two panels. We observe the power-law for both attacks, but the RMIA is more unstable than LiRA. The lines display the median over six seeds.

## 4.3 Model to predict dataset vulnerability

The trends seen in Figure 1 suggest the same power law relationship that we derived for the simplified model of membership inference in Section 3. We fit a linear regression model to predict $\log(\text{TPR} - \text{FPR})$ for each FPR $= 10^{-k}, k = 1, \ldots, 5$ separately using the $\log C$ and $\log S$ as covariates with statsmodels (Seabold and Perktold, 2010). The general form of the model can be found in Equation (13), where $\beta_S, \beta_C$ and $\beta_0$ are the learnable regression parameters.

$$\log_{10}(\text{TPR} - \text{FPR}) = \beta_S \log_{10}(S) + \beta_C \log_{10}(C) + \beta_0 \tag{13}$$

In Appendix D.2, we propose a variation of the regression model that predicts $\log_{10}(\text{TPR})$ instead of $\log_{10}(\text{TPR} - \text{FPR})$ but this alternative model performs worse on our empirical data and predicts TPR $<$ FPR in the tail when $S$ is very large.

We utilise MIA results of ViT-B (Head) (see Table A3) as the training data. Based on the $R^2$ (coefficient of determination) score ($R^2 = 0.930$ for the model trained on FPR $= 0.001$ data), our model fits the data extremely well. We provide further evidence for other FPR in Figure A.3 and

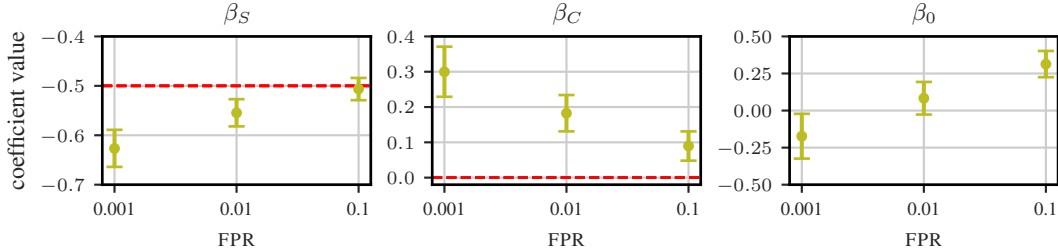

Figure 4: Coefficient values for different FPR when fitting a regression model based on Equation (13) fitted on data from ViT-B (Head) with LiRA (Table A3). The error bars display the 95% confidence intervals based on Student's t-distribution. Theoretical values in the simplified model is shown by pink dotted lines ($\beta_S = 0.5$ and $\beta_C = 0$).

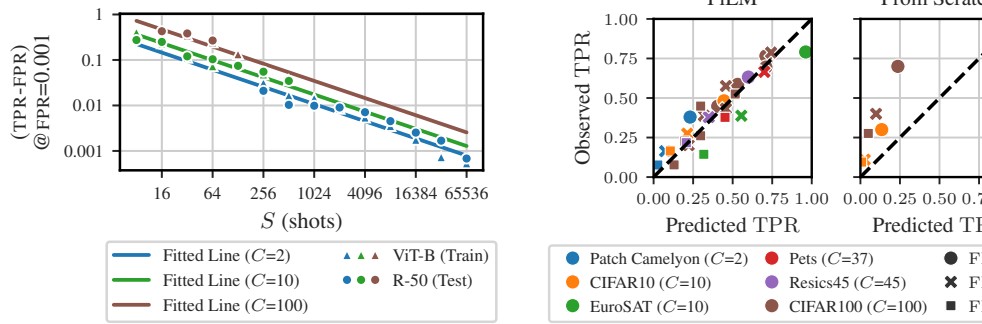

(a) The dots show the median TPR for the train set (ViT-B; Table A3) and the test set (R-50; Table A4) over six seeds (datasets: Patch Camelyon, EuroSAT and CIFAR100). The linear model is robust to changing the feature extractor from ViT-B to R-50.

(b) Regression model is robust to changing the fine-tuning method from Head to FiLM, but from scratch training seems to be more vulnerable than predicted. (i) left: fine-tuned with FiLM (see Table A7) (ii) right: trained from scratch. Data is from Carlini et al. (2022).

Figure 5: Performance of the regression model based on Equation (13) fitted on data from Table A3.

Table A8 in the Appendix. Figure 4 shows the parameters of the prediction model fitted to the training data. For larger FPR, the coefficient $\beta_S$ is around $-0.5$, as our theoretical analysis predicts.

**Prediction quality on other MIA target models** We analyze how the regression model trained on the ViT-B (Head) data generalizes to other target models. The main points are:

- *R-50 (Head):* Figure 5a shows that the regression model is robust to a change of the feature extractor, as it is able to predict the TPR for R-50 (Head) (test $R^2 = 0.790$).

- *R-50 (FiLM):* Figure 5b shows that the prediction quality is good for R-50 (FiLM) models. These models are fine-tuned with parameter-efficient FiLM (Perez et al., 2018) layers (See Appendix C.1). Tobaben et al. (2023) demonstrated that FiLM layers are a competitive alternative to training all parameters. We supplement the MIA results of Tobaben et al. (2023) with own FiLM training runs. Refer to Table A7 in the Appendix.

- *From-Scratch-Training:* Carlini et al. (2022) provide limited results on from-scratch-training. To the best of our knowledge these are the only published LiRA results on image classification models. Figure 5b displays that our prediction model underestimates the vulnerability of the from-scratch trained target models. We have identified two potential explanations for this: (i) In from-scratch-training all weights of the model need to be trained from the sensitive data and thus potentially from-scratch-training could be more vulnerable than fine-tuning. (ii) The strongest attack in Carlini et al. (2022) uses data augmentations to improve the performance. We are not using this optimization. Additionally, as noted in Remark 3, our theoretical analysis is based on a simple model and the power-law might not occur at all for larger neural networks trained from scratch.

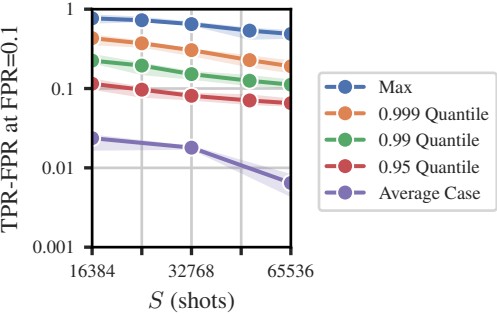

Figure 6: Individual vulnerability for ViT-B (Head) when fine-tuning on Patch Camelyon. We observe a similar power-law relationship for individuals when looking at the quantiles but the max decreases slower. The Average Case shows the average LiRA vulnerability over the individuals also illustrated in the right panel of Figure 3.

Table 1: The minimum $S$ with $C=2$ predicted by the models in Sections 4.3 and 4.4 to empirically match the DP bounds ($\delta=10^{-5}$) calculated through (Kairouz et al., 2015) in terms of TPR at FPR.

| $\epsilon$ | min $S$ in average case | | | min $S$ in worst-case |
| --- | --- | --- | --- | --- |
| | FPR=0.1 | FPR=0.01 | FPR=0.001 | FPR=0.1 |
| 0.25 | 5 400 | 69 000 | 320 000 | $5.5 \times 10^9$ |
| 0.50 | 1 100 | 16 000 | 88 000 | $2.6 \times 10^8$ |
| 0.75 | 360 | 5 900 | 38 000 | $3.5 \times 10^7$ |
| 1.00 | 160 | 2 700 | 19 000 | $7.0 \times 10^6$ |

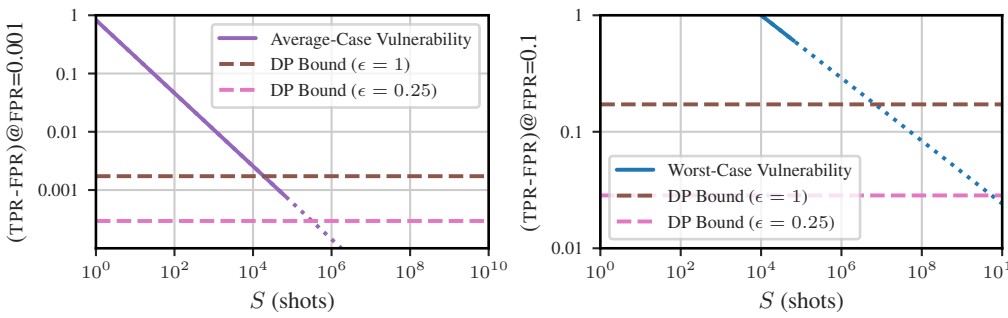

Figure 7: Illustration of the extrapolation for the average-case at FPR $= 10^{-3}$ (left) and worst-case at FPR $= 10^{-1}$ (right).

## 4.4 Individual MIA vulnerability

In order to assess the per-sample MIA vulnerability, we run the experiment with 257 models with each of them once acting as a target and as a shadow model otherwise, compute the TPR at FPR for every sample separately. In Figure 6, we display the individual vulnerability as a function of $S$ (shots) for Patch Camelyon and compare it with corresponding average case vulnerability from Figure 3. The plot shows the maximal vulnerability of all samples and different quantiles over six seeds. The solid line is the median and the errorbars display the min and max over seeds. The quantiles are more robust to extreme outliers and show decreasing trends already at much lower $S$ than the maximum vulnerability.

When fitting the model in Equation (13) we observe that the coefficients $\beta_S$ that model the relationship between vulnerability and examples per class for the quantiles are $-0.5603$, $-0.5688$ and $-0.4796$ which is close to the theoretical value of $-0.5$ derived in the theoretical analysis in Section 3.4. However, the maximum vulnerability decreases with a lower slope of $-0.2695$ which is considerably smaller. With larger $S$ the slope of the max vulnerability increases, e.g., with $S \geq 32768$ the coefficient is $-0.3478$ suggesting that higher $S$ is required for the most vulnerable points.

## 4.5 Comparison between empirical models and universal DP bounds

While the practical evaluation through MIAs is statistical and does not provide universal formal guarantees like DP, the power-law can aid understanding about how the practical vulnerability to MIA behaves in a more realistic threat model when the examples per class increase.

Using the translation between the (TPR, FPR) to $(\epsilon, \delta)$-DP guarantees proposed by Kairouz et al. (2015), we compute the minimum $S$ predicted by our empirical models such that the predicted TPR

matches the theoretical bound at target DP level. We illustrate this in Figure 7. (See Appendix E for a more detailed description.) Table 1 shows the resulting lower bounds of $S$ for various DP levels and values of FPR. While our empirical observations do not provide any formal guarantees, the comparison serves as an illustration to better understand the different requirements of the average and individual vulnerability. We can see that both for the average and the worst case, obtaining a low FPR would require a large amount of samples per class in order to match the TPR for meaningfully strong DP bounds.

## 5   Discussion

Trying to bridge empirical MIA vulnerability and formal DP guarantees is not an easy task because of different threat models and different nature of bounds (statistical vs. universal). While we were able to show both theoretically (Section 3) and empirically (Section 4) that having more examples per class provides protection against MIA in fine-tuned neural networks, the numbers required for significant protection (Section 4.5) limit the practical utility of this observation.

Using a level of formal DP bounds that provide meaningful protection as a yardstick, at least tens of thousands of examples per class are needed for every class even at FPR $= 0.001$. The $\epsilon$ values used here are formal upper bounds and must not be compared to actual DP deep learning with comparable privacy budgets, as the latter would be far less vulnerable. At this number of samples per class and a good pre-trained model, the impact of DP training is often negligible. This stresses the importance of formal guarantees like DP for privacy protection.

Our formal analysis focuses on a setting that can be linked to fine-tuning. As shown in Figure 5b, from-scratch training likely has higher vulnerability. Formally analyzing more models is an interesting area for future research.

Since our MIA evaluation assumes that only a target point is known to the adversary and the rest of the dataset is random, this stochasticity likely introduces some protection. Hence the power-law may be invalidated under stronger MIA settings where the adversary has access to other points in the private dataset (Bai et al., 2025).

Our experiments show that there is a difference between classes in terms of vulnerability and an interesting direction for future work is to understand the properties of classes that influence this vulnerability, e.g., variability within or between classes or their separability. Another direction for future work is understanding the impact of pre-training data on the vulnerability of fine-tuning data.

**Broader Impact**   Our work systematically studies factors influencing privacy risk of trained ML models. This has significant positive impact on users training ML models on personal data by allowing them to understand and limit the risks.

**Limitations**   We mostly consider LiRA in our paper which is optimal for our simplified model of membership inference (Section 3.4), but for the transfer-learning experiments (Section 4) there might be stronger attacks in the future. Furthermore, our simplified model assumes well-behaved underlying distributions, meaning that the data is normally distributed around the class centers. We leave the analysis of other data distributions (e.g., heavy-tailed distributions) to future work. Formal bounds on MIA vulnerability would require something like DP. In addition, both our theoretical and empirical analysis focus on deep transfer learning using fine-tuning. Models trained from scratch are likely to be more vulnerable. Concurrently, a related paper (Hayes et al., 2025) has studied the relation between LiRA vulnerability and dataset size when attacking LLMs during the pre-training phase. Extending this analysis is promising but computationally very expensive.

We provide the code in an open repository[4]. All used pre-trained models and datasets are publicly available.

## Acknowledgments

This work was supported by the Research Council of Finland (Flagship programme: Finnish Center for Artificial Intelligence, FCAI, Grant 356499 and Grant 359111), the Strategic Research Council

---

[4] https://github.com/DPBayes/impact-dataset-properties-MI-vulnerability-deep-TL

at the Research Council of Finland (Grant 358247), the European Union (Project 101070617) as well as JSPS KAKENHI Grant Number 25KJ1515. Views and opinions expressed are however those of the author(s) only and do not necessarily reflect those of the European Union or the European Commission. Neither the European Union nor the granting authority can be held responsible for them. he authors wish to acknowledge CSC – IT Center for Science, Finland, for computational resources. We thank Mikko A. Heikkilä and Ossi Räisä for helpful comments and suggestions and John F. Bronskill for helpful discussions regarding few-shot learning.

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

# Supplementary Material

## Contents

# A  Details of Section 3

## A.1  Formulating LiRA

Let $\mathcal{M}$ be our target model and $\ell(\mathcal{M}(\boldsymbol{x}), y)$ be the loss of the model on a target example $(\boldsymbol{x}, y)$. The goal of MIA is to determine whether $(\boldsymbol{x}, y) \in \mathcal{D}_{\text{target}}$. This can be formulated as a hypothesis test:

$$H_0 : (\boldsymbol{x}, y) \notin \mathcal{D}_{\text{target}} \tag{A1}$$
$$H_1 : (\boldsymbol{x}, y) \in \mathcal{D}_{\text{target}}. \tag{A2}$$

Following (Carlini et al., 2022), we formulate the Likelihood Ratio Attack (LiRA). LiRA exploits the difference of losses on the target model under $H_0$ and $H_1$. To model the IN/OUT loss distributions with few shadow models, LiRA employs a parametric modelling. Particularly, LiRA models $t_{\boldsymbol{x}}$ by a normal distribution. That is, the hypothesis test formulated above can be rewritten as

$$H_0' : t_{\boldsymbol{x}} \sim \mathcal{N}(\hat{\mu}_{\text{out}}, \hat{\sigma}_{\text{out}}^2) \tag{A3}$$
$$H_1' : t_{\boldsymbol{x}} \sim \mathcal{N}(\hat{\mu}_{\text{in}}, \hat{\sigma}_{\text{in}}^2). \tag{A4}$$

The likelihood ratio is now

$$\text{LR}(\boldsymbol{x}) = \frac{\mathcal{N}(t_{\boldsymbol{x}}; \hat{\mu}_{\text{in}}, \hat{\sigma}_{\text{in}}^2)}{\mathcal{N}(t_{\boldsymbol{x}}; \hat{\mu}_{\text{out}}, \hat{\sigma}_{\text{out}}^2)}. \tag{A5}$$

LiRA rejects $H_0'$ if and only if

$$\text{LR}(\boldsymbol{x}) \geq \tau, \tag{A6}$$

concluding that $H_1'$ is true, i.e., identifying the membership of $(\boldsymbol{x}, y)$. Thus, the true positive rate of this hypothesis test given as

$$\text{TPR}_{\text{LiRA}}(\boldsymbol{x}) = \Pr_{\mathcal{D}_{\text{target}} \sim \mathbb{D}^{|\mathcal{D}|}, \phi^M} \left( \text{LR}(\boldsymbol{x}) \geq \tau \mid (\boldsymbol{x}, y) \in \mathcal{D}_{\text{target}} \right), \tag{A7}$$

where $\phi^M$ denotes the randomness in the shadow set sampling and shadow model training.

## A.2  On the assumption of shared scale

In Section 3 we assumed that for LiRA $\sigma_{\text{in}} = \sigma_{\text{out}}$ and $\hat{\sigma}_{\text{in}} = \hat{\sigma}_{\text{out}}$. Using the simplified model formulated in Section 3.4, we show that for large enough number $S$ of examples per class these assumptions are reasonable.

Let $\mathcal{D}_{\text{target}} = \{(\boldsymbol{x}_{j,1}, j), ..., (\boldsymbol{x}_{j,S}, j)\}_{j=1}^{C}$. Then the IN/OUT LiRA scores are given as

$$s_y^{(\text{in})} = \langle \boldsymbol{x}, \frac{1}{S} \left( \sum_{i=1}^{S-1} \boldsymbol{x}_{y,i} + \boldsymbol{x} \right) \rangle = \langle \boldsymbol{x}, \frac{1}{S} \sum_{i=1}^{S} \boldsymbol{x}_{y,i} \rangle + \langle \boldsymbol{x}, \frac{1}{S}(\boldsymbol{x} - \boldsymbol{x}_{y,S}) \rangle \tag{A8}$$

$$s_y^{(\text{out})} = \langle \boldsymbol{x}, \frac{1}{S} \sum_{i=1}^{S} \boldsymbol{x}_{y,i} \rangle. \tag{A9}$$

Since for the simplified model scores follow Gaussian distributions, $\sigma_{\text{in}} = \hat{\sigma}_{\text{in}}$ and $\sigma_{\text{out}} = \hat{\sigma}_{\text{out}}$. It follows that

$$\sigma_{\text{in}}^2 = \hat{\sigma}_{\text{in}}^2 = \text{Var}(s_y^{(\text{in})}) = \frac{1}{S} \text{Var}(\langle \boldsymbol{x}, \boldsymbol{x}_{y,i} \rangle) - \frac{1}{S^2} \text{Var}(\langle \boldsymbol{x}, \boldsymbol{x}_{y,i} \rangle) = \frac{1}{S}\left(1 - \frac{1}{S}\right) \boldsymbol{x}^T \Sigma \boldsymbol{x} \tag{A10}$$

$$\sigma_{\text{out}}^2 = \hat{\sigma}_{\text{out}}^2 = \text{Var}(s_y^{(\text{out})}) = \frac{1}{S} \text{Var}(\langle \boldsymbol{x}, \boldsymbol{x}_{y,i} \rangle) = \frac{1}{S} \boldsymbol{x}^T \Sigma \boldsymbol{x}. \tag{A11}$$

Thus, the differences $\sigma_{\text{in}} - \sigma_{\text{out}}$ and $\hat{\sigma}_{\text{in}} - \hat{\sigma}_{\text{out}}$ are negligible for large $S$.

## A.3  Proof of Lemma 1

**Lemma 1** (Per-example LiRA vulnerability). *Suppose that the true distribution of $t_{\boldsymbol{x}}$ is of location-scale family with locations $\mu_{\text{in}}(\boldsymbol{x}), \mu_{\text{out}}(\boldsymbol{x})$ and scale $\sigma(\boldsymbol{x})$, and that LiRA models $t_{\boldsymbol{x}}$ by $\mathcal{N}(\hat{\mu}_{\text{in}}(\boldsymbol{x}), \hat{\sigma}(\boldsymbol{x})^2)$ and $\mathcal{N}(\hat{\mu}_{\text{out}}(\boldsymbol{x}), \hat{\sigma}(\boldsymbol{x})^2)$. Assume that an attacker has access to the underlying*

distribution $\mathbb{D}$. Then for a large enough number of examples per class and infinitely many shadow models, the LiRA vulnerability of a fixed target example is

$$
\text{TPR}_{\text{LiRA}}(\boldsymbol{x}) = \begin{cases} 1 - F_Z\left(F_Z^{-1}(1 - \text{FPR}_{\text{LiRA}}(\boldsymbol{x})) - \frac{\mu_{\text{in}}(\boldsymbol{x}) - \mu_{\text{out}}(\boldsymbol{x})}{\sigma(\boldsymbol{x})}\right) & \text{if } \hat{\mu}_{\text{in}}(\boldsymbol{x}) > \hat{\mu}_{\text{out}}(\boldsymbol{x}) \\ F_Z\left(F_Z^{-1}(\text{FPR}_{\text{LiRA}}(\boldsymbol{x})) - \frac{\mu_{\text{in}}(\boldsymbol{x}) - \mu_{\text{out}}(\boldsymbol{x})}{\sigma(\boldsymbol{x})}\right) & \text{if } \hat{\mu}_{\text{in}}(\boldsymbol{x}) < \hat{\mu}_{\text{out}}(\boldsymbol{x}), \end{cases}
\tag{8}
$$

where $F_Z$ is the cdf of $Z$ with the standard location and unit scale, assuming that the inverse of $F_Z$ exists.

*Proof.* We abuse notations by denoting $\mu_{\text{in}}$ to refer to $\mu_{\text{in}}(\boldsymbol{x})$ and similarly for other statistics. We have

$$
\log \frac{\mathcal{N}(t_{\boldsymbol{x}}; \hat{\mu}_{\text{in}}, \hat{\sigma}^2)}{\mathcal{N}(t_{\boldsymbol{x}}; \hat{\mu}_{\text{out}}, \hat{\sigma}^2)} \geq \log \tau
\tag{A12}
$$

$$
-\frac{1}{2}\left(\frac{t_{\boldsymbol{x}} - \hat{\mu}_{\text{in}}}{\hat{\sigma}}\right)^2 + \frac{1}{2}\left(\frac{t_{\boldsymbol{x}} - \hat{\mu}_{\text{out}}}{\hat{\sigma}}\right)^2 \geq \log \tau
\tag{A13}
$$

$$
\frac{1}{2\hat{\sigma}^2}(2t_{\boldsymbol{x}}\hat{\mu}_{\text{in}} - \hat{\mu}_{\text{in}}^2 - 2t_{\boldsymbol{x}}\hat{\mu}_{\text{out}} + \hat{\mu}_{\text{out}}^2) \geq \log \tau
\tag{A14}
$$

$$
\frac{1}{2\hat{\sigma}^2}(\hat{\mu}_{\text{in}} - \hat{\mu}_{\text{out}})(2t_{\boldsymbol{x}} - \hat{\mu}_{\text{in}} - \hat{\mu}_{\text{out}}) \geq \log \tau
\tag{A15}
$$

$$
\begin{cases} t_{\boldsymbol{x}} \geq \frac{\hat{\sigma}^2 \log \tau}{\hat{\mu}_{\text{in}} - \hat{\mu}_{\text{out}}} + \frac{\hat{\mu}_{\text{in}} + \hat{\mu}_{\text{out}}}{2} & \text{if } \hat{\mu}_{\text{in}} > \hat{\mu}_{\text{out}} \\ t_{\boldsymbol{x}} \leq \frac{\hat{\sigma}^2 \log \tau}{\hat{\mu}_{\text{in}} - \hat{\mu}_{\text{out}}} + \frac{\hat{\mu}_{\text{in}} + \hat{\mu}_{\text{out}}}{2} & \text{if } \hat{\mu}_{\text{in}} < \hat{\mu}_{\text{out}}. \end{cases}
\tag{A16}
$$

Then if $\hat{\mu}_{\text{in}} > \hat{\mu}_{\text{out}}$, in the limit of infinitely many shadow models

$$
\text{FPR}_{\text{LiRA}}(\boldsymbol{x}) = \Pr_Z\left(\mu_{\text{out}} + \sigma Z \geq \frac{\hat{\sigma}^2 \log \tau}{\hat{\mu}_{\text{in}} - \hat{\mu}_{\text{out}}} + \frac{\hat{\mu}_{\text{in}} + \hat{\mu}_{\text{out}}}{2}\right)
\tag{A17}
$$

$$
= \Pr_Z\left(Z \geq \frac{\hat{\sigma}^2 \log \tau}{\sigma(\hat{\mu}_{\text{in}} - \hat{\mu}_{\text{out}})} + \frac{\hat{\mu}_{\text{in}} + \hat{\mu}_{\text{out}}}{2\sigma} - \frac{\mu_{\text{out}}}{\sigma}\right)
\tag{A18}
$$

$$
= 1 - F_Z\left(\frac{\hat{\sigma}^2 \log \tau}{\sigma(\hat{\mu}_{\text{in}} - \hat{\mu}_{\text{out}})} + \frac{\hat{\mu}_{\text{in}} + \hat{\mu}_{\text{out}}}{2\sigma} - \frac{\mu_{\text{out}}}{\sigma}\right),
\tag{A19}
$$

and if $\hat{\mu}_{\text{in}} < \hat{\mu}_{\text{out}}$, similarly,

$$
\text{FPR}_{\text{LiRA}}(\boldsymbol{x}) = \Pr_Z\left(\mu_{\text{out}} + \sigma Z \leq \frac{\hat{\sigma}^2 \log \tau}{\hat{\mu}_{\text{in}} - \hat{\mu}_{\text{out}}} + \frac{\hat{\mu}_{\text{in}} + \hat{\mu}_{\text{out}}}{2}\right)
\tag{A20}
$$

$$
= F_Z\left(\frac{\hat{\sigma}^2 \log \tau}{\sigma(\hat{\mu}_{\text{in}} - \hat{\mu}_{\text{out}})} + \frac{\hat{\mu}_{\text{in}} + \hat{\mu}_{\text{out}}}{2\sigma} - \frac{\mu_{\text{out}}}{\sigma}\right).
\tag{A21}
$$

Thus

$$
\frac{\hat{\sigma}^2 \log \tau}{\sigma(\hat{\mu}_{\text{in}} - \hat{\mu}_{\text{out}})} + \frac{\hat{\mu}_{\text{in}} + \hat{\mu}_{\text{out}}}{2\sigma} - \frac{\mu_{\text{out}}}{\sigma} = \begin{cases} F_Z^{-1}(1 - \text{FPR}_{\text{LiRA}}(\boldsymbol{x})) & \text{if } \hat{\mu}_{\text{in}} > \hat{\mu}_{\text{out}} \\ F_Z^{-1}(\text{FPR}_{\text{LiRA}}(\boldsymbol{x})) & \text{if } \hat{\mu}_{\text{in}} < \hat{\mu}_{\text{out}}. \end{cases}
\tag{A22}
$$

It follows that if $\hat{\mu}_{\text{in}} > \hat{\mu}_{\text{out}}$,

$$
\text{TPR}_{\text{LiRA}}(\boldsymbol{x}) = \Pr_Z\left(\mu_{\text{in}} + \sigma Z \geq \frac{\hat{\sigma}^2 \log \tau}{\hat{\mu}_{\text{in}} - \hat{\mu}_{\text{out}}} + \frac{\hat{\mu}_{\text{in}} + \hat{\mu}_{\text{out}}}{2}\right)
\tag{A23}
$$

$$
= \Pr_Z\left(Z \geq \frac{\hat{\sigma}^2 \log \tau}{\sigma(\hat{\mu}_{\text{in}} - \hat{\mu}_{\text{out}})} + \frac{\hat{\mu}_{\text{in}} + \hat{\mu}_{\text{out}}}{2\sigma} - \frac{\mu_{\text{in}}}{\sigma}\right)
\tag{A24}
$$

$$
= 1 - F_Z\left(F_Z^{-1}(1 - \text{FPR}_{\text{LiRA}}(\boldsymbol{x})) - \frac{\mu_{\text{in}} - \mu_{\text{out}}}{\sigma}\right).
\tag{A25}
$$

If $\hat{\mu}_{\text{in}} < \hat{\mu}_{\text{out}}$, then

$$\text{TPR}_{\text{LiRA}}(\boldsymbol{x}) = \Pr_{Z}\left(\mu_{\text{in}} + \sigma Z \leq \frac{\hat{\sigma}^2 \log \tau}{\hat{\mu}_{\text{in}} - \hat{\mu}_{\text{out}}} + \frac{\hat{\mu}_{\text{in}} + \hat{\mu}_{\text{out}}}{2}\right) \tag{A26}$$

$$= \Pr_{Z}\left(Z \leq \frac{\hat{\sigma}^2 \log \tau}{\sigma(\hat{\mu}_{\text{in}} - \hat{\mu}_{\text{out}})} + \frac{\hat{\mu}_{\text{in}} + \hat{\mu}_{\text{out}}}{2\sigma} - \frac{\mu_{\text{in}}}{\sigma}\right) \tag{A27}$$

$$= F_Z\left(F_Z^{-1}(\text{FPR}_{\text{LiRA}}(\boldsymbol{x})) - \frac{\mu_{\text{in}} - \mu_{\text{out}}}{\sigma}\right). \tag{A28}$$

$\square$

## A.4 Offline LiRA

Carlini et al. (2022) proposes offline LiRA that only trains OUT shadow models for computational efficiency. Instead of likelihood ratio, the score for offline LiRA is given as

$$\Lambda(\boldsymbol{x}) = \Pr_{Z}(\hat{\mu}_{\text{out}}(\boldsymbol{x}) + \hat{\sigma}_{\text{out}}(\boldsymbol{x})Z \leq t_{\boldsymbol{x}}), \tag{A29}$$

where $Z$ is a location-scale distribution with standard location and unit scale. Then TPR of offline LiRA becomes

$$\text{TPR}_{\text{offLiRA}}(\boldsymbol{x}) = \Pr_{\mathcal{D}_{\text{target}} \sim \mathbb{D}^{|\mathcal{D}|}, \phi^M} \left(\Lambda(\boldsymbol{x}) \geq \tau \mid (\boldsymbol{x}, y) \in \mathcal{D}_{\text{target}}\right), \tag{A30}$$

where $\gamma$ is a tunable parameter. FPR is also given similarly. Below we show a result for offline LiRA similar to Lemma 1, but under a slightly more strict assumption that an attacker accurately estimates the IN/OUT score distributions. In the following, as for online LiRA, we assume $\sigma_{\text{in}} = \sigma_{\text{out}}$.

**Lemma A1** (Per-example offline LiRA vulnerability). *Suppose that $t_{\boldsymbol{x}}$ follows the normal distribution with means $\mu_{\text{in}}(\boldsymbol{x}), \mu_{\text{out}}(\boldsymbol{x})$ and standard deviation $\sigma(\boldsymbol{x})$. Assume that an attacker has access to the underlying distribution $\mathbb{D}$. Then for a large enough number of examples per class and infinitely many shadow models, the offline LiRA vulnerability of a fixed target example is*

$$\text{TPR}_{\text{offLiRA}}(\boldsymbol{x}) = \Phi\left(\Phi^{-1}\left(\text{FPR}_{\text{offLiRA}}\right) + \frac{\mu_{\text{in}}(\boldsymbol{x}) - \mu_{\text{out}}(\boldsymbol{x})}{\sigma(\boldsymbol{x})}\right) \tag{A31}$$

*where $\Phi$ is the standard normal cdf.*

*Proof.* For infinitely many shadow models, we have

$$\text{FPR}_{\text{offLiRA}}(\boldsymbol{x}) = \Pr_{\mathcal{D}_{\text{target}} \sim \mathbb{D}^{|\mathcal{D}|}} \left(\Lambda(\boldsymbol{x}) \geq \tau \mid (\boldsymbol{x}, y) \notin \mathcal{D}_{\text{target}}\right), \tag{A32}$$

and the score is now

$$\Lambda(\boldsymbol{x}) = \Pr_{\eta}(\mu_{\text{out}}(\boldsymbol{x}) + \sigma(\boldsymbol{x})\eta \leq t_{\boldsymbol{x}}), \tag{A33}$$

where $\eta \sim \mathcal{N}(0, 1)$. When $(\boldsymbol{x}, y) \notin \mathcal{D}_{\text{target}}$, $t_{\boldsymbol{x}} = \mu_{\text{out}}(\boldsymbol{x}) + \sigma(\boldsymbol{x})Z$. Thus we have

$$\text{FPR}_{\text{offLiRA}} = \Pr_{Z}\left(\Lambda(\boldsymbol{x}) \geq \gamma\right) \tag{A34}$$

$$= \Pr_{Z}\left(\Pr_{\eta}(\mu_{\text{out}}(\boldsymbol{x}) + \sigma(\boldsymbol{x})\eta \leq \mu_{\text{out}}(\boldsymbol{x}) + \sigma(\boldsymbol{x})Z) \geq \gamma\right) \tag{A35}$$

$$= \Pr_{Z}\left(\Pr_{\eta}(\eta \leq Z) \geq \gamma\right) \tag{A36}$$

$$= \Pr_{Z}(\Phi(Z) \geq \gamma) \tag{A37}$$

$$= 1 - \gamma. \tag{A38}$$

On the other hand, when $(\boldsymbol{x}, y) \in \mathcal{D}_{\text{target}}$, $t_{\boldsymbol{x}} = \mu_{\text{in}}(\boldsymbol{x}) + \sigma(\boldsymbol{x})Z$. Thus we obtain

$$\text{TPR}_{\text{offLiRA}} = \Pr_Z \left( \Pr_\eta (\mu_{\text{out}}(\boldsymbol{x}) + \sigma(\boldsymbol{x})\eta \le \mu_{\text{in}}(\boldsymbol{x}) + \sigma(\boldsymbol{x})Z) \ge \gamma \right) \tag{A39}$$

$$= \Pr_Z \left( \Pr_\eta \left( \eta \le \frac{\mu_{\text{in}}(\boldsymbol{x}) - \mu_{\text{out}}(\boldsymbol{x})}{\sigma(\boldsymbol{x})} + Z \right) \ge 1 - \text{FPR}_{\text{offLiRA}} \right) \tag{A40}$$

$$= \Pr_Z \left( \Pr_\eta \left( \eta \le -\frac{\mu_{\text{in}}(\boldsymbol{x}) - \mu_{\text{out}}(\boldsymbol{x})}{\sigma(\boldsymbol{x})} - Z \right) \le \text{FPR}_{\text{offLiRA}} \right) \tag{A41}$$

$$= \Pr_Z \left( -\frac{\mu_{\text{in}}(\boldsymbol{x}) - \mu_{\text{out}}(\boldsymbol{x})}{\sigma(\boldsymbol{x})} - Z \le \Phi^{-1}(\text{FPR}_{\text{offLiRA}}) \right) \tag{A42}$$

$$= \Phi \left( \Phi^{-1}(\text{FPR}_{\text{offLiRA}}) + \frac{\mu_{\text{in}}(\boldsymbol{x}) - \mu_{\text{out}}(\boldsymbol{x})}{\sigma(\boldsymbol{x})} \right). \tag{A43}$$

$\square$

Consequently, the power-law also holds for offline LiRA in the simplified model:

**Corollary A2** (Per-example offline LiRA power-law). *Fix a target example $(\boldsymbol{x}, y)$. For the simplified model with arbitrary $C$ and infinitely many shadow models, the per-example offline LiRA vulnerability is given as*

$$\log(\text{TPR}_{\text{offLiRA}}(\boldsymbol{x}) - \text{FPR}_{\text{offLiRA}}(\boldsymbol{x}))$$
$$= -\frac{1}{2}\log S - \frac{1}{2}\Phi^{-1}(\text{FPR}_{\text{offLiRA}}(\boldsymbol{x}))^2 + \log \frac{|\langle \boldsymbol{x}, \boldsymbol{x} - \boldsymbol{m}_{\boldsymbol{x}} \rangle|}{\sqrt{\boldsymbol{x}^T \Sigma \boldsymbol{x}}\sqrt{2\pi}} + \log(1 + \xi(S)) \tag{A44}$$

*where $\boldsymbol{m}_{\boldsymbol{x}}$ is the true mean of class $y$ and $\xi(S) = O(1/\sqrt{S})$. For large $S$ we have*

$$\log(\text{TPR}_{\text{offLiRA}}(\boldsymbol{x}) - \text{FPR}_{\text{offLiRA}}(\boldsymbol{x})) \approx -\frac{1}{2}\log S - \frac{1}{2}\Phi^{-1}(\text{FPR}_{\text{LiRA}}(\boldsymbol{x}))^2 + \log \frac{|\langle \boldsymbol{x}, \boldsymbol{x} - \boldsymbol{m}_{\boldsymbol{x}} \rangle|}{\sqrt{\boldsymbol{x}^T \Sigma \boldsymbol{x}}\sqrt{2\pi}}. \tag{A45}$$

### A.5  Relaxing the assumption of Lemma 1

In Lemma 1 we assume that an attacker has access to the true underlying distribution. However, in real-world settings the precise underlying distribution may not be available for an attacker. In the following, noting that the Equation (8) mainly relies on the true location parameters $\mu_{\text{in}}(\boldsymbol{x}), \mu_{\text{out}}(\boldsymbol{x})$ and scale parameter $\sigma(\boldsymbol{x})$, we relax this assumption of distribution availability so that the attacker only needs an approximated underlying distribution for the optimal LiRA that achieves the performance in Lemma 1.

First, notice that if we completely drop this assumption so that an attacker trains shadow models with an arbitrary underlying distribution, then we may not be able to choose a desired $\text{FPR}_{\text{LiRA}}(\boldsymbol{x})$. From Equation (A22) we have

$$\frac{\hat{\sigma}^2 \log \tau}{\sigma(\hat{\mu}_{\text{in}} - \hat{\mu}_{\text{out}})} + \frac{\hat{\mu}_{\text{in}} + \hat{\mu}_{\text{out}}}{2\sigma} - \frac{\mu_{\text{out}}}{\sigma} = \begin{cases} F_Z^{-1}(1 - \text{FPR}_{\text{LiRA}}(\boldsymbol{x})) & \text{if } \hat{\mu}_{\text{in}} > \hat{\mu}_{\text{out}} \\ F_Z^{-1}(\text{FPR}_{\text{LiRA}}(\boldsymbol{x})) & \text{if } \hat{\mu}_{\text{in}} < \hat{\mu}_{\text{out}} \end{cases} \tag{A46}$$

$$\log \tau = \begin{cases} \frac{\hat{\mu}_{\text{in}} - \hat{\mu}_{\text{out}}}{\hat{\sigma}^2} \left( \sigma F_Z^{-1}(1 - \text{FPR}_{\text{LiRA}}(\boldsymbol{x})) - \frac{\hat{\mu}_{\text{in}} + \hat{\mu}_{\text{out}}}{2} + \mu_{\text{out}} \right) & \text{if } \hat{\mu}_{\text{in}} > \hat{\mu}_{\text{out}} \\ \frac{\hat{\mu}_{\text{in}} - \hat{\mu}_{\text{out}}}{\hat{\sigma}^2} \left( \sigma F_Z^{-1}(\text{FPR}_{\text{LiRA}}(\boldsymbol{x})) - \frac{\hat{\mu}_{\text{in}} + \hat{\mu}_{\text{out}}}{2} + \mu_{\text{out}} \right) & \text{if } \hat{\mu}_{\text{in}} < \hat{\mu}_{\text{out}}. \end{cases} \tag{A47}$$

Since it does not make sense to choose a rejection region of the likelihood ratio test such that $\tau < 1$, we assume $\tau \ge 1$. Then we have

$$\begin{cases} \frac{\hat{\mu}_{\text{in}} - \hat{\mu}_{\text{out}}}{\hat{\sigma}^2} \left( \sigma F_Z^{-1}(1 - \text{FPR}_{\text{LiRA}}(\boldsymbol{x})) - \frac{\hat{\mu}_{\text{in}} + \hat{\mu}_{\text{out}}}{2} + \mu_{\text{out}} \right) \ge 0 & \text{if } \hat{\mu}_{\text{in}} > \hat{\mu}_{\text{out}} \\ \frac{\hat{\mu}_{\text{in}} - \hat{\mu}_{\text{out}}}{\hat{\sigma}^2} \left( \sigma F_Z^{-1}(\text{FPR}_{\text{LiRA}}(\boldsymbol{x})) - \frac{\hat{\mu}_{\text{in}} + \hat{\mu}_{\text{out}}}{2} + \mu_{\text{out}} \right) \ge 0 & \text{if } \hat{\mu}_{\text{in}} < \hat{\mu}_{\text{out}}. \end{cases} \tag{A48}$$

Therefore, a sufficient condition about attacker's knowledge on the underlying distribution for Lemma 1 to hold is

$$\begin{cases} \sigma F_Z^{-1}(1 - \text{FPR}_{\text{LiRA}}(\boldsymbol{x})) - \frac{\hat{\mu}_{\text{in}} + \hat{\mu}_{\text{out}}}{2} + \mu_{\text{out}} \ge 0 & \text{if } \hat{\mu}_{\text{in}} > \hat{\mu}_{\text{out}} \\ \sigma F_Z^{-1}(\text{FPR}_{\text{LiRA}}(\boldsymbol{x})) - \frac{\hat{\mu}_{\text{in}} + \hat{\mu}_{\text{out}}}{2} + \mu_{\text{out}} \le 0 & \text{if } \hat{\mu}_{\text{in}} < \hat{\mu}_{\text{out}}. \end{cases} \tag{A49}$$

Rearranging the terms yields

$$\begin{cases} \frac{\hat{\mu}_{\text{in}} + \hat{\mu}_{\text{out}}}{2} - \mu_{\text{out}} \leq \sigma F_Z^{-1}(1 - \text{FPR}_{\text{LiRA}}(\boldsymbol{x})) & \text{if } \hat{\mu}_{\text{in}} > \hat{\mu}_{\text{out}} \\ \frac{\hat{\mu}_{\text{in}} + \hat{\mu}_{\text{out}}}{2} - \mu_{\text{out}} \geq \sigma F_Z^{-1}(\text{FPR}_{\text{LiRA}}(\boldsymbol{x})) & \text{if } \hat{\mu}_{\text{in}} < \hat{\mu}_{\text{out}}. \end{cases} \tag{A50}$$

For example, if the estimated mean $\hat{\mu}_{\text{out}} (< \hat{\mu}_{\text{in}})$ is too large compared to the true parameter $\mu_{\text{out}}$, the left hand side for the case $\hat{\mu}_{\text{in}} > \hat{\mu}_{\text{out}}$ becomes very large, thereby forcing us to choose sufficiently small $\text{FPR}_{\text{LiRA}}(\boldsymbol{x})$. Similarly, if $\hat{\mu}_{\text{out}}(> \hat{\mu}_{\text{in}})$ is much smaller than $\mu_{\text{out}}$, then the range of possible values of $\text{FPR}_{\text{LiRA}}(\boldsymbol{x})$ will be limited. We summarise this discussion in the following:

**Lemma A3** (Lemma 1 with relaxed assumptions)**.** *Suppose that the true IN/OUT distributions of $t_{\boldsymbol{x}}$ are of a location-scale family with locations $\mu_{\text{in}}(\boldsymbol{x}), \mu_{\text{out}}(\boldsymbol{x})$ and a shared scale $\sigma(\boldsymbol{x})$ such that the distributions have finite first and second moments. Assume that LiRA models $t_{\boldsymbol{x}}$ by $\mathcal{N}(\hat{\mu}_{\text{in}}(\boldsymbol{x}), \hat{\sigma}(\boldsymbol{x})^2)$ and $\mathcal{N}(\hat{\mu}_{\text{out}}(\boldsymbol{x}), \hat{\sigma}(\boldsymbol{x})^2)$, and that in the limit of infinitely many shadow models, estimated parameters $\hat{\mu}_{\text{in}}(\boldsymbol{x}), \hat{\mu}_{\text{out}}(\boldsymbol{x})$ and $\hat{\sigma}(\boldsymbol{x})$ satisfy the following:*

$$\begin{cases} \frac{\hat{\mu}_{\text{in}} + \hat{\mu}_{\text{out}}}{2} - \mu_{\text{out}} \leq \sigma F_Z^{-1}(1 - \text{FPR}_{\text{LiRA}}(\boldsymbol{x})) & \text{if } \hat{\mu}_{\text{in}} > \hat{\mu}_{\text{out}} \\ \frac{\hat{\mu}_{\text{in}} + \hat{\mu}_{\text{out}}}{2} - \mu_{\text{out}} \geq \sigma F_Z^{-1}(\text{FPR}_{\text{LiRA}}(\boldsymbol{x})) & \text{if } \hat{\mu}_{\text{in}} < \hat{\mu}_{\text{out}}. \end{cases} \tag{A51}$$

*Then in the limit of infinitely many shadow models, the LiRA vulnerability of a fixed target example $(\boldsymbol{x}, y)$ is*

$$\text{TPR}_{\text{LiRA}}(\boldsymbol{x}) = \begin{cases} 1 - F_Z\left(F_Z^{-1}(1 - \text{FPR}_{\text{LiRA}}(\boldsymbol{x})) - \frac{\mu_{\text{in}}(\boldsymbol{x}) - \mu_{\text{out}}(\boldsymbol{x})}{\sigma(\boldsymbol{x})}\right) & \text{if } \hat{\mu}_{\text{in}}(\boldsymbol{x}) > \hat{\mu}_{\text{out}}(\boldsymbol{x}) \\ F_Z\left(F_Z^{-1}(\text{FPR}_{\text{LiRA}}(\boldsymbol{x})) - \frac{\mu_{\text{in}}(\boldsymbol{x}) - \mu_{\text{out}}(\boldsymbol{x})}{\sigma(\boldsymbol{x})}\right) & \text{if } \hat{\mu}_{\text{in}}(\boldsymbol{x}) < \hat{\mu}_{\text{out}}(\boldsymbol{x}), \end{cases} \tag{A52}$$

*where $F_Z$ is the cdf of $t$ with the standard location and unit scale, assuming that the inverse of $F_Z$ exists.*

### A.6 Proof of Theorem 2

**Theorem 2** (Per-example LiRA power-law)**.** *Fix a target example $(\boldsymbol{x}, y)$. For the simplified model with arbitrary $C$ and infinitely many shadow models, the per-example LiRA vulnerability is given as*

$$\begin{aligned} \log(\text{TPR}_{\text{LiRA}}(\boldsymbol{x}) - \text{FPR}_{\text{LiRA}}(\boldsymbol{x})) \\ = -\frac{1}{2}\log S - \frac{1}{2}\Phi^{-1}(\text{FPR}_{\text{LiRA}}(\boldsymbol{x}))^2 + \log\frac{|\langle\boldsymbol{x}, \boldsymbol{x} - \boldsymbol{m}_{\boldsymbol{x}}\rangle|}{\sqrt{\boldsymbol{x}^T\Sigma\boldsymbol{x}}\sqrt{2\pi}} + \log(1 + \xi(S)), \quad (9) \end{aligned}$$

*where $\boldsymbol{m}_{\boldsymbol{x}}$ is the true mean of class $y$ and $\xi(S) = O(1/\sqrt{S})$. For large $S$ we have*

$$\log(\text{TPR}_{\text{LiRA}}(\boldsymbol{x}) - \text{FPR}_{\text{LiRA}}(\boldsymbol{x})) \approx -\frac{1}{2}\log S - \frac{1}{2}\Phi^{-1}(\text{FPR}_{\text{LiRA}}(\boldsymbol{x}))^2 + \log\frac{|\langle\boldsymbol{x}, \boldsymbol{x} - \boldsymbol{m}_{\boldsymbol{x}}\rangle|}{\sqrt{\boldsymbol{x}^T\Sigma\boldsymbol{x}}\sqrt{2\pi}}. \quad (10)$$

*Proof.* Let $\mathcal{D}_{\text{target}} = \{(\boldsymbol{x}_{j,1}, j), ..., (\boldsymbol{x}_{j,S}, j)\}_{j=1}^C$. Then the LiRA score of the target $(\boldsymbol{x}, y)$ is

$$s_y^{(\text{in})} = \langle\boldsymbol{x}, \frac{1}{S}\left(\sum_{i=1}^{S-1}\boldsymbol{x}_{y,i} + \boldsymbol{x}\right)\rangle = \langle\boldsymbol{x}, \frac{1}{S}\sum_{i=1}^{S}\boldsymbol{x}_{y,i}\rangle + \langle\boldsymbol{x}, \frac{1}{S}(\boldsymbol{x} - \boldsymbol{x}_{y,S})\rangle \tag{A53}$$

$$s_y^{(\text{out})} = \langle\boldsymbol{x}, \frac{1}{S}\sum_{i=1}^{S}\boldsymbol{x}_{y,i}\rangle, \tag{A54}$$

respectively, when $(\boldsymbol{x}, y) \in \mathcal{D}_{\text{target}}$ and when $(\boldsymbol{x}, y) \notin \mathcal{D}_{\text{target}}$. Thus we obtain

$$\mu_{\text{in}} - \mu_{\text{out}} = \mathbb{E}[s_y^{(\text{in})} - s_y^{(\text{out})}] = \frac{1}{S}\langle\boldsymbol{x}, \boldsymbol{x} - \boldsymbol{m}_{\boldsymbol{x}}\rangle \tag{A55}$$

$$\sigma^2 = \text{Var}(s_y^{(\text{out})}) = \frac{1}{S}\text{Var}(\langle\boldsymbol{x}, \boldsymbol{x}_{y,i}\rangle) = \frac{1}{S}\boldsymbol{x}^T\Sigma\boldsymbol{x}. \tag{A56}$$

Noting that the LiRA score follows a normal distribution, by Lemma 1 we have

$$\text{TPR}_{\text{LiRA}}(\boldsymbol{x}) = \Phi\left(\Phi^{-1}(\text{FPR}_{\text{LiRA}}(\boldsymbol{x})) + \frac{|\langle\boldsymbol{x}, \boldsymbol{x} - \boldsymbol{m}_{\boldsymbol{x}}\rangle|}{\sqrt{S}\sqrt{\boldsymbol{x}^T\Sigma\boldsymbol{x}}}\right), \tag{A57}$$

where $\Phi$ is the cdf of the standard normal distribution. This completes the first half of the theorem. Now let $\phi(u)$ denote the pdf of the standard normal distribution, and let

$$r = \frac{|\langle \boldsymbol{x}, \boldsymbol{x} - \boldsymbol{m_x}\rangle|}{\sqrt{S}\sqrt{\boldsymbol{x}^T \Sigma \boldsymbol{x}}}. \tag{A58}$$

Using Taylor expansion of $\Phi(\Phi^{-1}(\text{FPR}_{\text{LiRA}}(\boldsymbol{x})) + r)$ around $r = 0$, we have

$$\text{TPR}_{\text{LiRA}}(\boldsymbol{x}) = \sum_{k=0}^{\infty} \frac{\Phi^{(k)}(\Phi^{-1}(\text{FPR}_{\text{LiRA}}(\boldsymbol{x}))}{k!} r^k \tag{A59}$$

$$= \text{FPR}_{\text{LiRA}}(\boldsymbol{x}) + \sum_{k=1}^{\infty} \frac{\Phi^{(k)}(\Phi^{-1}(\text{FPR}_{\text{LiRA}}(\boldsymbol{x}))}{k!} r^k \tag{A60}$$

$$= \text{FPR}_{\text{LiRA}}(\boldsymbol{x}) + \sum_{k=1}^{\infty} \frac{\phi^{(k-1)}(\Phi^{-1}(\text{FPR}_{\text{LiRA}}(\boldsymbol{x}))}{k!} r^k \tag{A61}$$

$$= \text{FPR}_{\text{LiRA}}(\boldsymbol{x}) + \sum_{k=1}^{\infty} \frac{(-1)^{k-1}\text{He}_{k-1}(\Phi^{-1}(\text{FPR}_{\text{LiRA}}(\boldsymbol{x})))\phi(\Phi^{-1}(\text{FPR}_{\text{LiRA}}(\boldsymbol{x}))}{k!} r^k \tag{A62}$$

$$= \text{FPR}_{\text{LiRA}}(\boldsymbol{x}) + r\phi(\Phi^{-1}(\text{FPR}_{\text{LiRA}}(\boldsymbol{x})) \sum_{k=0}^{\infty} \frac{(-1)^k \text{He}_k(\Phi^{-1}(\text{FPR}_{\text{LiRA}}(\boldsymbol{x})))}{(k+1)!} r^k, \tag{A63}$$

where He denotes Hermite polynomials. It follows that

$$\log(\text{TPR}_{\text{LiRA}}(\boldsymbol{x}) - \text{FPR}_{\text{LiRA}}(\boldsymbol{x})) \tag{A64}$$

$$= \log r + \log \phi(\Phi^{-1}(\text{FPR}_{\text{LiRA}}(\boldsymbol{x})) + \log\left(\sum_{k=0}^{\infty} \frac{(-1)^k \text{He}_k(\Phi^{-1}(\text{FPR}_{\text{LiRA}}(\boldsymbol{x})))}{(k+1)!} r^k\right) \tag{A65}$$

$$= -\frac{1}{2}\log S - \frac{1}{2}\Phi^{-1}(\text{FPR}_{\text{LiRA}}(\boldsymbol{x}))^2 + \log \frac{|\langle \boldsymbol{x}, \boldsymbol{x} - \boldsymbol{m_x}\rangle|}{\sqrt{\boldsymbol{x}^T \Sigma \boldsymbol{x}}\sqrt{2\pi}} \tag{A66}$$

$$+ \log\left(\sum_{k=0}^{\infty} \frac{(-1)^k \text{He}_k(\Phi^{-1}(\text{FPR}_{\text{LiRA}}(\boldsymbol{x})))}{(k+1)!}\left(\frac{|\langle \boldsymbol{x}, \boldsymbol{x} - \boldsymbol{m_x}\rangle|}{\sqrt{S}\sqrt{\boldsymbol{x}^T \Sigma \boldsymbol{x}}}\right)^k\right) \tag{A67}$$

$$= -\frac{1}{2}\log S - \frac{1}{2}\Phi^{-1}(\text{FPR}_{\text{LiRA}}(\boldsymbol{x}))^2 + \log \frac{|\langle \boldsymbol{x}, \boldsymbol{x} - \boldsymbol{m_x}\rangle|}{\sqrt{\boldsymbol{x}^T \Sigma \boldsymbol{x}}\sqrt{2\pi}} \tag{A68}$$

$$+ \log\left(1 + \sum_{k=1}^{\infty} \frac{(-1)^k \text{He}_k(\Phi^{-1}(\text{FPR}_{\text{LiRA}}(\boldsymbol{x})))}{(k+1)!}\left(\frac{|\langle \boldsymbol{x}, \boldsymbol{x} - \boldsymbol{m_x}\rangle|}{\sqrt{S}\sqrt{\boldsymbol{x}^T \Sigma \boldsymbol{x}}}\right)^k\right) \tag{A69}$$

$$= -\frac{1}{2}\log S - \frac{1}{2}\Phi^{-1}(\text{FPR}_{\text{LiRA}}(\boldsymbol{x}))^2 + \log \frac{|\langle \boldsymbol{x}, \boldsymbol{x} - \boldsymbol{m_x}\rangle|}{\sqrt{\boldsymbol{x}^T \Sigma \boldsymbol{x}}\sqrt{2\pi}} + \log(1 + \xi(S)), \tag{A70}$$

where we have $\xi(S) = O(1/\sqrt{S})$. For large enough $S$, ignoring the residual term, we approximate

$$\log(\text{TPR}_{\text{LiRA}}(\boldsymbol{x}) - \text{FPR}_{\text{LiRA}}(\boldsymbol{x})) \approx -\frac{1}{2}\log S - \frac{1}{2}\Phi^{-1}(\text{FPR}_{\text{LiRA}}(\boldsymbol{x}))^2 + \log \frac{|\langle \boldsymbol{x}, \boldsymbol{x} - \boldsymbol{m_x}\rangle|}{\sqrt{\boldsymbol{x}^T \Sigma \boldsymbol{x}}\sqrt{2\pi}}. \tag{A71}$$

$\square$

## A.7 Proof of Corollary 4

**Corollary 4** (Average-case LiRA power-law)**.** *For the simplified model with arbitrary $C$, sufficiently large $S$ and infinitely many shadow models, we have*

$$\log(\overline{\text{TPR}}_{\text{LiRA}} - \overline{\text{FPR}}_{\text{LiRA}}) \approx -\frac{1}{2}\log S - \frac{1}{2}\Phi^{-1}(\overline{\text{FPR}}_{\text{LiRA}})^2 + \log\left(\mathbb{E}_{(\boldsymbol{x},y)\sim\mathbb{D}}\left[\frac{|\langle \boldsymbol{x}, \boldsymbol{x} - \boldsymbol{m_x}\rangle|}{\sqrt{\boldsymbol{x}^T \Sigma \boldsymbol{x}}\sqrt{2\pi}}\right]\right). \tag{12}$$

*Proof.* By theorem 2 and the law of unconscious statistician, we have for large $S$

$$\overline{\text{TPR}}_{\text{LiRA}} - \overline{\text{FPR}}_{\text{LiRA}} = \int_{\mathscr{D}} p(\boldsymbol{x})(\text{TPR}_{\text{LiRA}}(\boldsymbol{x}) - \text{FPR}_{\text{LiRA}}(\boldsymbol{x}))\mathrm{d}\boldsymbol{x} \tag{A72}$$

$$\approx \int_{\mathscr{D}} p(\boldsymbol{x}) \frac{1}{\sqrt{2\pi}} e^{-\frac{1}{2}\Phi^{-1}(\overline{\text{FPR}}_{\text{LiRA}})^2} \frac{\langle \boldsymbol{x}, \boldsymbol{x} - \boldsymbol{m_x} \rangle}{\sqrt{S}\sqrt{\boldsymbol{x}^T \Sigma \boldsymbol{x}}} \mathrm{d}\boldsymbol{x} \tag{A73}$$

$$= \frac{1}{\sqrt{2\pi}} e^{-\frac{1}{2}\Phi^{-1}(\overline{\text{FPR}}_{\text{LiRA}})^2} \frac{1}{\sqrt{S}} \int_{\mathscr{D}} p(\boldsymbol{x}) \frac{\langle \boldsymbol{x}, \boldsymbol{x} - \boldsymbol{m_x} \rangle}{\sqrt{\boldsymbol{x}^T \Sigma \boldsymbol{x}}} \mathrm{d}\boldsymbol{x} \tag{A74}$$

$$= \frac{1}{\sqrt{S}} e^{-\frac{1}{2}\Phi^{-1}(\overline{\text{FPR}}_{\text{LiRA}})^2} \mathbb{E}_{(\boldsymbol{x},y)\sim\mathbb{D}} \left[ \frac{\langle \boldsymbol{x}, \boldsymbol{x} - \boldsymbol{m_x} \rangle}{\sqrt{2\pi}\sqrt{\boldsymbol{x}^T \Sigma \boldsymbol{x}}} \right], \tag{A75}$$

where $p(\boldsymbol{x})$ is the density of $\mathbb{D}$ at $(\boldsymbol{x}, y)$, and $\mathscr{D}$ is the data domain. Note that here we fixed $\text{FPR}_{\text{LiRA}}(\boldsymbol{x}) = \overline{\text{FPR}}_{\text{LiRA}}$ for all $\boldsymbol{x}$. Then we obtain

$$\log(\overline{\text{TPR}}_{\text{LiRA}} - \overline{\text{FPR}}_{\text{LiRA}}) \approx -\frac{1}{2}\log S - \frac{1}{2}\Phi^{-1}(\overline{\text{FPR}}_{\text{LiRA}})^2 + \log\left(\mathbb{E}_{(\boldsymbol{x},y)\sim\mathbb{D}}\left[\frac{\langle \boldsymbol{x}, \boldsymbol{x} - \boldsymbol{m_x} \rangle}{\sqrt{2\pi}\sqrt{\boldsymbol{x}^T \Sigma \boldsymbol{x}}}\right]\right). \tag{A76}$$

$\square$

# B  Theoretical analysis of RMIA

Similar to LiRA, RMIA is based on shadow model training and computing the attack statistics based on a likelihood ratio. The main difference to LiRA is that RMIA does not compute the likelihood ratio based on aggregated IN/OUT statistics, but instead compares the target data point against random samples $(\boldsymbol{z}, y_{\boldsymbol{z}})$ from the target data distribution. After computing the likelihood ratios over multiple $(\boldsymbol{z}, y_{\boldsymbol{z}})$ values, the MIA score is estimated as a proportion of the ratios exceeding a preset bound. This approach makes RMIA a more effective attack when the number of shadow models is low.

## B.1  Formulating RMIA

In the following, let us denote a target point by $(\boldsymbol{x}, y_{\boldsymbol{x}})$ with $y_{\boldsymbol{x}}$ being the label of $\boldsymbol{x}$. Let us restate how RMIA (Zarifzadeh et al., 2024) builds the MIA score. RMIA augments the likelihood-ratio with a sample from the target data distribution to calibrate how likely you would obtain the target model if $(\boldsymbol{x}, y_{\boldsymbol{x}})$ is replaced with another sample $(\boldsymbol{z}, y_{\boldsymbol{z}})$. Denoting the target model parameters with $\theta$, RMIA computes the *pairwise* likelihood ratio

$$\text{LR}(\boldsymbol{x}, \boldsymbol{z}) = \frac{p(\theta \mid \boldsymbol{x}, y_{\boldsymbol{x}})}{p(\theta \mid \boldsymbol{z}, y_{\boldsymbol{z}})}, \tag{A77}$$

and the corresponding MIA score is given as

$$\text{Score}_{\text{RMIA}}(\boldsymbol{x}) = \Pr_{(\boldsymbol{z}, y_{\boldsymbol{z}})\sim\mathbb{D}} (\text{LR}(\boldsymbol{x}, \boldsymbol{z}) > \gamma), \tag{A78}$$

where $\mathbb{D}$ denotes the training data distribution. Zarifzadeh et al. (2024) show two approaches to compute $\text{LR}(\boldsymbol{x}, \boldsymbol{z})$: the direct approach and the effecient Bayesian approach. In the following theoretical analysis, we focus on the direct approach that is an approximation of the efficient Bayesian approach, as Zarifzadeh et al. (2024) empirically demonstrates that these approaches exhibit similar performances.

Let $\hat{\mu}_{\boldsymbol{a},\boldsymbol{b}}$ and $\hat{\sigma}_{\boldsymbol{a},\boldsymbol{b}}$ denote, respectively, the mean and standard deviation of $t_{\boldsymbol{b}} = \ell(\mathcal{M}(\boldsymbol{b}), y_{\boldsymbol{b}})$ estimated from shadow models, where $\boldsymbol{a}$ denotes which of $(\boldsymbol{x}, y_{\boldsymbol{x}})$ and $(\boldsymbol{z}, y_{\boldsymbol{z}})$ is in the training set. By Equation 11 in (Zarifzadeh et al., 2024), the pairwise likelihood ratio is given as

$$\text{LR}(\boldsymbol{x}, \boldsymbol{z}) = \frac{p(\theta \mid \boldsymbol{x}, y_{\boldsymbol{x}})}{p(\theta \mid \boldsymbol{z}, y_{\boldsymbol{z}})} \approx \frac{\mathcal{N}(t_{\boldsymbol{x}}; \hat{\mu}_{\boldsymbol{x},\boldsymbol{x}}, \hat{\sigma}_{\boldsymbol{x},\boldsymbol{x}}^2)\mathcal{N}(t_{\boldsymbol{z}}; \hat{\mu}_{\boldsymbol{x},\boldsymbol{z}}, \hat{\sigma}_{\boldsymbol{x},\boldsymbol{z}}^2)}{\mathcal{N}(t_{\boldsymbol{x}}; \hat{\mu}_{\boldsymbol{z},\boldsymbol{x}}, \hat{\sigma}_{\boldsymbol{z},\boldsymbol{x}}^2)\mathcal{N}(t_{\boldsymbol{z}}; \hat{\mu}_{\boldsymbol{z},\boldsymbol{z}}, \hat{\sigma}_{\boldsymbol{z},\boldsymbol{z}}^2)}, \tag{A79}$$

where $\hat{\mu}_{\boldsymbol{a},\boldsymbol{b}}$ and $\hat{\sigma}_{\boldsymbol{a},\boldsymbol{b}}$ are, respectively, the mean and standard deviation of $t_b$ estimated from shadow models when the training set contains $\boldsymbol{a}$ but not $\boldsymbol{b}$. Then RMIA exploits the probability of rejecting the pairwise likelihood ratio test over $(\boldsymbol{z}, y_{\boldsymbol{z}}) \sim \mathbb{D}$:

$$\text{Score}_{\text{RMIA}}(\boldsymbol{x}) = \Pr_{(\boldsymbol{z}, y_{\boldsymbol{z}})\sim\mathbb{D}} (\text{LR}(\boldsymbol{x}, \boldsymbol{z}) \geq \gamma), \tag{A80}$$

where $\mathbb{D}$ is the underlying data distribution. Similar to LiRA, the classifier is built by checking if $\text{Score}_{\text{RMIA}}(\boldsymbol{x}) > \tau$. In the following, we will use the direct computation of likelihood-ratio as described in Equation 11 of Zarifzadeh et al. (2024) which approximates $\text{LR}(\boldsymbol{x}, \boldsymbol{z})$ using normal distributions. Thus, RMIA rejects $H_0$ if and only if

$$\Pr_{(\boldsymbol{z}, y_{\boldsymbol{z}}) \sim \mathbb{D}} (\text{LR}(\boldsymbol{x}, \boldsymbol{z}) \geq \gamma) \geq \tau, \tag{A81}$$

identifying the membership of $\boldsymbol{x}$. Hence the true positive rate of per-example RMIA is given as

$$
\begin{aligned}
&\text{TPR}_{\text{RMIA}}(\boldsymbol{x}) \\
&= \Pr_{\mathcal{D}_{\text{target}} \sim \mathbb{D}^{|\mathcal{D}|}, \phi^M} \left( \Pr_{(\boldsymbol{z}, y_{\boldsymbol{z}}) \sim \mathbb{D}} (\text{LR}(\boldsymbol{x}, \boldsymbol{z}) \geq \gamma) \geq \tau \mid (\boldsymbol{x}, y_{\boldsymbol{x}}) \in \mathcal{D}_{\text{target}} \wedge (\boldsymbol{x}, y_{\boldsymbol{x}}) \notin \mathcal{D}_{\text{target}} \right),
\end{aligned} \tag{A82}
$$

where $\phi^M$ denotes the randomness in the shadow set sampling and shadow model training.

We define the average-case TPR for RMIA by taking the expectation over the data distribution:

$$\overline{\text{TPR}}_{\text{RMIA}} = \mathbb{E}_{(\boldsymbol{x}, y_{\boldsymbol{x}}) \sim \mathbb{D}}[\text{TPR}_{\text{RMIA}}(\boldsymbol{x})]. \tag{A83}$$

## B.2 Per-example RMIA

Next we focus on the per-example RMIA performance. As in the case of LiRA, we assume that $t_{\boldsymbol{x}}$ and $t_{\boldsymbol{z}}$ follow distributions of the location-scale family. We have

$$t_{\boldsymbol{x}} = \begin{cases} \mu_{\boldsymbol{x}, \boldsymbol{x}} + \sigma_{\boldsymbol{x}, \boldsymbol{x}} Z & \text{if } (\boldsymbol{x}, y_{\boldsymbol{x}}) \in \mathcal{D}_{\text{target}} \wedge (\boldsymbol{z}, y_{\boldsymbol{z}}) \notin \mathcal{D}_{\text{target}} \\ \mu_{\boldsymbol{z}, \boldsymbol{x}} + \sigma_{\boldsymbol{z}, \boldsymbol{x}} Z & \text{if } (\boldsymbol{x}, y_{\boldsymbol{x}}) \notin \mathcal{D}_{\text{target}} \wedge (\boldsymbol{z}, y_{\boldsymbol{z}}) \in \mathcal{D}_{\text{target}} \end{cases} \tag{A84}$$

$$t_{\boldsymbol{z}} = \begin{cases} \mu_{\boldsymbol{x}, \boldsymbol{z}} + \sigma_{\boldsymbol{x}, \boldsymbol{z}} Z & \text{if } (\boldsymbol{x}, y_{\boldsymbol{x}}) \in \mathcal{D}_{\text{target}} \wedge (\boldsymbol{z}, y_{\boldsymbol{z}}) \notin \mathcal{D}_{\text{target}} \\ \mu_{\boldsymbol{z}, \boldsymbol{z}} + \sigma_{\boldsymbol{z}, \boldsymbol{z}} Z & \text{if } (\boldsymbol{x}, y_{\boldsymbol{x}}) \notin \mathcal{D}_{\text{target}} \wedge (\boldsymbol{z}, y_{\boldsymbol{z}}) \in \mathcal{D}_{\text{target}}. \end{cases} \tag{A85}$$

where $Z$ follows a distribution of location-scale family with the standard location and unit scale. It is important to note that $\mu_{\boldsymbol{a}, \boldsymbol{b}}$ and $\sigma_{\boldsymbol{a}, \boldsymbol{b}}$ denote, respectively, a location and a scale, while previously defined $\hat{\mu}_{\boldsymbol{a}, \boldsymbol{b}}$ and $\hat{\sigma}_{\boldsymbol{a}, \boldsymbol{b}}$ are, respectively, a mean and a standard deviation. As for the analysis of LiRA, we assume that the target and shadow sets have a sufficient number of examples per class, and that $\sigma_{\boldsymbol{x}} = \sigma_{\boldsymbol{x}, \boldsymbol{x}} = \sigma_{\boldsymbol{z}, \boldsymbol{x}}, \sigma_{\boldsymbol{z}} = \sigma_{\boldsymbol{x}, \boldsymbol{z}} = \sigma_{\boldsymbol{z}, \boldsymbol{z}}, \hat{\sigma}_{\boldsymbol{x}} = \hat{\sigma}_{\boldsymbol{x}, \boldsymbol{x}} = \hat{\sigma}_{\boldsymbol{z}, \boldsymbol{x}}$ and $\hat{\sigma}_{\boldsymbol{z}} = \hat{\sigma}_{\boldsymbol{x}, \boldsymbol{z}} = \hat{\sigma}_{\boldsymbol{z}, \boldsymbol{z}}$, where $\sigma_{\boldsymbol{x}}$ and $\sigma_{\boldsymbol{z}}$ are, respectively, the true scales of $t_{\boldsymbol{x}}$ and $t_{\boldsymbol{z}}$, and $\hat{\sigma}_{\boldsymbol{x}}$ and $\hat{\sigma}_{\boldsymbol{z}}$ are, respectively, standard deviations of $t_{\boldsymbol{x}}$ and $t_{\boldsymbol{z}}$ estimated from shadow models. Similarly to the case of LiRA, these assumptions are justified using the simplified model. We have in the simplified model

$$\sigma_{\boldsymbol{x}, \boldsymbol{x}}^2 = \hat{\sigma}_{\boldsymbol{x}, \boldsymbol{x}}^2 = \text{Var}(s_{y_{\boldsymbol{x}}}^{(\boldsymbol{x})}(\boldsymbol{x})) = \frac{1}{S} \left( 1 - \frac{1}{S} \right) \boldsymbol{x}^T \Sigma \boldsymbol{x} \tag{A86}$$

$$\sigma_{\boldsymbol{z}, \boldsymbol{x}}^2 = \hat{\sigma}_{\boldsymbol{z}, \boldsymbol{x}}^2 = \text{Var}(s_{y_{\boldsymbol{x}}}^{(\boldsymbol{z})}(\boldsymbol{x})) = \begin{cases} \frac{1}{S} \left( 1 - \frac{1}{S} \right) \boldsymbol{x}^T \Sigma \boldsymbol{x} & \text{if } y_{\boldsymbol{x}} = y_{\boldsymbol{z}} \\ \frac{1}{S} \boldsymbol{x}^T \Sigma \boldsymbol{x} & \text{if } y_{\boldsymbol{x}} \neq y_{\boldsymbol{z}} \end{cases} \tag{A87}$$

$$\sigma_{\boldsymbol{x}, \boldsymbol{z}}^2 = \hat{\sigma}_{\boldsymbol{x}, \boldsymbol{z}}^2 = \text{Var}(s_{y_{\boldsymbol{z}}}^{(\boldsymbol{x})}(\boldsymbol{z})) = \begin{cases} \frac{1}{S} \left( 1 - \frac{1}{S} \right) \boldsymbol{z}^T \Sigma \boldsymbol{z} & \text{if } y_{\boldsymbol{x}} = y_{\boldsymbol{z}} \\ \frac{1}{S} \boldsymbol{z}^T \Sigma \boldsymbol{z} & \text{if } y_{\boldsymbol{x}} \neq y_{\boldsymbol{z}} \end{cases} \tag{A88}$$

$$\sigma_{\boldsymbol{z}, \boldsymbol{z}}^2 = \hat{\sigma}_{\boldsymbol{z}, \boldsymbol{z}}^2 = \text{Var}(s_{y_{\boldsymbol{z}}}^{(\boldsymbol{z})}(\boldsymbol{z})) = \frac{1}{S} \left( 1 - \frac{1}{S} \right) \boldsymbol{z}^T \Sigma \boldsymbol{z}. \tag{A89}$$

Therefore, the differences $\sigma_{\boldsymbol{x}, \boldsymbol{x}} - \sigma_{\boldsymbol{z}, \boldsymbol{x}}, \sigma_{\boldsymbol{x}, \boldsymbol{z}} - \sigma_{\boldsymbol{z}, \boldsymbol{z}}, \hat{\sigma}_{\boldsymbol{x}, \boldsymbol{x}} - \hat{\sigma}_{\boldsymbol{z}, \boldsymbol{x}}$ and $\hat{\sigma}_{\boldsymbol{x}, \boldsymbol{z}} - \hat{\sigma}_{\boldsymbol{z}, \boldsymbol{z}}$ are negligible for large enough $S$.

Now we derive the per-example RMIA vulnerability in terms of RMIA statistics computed from shadow models.

**Lemma A4** (Per-example RMIA vulnerability). *Suppose that the true IN/OUT distributions of $t_{\boldsymbol{x}}$ and $t_{\boldsymbol{z}}$ are of location-scale family with locations $\mu_{\boldsymbol{x}, \boldsymbol{x}}, \mu_{\boldsymbol{z}, \boldsymbol{x}}, \mu_{\boldsymbol{x}, \boldsymbol{z}}, \mu_{\boldsymbol{z}, \boldsymbol{z}}$ and scales $\sigma_{\boldsymbol{x}}, \sigma_{\boldsymbol{z}}$. Assume that RMIA models $t_{\boldsymbol{x}}$ and $t_{\boldsymbol{z}}$ by normal distributions with parameters computed from shadow models, and that an attacker has access to the underlying distribution $\mathbb{D}$. Then with infinitely many shadow models, the RMIA vulnerability of a fixed target example $(\boldsymbol{x}, y_{\boldsymbol{x}})$ satisfies*

$$\text{TPR}_{\text{RMIA}}(\boldsymbol{x}) \leq 1 - F_{|Z|} \left( F_{|Z|}^{-1}(1 - \alpha) - \frac{\mathbb{E}_{(\boldsymbol{z}, y_{\boldsymbol{z}}) \sim \mathbb{D}}[|q|]}{\mathbb{E}_{(\boldsymbol{z}, y_{\boldsymbol{z}}) \sim \mathbb{D}}[|A|]} \right) \tag{A90}$$

*for some constant* $\alpha \geq \text{FPR}_{\text{RMIA}}(\boldsymbol{x})$, *where* $F_{|Z|}$ *is the cdf of* $|Z|$ *and*

$$q = \frac{(\mu_{\boldsymbol{x},\boldsymbol{x}} - \mu_{\boldsymbol{z},\boldsymbol{x}})(\hat{\mu}_{\boldsymbol{x},\boldsymbol{x}} - \hat{\mu}_{\boldsymbol{z},\boldsymbol{x}})}{\hat{\sigma}_{\boldsymbol{x}}^2} + \frac{(\mu_{\boldsymbol{x},\boldsymbol{z}} - \mu_{\boldsymbol{z},\boldsymbol{z}})(\hat{\mu}_{\boldsymbol{x},\boldsymbol{z}} - \hat{\mu}_{\boldsymbol{z},\boldsymbol{z}})}{\hat{\sigma}_{\boldsymbol{z}}^2} \tag{A91}$$

$$A = \frac{\sigma_{\boldsymbol{x}}}{\hat{\sigma}_{\boldsymbol{x}}^2}(\hat{\mu}_{\boldsymbol{x},\boldsymbol{x}} - \hat{\mu}_{\boldsymbol{z},\boldsymbol{x}}) + \frac{\sigma_{\boldsymbol{z}}}{\hat{\sigma}_{\boldsymbol{z}}^2}(\hat{\mu}_{\boldsymbol{x},\boldsymbol{z}} - \hat{\mu}_{\boldsymbol{z},\boldsymbol{z}}), \tag{A92}$$

*assuming that the inverse of* $F_{|Z|}$ *exists.*

*Proof.* We abuse notations by denoting probabilities and expectations over sampling $\mathcal{D}_{\text{target}} \sim \mathbb{D}^n$ and $(\boldsymbol{z}, y_{\boldsymbol{z}}) \sim \mathbb{D}$ by subscripts $\mathcal{D}_{\text{target}}$ and $\boldsymbol{z}$. We have in the limit of infinitely many shadow models

$$\text{TPR}_{\text{RMIA}}(\boldsymbol{x}) = \Pr_{\mathcal{D}_{\text{target}}} \left( \Pr_{\boldsymbol{z}} \left( \text{LR}(\boldsymbol{x}, \boldsymbol{z}) \geq \gamma \right) \geq \tau \mid (\boldsymbol{x}, y_{\boldsymbol{x}}) \in \mathcal{D}_{\text{target}} \wedge (\boldsymbol{z}, y_{\boldsymbol{z}}) \notin \mathcal{D}_{\text{target}} \right) \tag{A93}$$

$$\text{FPR}_{\text{RMIA}}(\boldsymbol{x}) = \Pr_{\mathcal{D}_{\text{target}}} \left( \Pr_{\boldsymbol{z}} \left( \text{LR}(\boldsymbol{x}, \boldsymbol{z}) \geq \gamma \right) \geq \tau \mid (\boldsymbol{x}, y_{\boldsymbol{x}}) \notin \mathcal{D}_{\text{target}} \wedge (\boldsymbol{z}, y_{\boldsymbol{z}}) \in \mathcal{D}_{\text{target}} \right), \tag{A94}$$

where

$$\text{LR}(\boldsymbol{x}, \boldsymbol{z}) = \frac{\mathcal{N}(t_{\boldsymbol{x}}; \hat{\mu}_{\boldsymbol{x},\boldsymbol{x}}, \hat{\sigma}_{\boldsymbol{x}}^2)\mathcal{N}(t_{\boldsymbol{z}}; \hat{\mu}_{\boldsymbol{x},\boldsymbol{z}}, \hat{\sigma}_{\boldsymbol{z}}^2)}{\mathcal{N}(t_{\boldsymbol{x}}; \hat{\mu}_{\boldsymbol{z},\boldsymbol{x}}, \hat{\sigma}_{\boldsymbol{x}}^2)\mathcal{N}(t_{\boldsymbol{z}}; \hat{\mu}_{\boldsymbol{z},\boldsymbol{z}}, \hat{\sigma}_{\boldsymbol{z}}^2)}. \tag{A95}$$

Note that the probabilities are over target dataset sampling in the limit of infinitely many shadow models. We have

$$\lambda := \log(\text{LR}(\boldsymbol{x}, \boldsymbol{z})) \tag{A96}$$

$$= \log \left( \frac{\frac{1}{\sqrt{2\pi}\hat{\sigma}_{\boldsymbol{x}}} \exp\left(-\frac{1}{2}\left(\frac{t_{\boldsymbol{x}} - \hat{\mu}_{\boldsymbol{x},\boldsymbol{x}}}{\hat{\sigma}_{\boldsymbol{x}}}\right)^2\right) \frac{1}{\sqrt{2\pi}\hat{\sigma}_{\boldsymbol{z}}} \exp\left(-\frac{1}{2}\left(\frac{t_{\boldsymbol{z}} - \hat{\mu}_{\boldsymbol{x},\boldsymbol{z}}}{\hat{\sigma}_{\boldsymbol{z}}}\right)^2\right)}{\frac{1}{\sqrt{2\pi}\hat{\sigma}_{\boldsymbol{x}}} \exp\left(-\frac{1}{2}\left(\frac{t_{\boldsymbol{x}} - \hat{\mu}_{\boldsymbol{z},\boldsymbol{x}}}{\hat{\sigma}_{\boldsymbol{x}}}\right)^2\right) \frac{1}{\sqrt{2\pi}\hat{\sigma}_{\boldsymbol{z}}} \exp\left(-\frac{1}{2}\left(\frac{t_{\boldsymbol{z}} - \hat{\mu}_{\boldsymbol{z},\boldsymbol{z}}}{\hat{\sigma}_{\boldsymbol{z}}}\right)^2\right)} \right) \tag{A97}$$

$$= -\frac{1}{2}\left(\frac{t_{\boldsymbol{x}} - \hat{\mu}_{\boldsymbol{x},\boldsymbol{x}}}{\hat{\sigma}_{\boldsymbol{x}}}\right)^2 + \frac{1}{2}\left(\frac{t_{\boldsymbol{x}} - \hat{\mu}_{\boldsymbol{z},\boldsymbol{x}}}{\hat{\sigma}_{\boldsymbol{x}}}\right)^2 - \frac{1}{2}\left(\frac{t_{\boldsymbol{z}} - \hat{\mu}_{\boldsymbol{x},\boldsymbol{z}}}{\hat{\sigma}_{\boldsymbol{z}}}\right)^2 + \frac{1}{2}\left(\frac{t_{\boldsymbol{z}} - \hat{\mu}_{\boldsymbol{z},\boldsymbol{z}}}{\hat{\sigma}_{\boldsymbol{z}}}\right)^2 \tag{A98}$$

$$= \frac{1}{2\hat{\sigma}_{\boldsymbol{x}}^2}(2t_{\boldsymbol{x}}\hat{\mu}_{\boldsymbol{x},\boldsymbol{x}} - \hat{\mu}_{\boldsymbol{x},\boldsymbol{x}}^2 - 2t_{\boldsymbol{x}}\hat{\mu}_{\boldsymbol{z},\boldsymbol{x}} + \hat{\mu}_{\boldsymbol{z},\boldsymbol{x}}^2) + \frac{1}{2\hat{\sigma}_{\boldsymbol{z}}^2}(2t_{\boldsymbol{z}}\hat{\mu}_{\boldsymbol{x},\boldsymbol{z}} - \hat{\mu}_{\boldsymbol{x},\boldsymbol{z}}^2 - 2t_{\boldsymbol{z}}\hat{\mu}_{\boldsymbol{z},\boldsymbol{z}} + \hat{\mu}_{\boldsymbol{z},\boldsymbol{z}}^2) \tag{A99}$$

$$= \frac{\hat{\mu}_{\boldsymbol{x},\boldsymbol{x}} - \hat{\mu}_{\boldsymbol{z},\boldsymbol{x}}}{2\hat{\sigma}_{\boldsymbol{x}}^2}(2t_{\boldsymbol{x}} - \hat{\mu}_{\boldsymbol{x},\boldsymbol{x}} - \hat{\mu}_{\boldsymbol{z},\boldsymbol{x}}) + \frac{\hat{\mu}_{\boldsymbol{x},\boldsymbol{z}} - \hat{\mu}_{\boldsymbol{z},\boldsymbol{z}}}{2\hat{\sigma}_{\boldsymbol{z}}^2}(2t_{\boldsymbol{z}} - \hat{\mu}_{\boldsymbol{x},\boldsymbol{z}} - \hat{\mu}_{\boldsymbol{z},\boldsymbol{z}}). \tag{A100}$$

When $(\boldsymbol{x}, y_{\boldsymbol{x}}) \in \mathcal{D}_{\text{target}}$ and $(\boldsymbol{z}, y_{\boldsymbol{z}}) \notin \mathcal{D}_{\text{target}}$, $t_{\boldsymbol{x}} = \mu_{\boldsymbol{x},\boldsymbol{x}} + \sigma_{\boldsymbol{x}} Z$ and $t_{\boldsymbol{z}} = \mu_{\boldsymbol{x},\boldsymbol{z}} + \sigma_{\boldsymbol{z}} Z$. Thus, $\lambda$ becomes

$$\lambda_{\boldsymbol{x}} := \frac{\hat{\mu}_{\boldsymbol{x},\boldsymbol{x}} - \hat{\mu}_{\boldsymbol{z},\boldsymbol{x}}}{2\hat{\sigma}_{\boldsymbol{x}}^2}(2\mu_{\boldsymbol{x},\boldsymbol{x}} + 2\sigma_{\boldsymbol{x}} Z - \hat{\mu}_{\boldsymbol{x},\boldsymbol{x}} - \hat{\mu}_{\boldsymbol{z},\boldsymbol{x}}) \tag{A101}$$

$$+ \frac{\hat{\mu}_{\boldsymbol{x},\boldsymbol{z}} - \hat{\mu}_{\boldsymbol{z},\boldsymbol{z}}}{2\hat{\sigma}_{\boldsymbol{z}}^2}(2\mu_{\boldsymbol{x},\boldsymbol{z}} + 2\sigma_{\boldsymbol{z}} Z - \hat{\mu}_{\boldsymbol{x},\boldsymbol{z}} - \hat{\mu}_{\boldsymbol{z},\boldsymbol{z}}). \tag{A102}$$

Similarly, when $(x, y_{\boldsymbol{x}}) \notin \mathcal{D}_{\text{target}}$ and $(z, y_{\boldsymbol{z}}) \in \mathcal{D}_{\text{target}}$, $t_{\boldsymbol{x}} = \mu_{\boldsymbol{z},\boldsymbol{x}} + \sigma_{\boldsymbol{x}} Z$ and $t_{\boldsymbol{z}} = \mu_{\boldsymbol{z},\boldsymbol{x}} + \sigma_{\boldsymbol{z}} Z$. Then $\lambda$ becomes

$$\lambda_{\boldsymbol{z}} := \frac{\hat{\mu}_{\boldsymbol{x},\boldsymbol{x}} - \hat{\mu}_{\boldsymbol{z},\boldsymbol{x}}}{2\hat{\sigma}_{\boldsymbol{x}}^2}(2\mu_{\boldsymbol{z},\boldsymbol{x}} + 2\sigma_{\boldsymbol{x}} Z - \hat{\mu}_{\boldsymbol{x},\boldsymbol{x}} - \hat{\mu}_{\boldsymbol{z},\boldsymbol{x}}) \tag{A103}$$

$$+ \frac{\hat{\mu}_{\boldsymbol{x},\boldsymbol{z}} - \hat{\mu}_{\boldsymbol{z},\boldsymbol{z}}}{2\hat{\sigma}_{\boldsymbol{z}}^2}(2\mu_{\boldsymbol{z},\boldsymbol{z}} + 2\sigma_{\boldsymbol{z}} Z - \hat{\mu}_{\boldsymbol{x},\boldsymbol{z}} - \hat{\mu}_{\boldsymbol{z},\boldsymbol{z}}) \tag{A104}$$

$$= \left( \underbrace{\frac{\sigma_{\boldsymbol{x}}}{\hat{\sigma}_{\boldsymbol{x}}^2}(\hat{\mu}_{\boldsymbol{x},\boldsymbol{x}} - \hat{\mu}_{\boldsymbol{z},\boldsymbol{x}}) + \frac{\sigma_{\boldsymbol{z}}}{\hat{\sigma}_{\boldsymbol{z}}^2}(\hat{\mu}_{\boldsymbol{x},\boldsymbol{z}} - \hat{\mu}_{\boldsymbol{z},\boldsymbol{z}})}_{A} \right) Z \tag{A105}$$

$$+ \underbrace{\frac{\hat{\mu}_{\boldsymbol{x},\boldsymbol{x}} - \hat{\mu}_{\boldsymbol{z},\boldsymbol{x}}}{2\hat{\sigma}_{\boldsymbol{x}}^2}(2\mu_{\boldsymbol{z},\boldsymbol{x}} - \hat{\mu}_{\boldsymbol{x},\boldsymbol{x}} - \hat{\mu}_{\boldsymbol{z},\boldsymbol{x}}) + \frac{\hat{\mu}_{\boldsymbol{x},\boldsymbol{z}} - \hat{\mu}_{\boldsymbol{z},\boldsymbol{z}}}{2\hat{\sigma}_{\boldsymbol{z}}^2}(2\mu_{\boldsymbol{z},\boldsymbol{z}} - \hat{\mu}_{\boldsymbol{x},\boldsymbol{z}} - \hat{\mu}_{\boldsymbol{z},\boldsymbol{z}})}_{B} \tag{A106}$$

$$= AZ + B. \tag{A107}$$

Notice that $A$ and $B$ are functions of $\boldsymbol{z}$ and independent of $Z$. Also note that taking probability over $Z$ corresponds to calculating probability over $\mathcal{D}_{\text{target}} \sim \mathbb{D}^n$. Thus, since we can set $\gamma > 1$, using Markov's inequality and the triangle inequality, we have in the limit of infinitely many shadow models

$$\text{FPR}_{\text{RMIA}}(x) = \Pr_Z \left( \Pr_{\boldsymbol{z}}(\lambda_{\boldsymbol{z}} \geq \log \gamma) \geq \tau \right) \tag{A108}$$

$$\leq \Pr_Z \left( \Pr_{\boldsymbol{z}}(|\lambda_{\boldsymbol{z}}| \geq \log \gamma) \geq \tau \right) \tag{A109}$$

$$\leq \Pr_Z \left( \frac{\mathbb{E}_{\boldsymbol{z}}[|\lambda_{\boldsymbol{z}}|]}{\log \gamma} \geq \tau \right) \tag{A110}$$

$$= \Pr_Z \left( \mathbb{E}_{\boldsymbol{z}}[|\lambda_{\boldsymbol{z}}|] \geq \tau \log \gamma \right) \tag{A111}$$

$$\leq \Pr_Z \left( \mathbb{E}_{\boldsymbol{z}}[|A|]|Z| + \mathbb{E}_{\boldsymbol{z}}[|B|] \geq \tau \log \gamma \right) \tag{A112}$$

$$= \Pr_Z \left( |Z| \geq \frac{\tau \log \gamma - \mathbb{E}_{\boldsymbol{z}}[|B|]}{\mathbb{E}_{\boldsymbol{z}}[|A|]} \right) \tag{A113}$$

Therefore, we can upper-bound $\text{FPR}_{\text{RMIA}}(x) \leq \alpha$ by setting

$$\alpha = 1 - F_{|Z|} \left( \frac{\tau \log \gamma - \mathbb{E}_{\boldsymbol{z}}[|B|]}{\mathbb{E}_{\boldsymbol{z}}[|A|]} \right). \tag{A114}$$

That is,

$$\frac{\tau \log \gamma - \mathbb{E}_{\boldsymbol{z}}[|B|]}{\mathbb{E}_{\boldsymbol{z}}[|A|]} = F_{|Z|}^{-1}(1 - \alpha). \tag{A115}$$

Now let

$$q = \lambda_{\boldsymbol{x}} - \lambda_{\boldsymbol{z}} = \frac{(\mu_{\boldsymbol{x},\boldsymbol{x}} - \mu_{\boldsymbol{z},\boldsymbol{x}})(\hat{\mu}_{\boldsymbol{x},\boldsymbol{x}} - \hat{\mu}_{\boldsymbol{z},\boldsymbol{x}})}{\hat{\sigma}_{\boldsymbol{x}}^2} + \frac{(\mu_{\boldsymbol{x},\boldsymbol{z}} - \mu_{\boldsymbol{z},\boldsymbol{z}})(\hat{\mu}_{\boldsymbol{x},\boldsymbol{z}} - \hat{\mu}_{\boldsymbol{z},\boldsymbol{z}})}{\hat{\sigma}_{\boldsymbol{z}}^2}. \tag{A116}$$

Note that $q$ is also independent of $t$, thereby $\mathbb{E}_{\boldsymbol{z}}[|q|]$ being a constant. By the similar argument using Markov's inequality and the triangle inequality, we have

$$\text{TPR}_{\text{RMIA}}(\boldsymbol{x}) = \Pr_Z \left( \Pr_{\boldsymbol{z}}(\lambda_{\boldsymbol{x}} \geq \log \gamma) \geq \tau \right) \tag{A117}$$

$$\leq \Pr_Z \left( \Pr_{\boldsymbol{z}}(|\lambda_{\boldsymbol{x}}| \geq \log \gamma) \geq \tau \right) \tag{A118}$$

$$\leq \Pr_Z \left( \frac{\mathbb{E}_{\boldsymbol{z}}[|\lambda_{\boldsymbol{x}}|]}{\log \gamma} \geq \tau \right) \tag{A119}$$

$$= \Pr_Z \left( \mathbb{E}_{\boldsymbol{z}}[|\lambda_{\boldsymbol{x}}|] \geq \tau \log \gamma \right) \tag{A120}$$

$$\leq \Pr_Z \left( \mathbb{E}_{\boldsymbol{z}}[|A|]|Z| + \mathbb{E}_{\boldsymbol{z}}[|B|] + \mathbb{E}_{\boldsymbol{z}}[|q|] \geq \tau \log \gamma \right) \tag{A121}$$

$$= \Pr_Z \left( |Z| \geq \frac{\tau \log \gamma - \mathbb{E}_{\boldsymbol{z}}[|B|]}{\mathbb{E}_{\boldsymbol{z}}[|A|]} - \frac{\mathbb{E}_{\boldsymbol{z}}[|q|]}{\mathbb{E}_{\boldsymbol{z}}[|A|]} \right). \tag{A122}$$

Hence we obtain

$$\text{TPR}_{\text{RMIA}}(x) \leq 1 - F_{|Z|}^{-1} \left( F_{|Z|}^{-1}(1 - \alpha) - \frac{\mathbb{E}_{\boldsymbol{z}}[|q|]}{\mathbb{E}_{\boldsymbol{z}}[|A|]} \right). \tag{A123}$$

$\square$

Note that, unlike per-example LiRA, we must assume that the attacker has access to the underlying distribution for the optimal RMIA as the Equations (A91) and (A92) depend on the parameters computed from shadow models.

### B.3 RMIA power-law

Employing Lemma A4 and the simplified model, we obtain the following upper bound for RMIA performance.

**Theorem 5** (Per-example RMIA power-law). *Fix a target example $(\boldsymbol{x}, y_{\boldsymbol{x}})$. For the simplified model with large $S$ and infinitely many shadow models, the per-example RMIA vulnerability is given as*

$$\text{TPR}_{\text{RMIA}}(\boldsymbol{x}) \leq 1 - F_{|Z|}\left(F_{|Z|}^{-1}(1-\alpha) - \frac{\Psi(\boldsymbol{x}, C)}{\sqrt{S}}\right), \tag{A124}$$

*where $\alpha \geq \text{FPR}_{\text{RMIA}}(\boldsymbol{x})$, $F_{|Z|}$ is the cdf of the standard folded normal distribution, and*

$$\Psi(\boldsymbol{x}, C) = \frac{\mathbb{E}_{\boldsymbol{z}}\left[\frac{\langle \boldsymbol{x}, \boldsymbol{x}-\boldsymbol{z}\rangle^2}{||\boldsymbol{x}||^2} + \frac{\langle \boldsymbol{z}, \boldsymbol{x}-\boldsymbol{z}\rangle^2}{||\boldsymbol{z}||^2} \mid y_{\boldsymbol{z}} = y_{\boldsymbol{x}}\right] + (C-1)\mathbb{E}_{\boldsymbol{z}}\left[\frac{\langle \boldsymbol{x}, \boldsymbol{x}-\boldsymbol{m}_{\boldsymbol{x}}\rangle^2}{||\boldsymbol{x}||^2} + \frac{\langle \boldsymbol{z}, \boldsymbol{z}-\boldsymbol{m}_{\boldsymbol{z}}\rangle^2}{||\boldsymbol{z}||^2} \mid y_{\boldsymbol{z}} \neq y_{\boldsymbol{x}}\right]}{\mathbb{E}_{\boldsymbol{z}}\left[\left|\frac{\langle \boldsymbol{x}, \boldsymbol{x}-\boldsymbol{z}\rangle}{||\boldsymbol{x}||} + \frac{\langle \boldsymbol{z}, \boldsymbol{x}-\boldsymbol{z}\rangle}{||\boldsymbol{z}||}\right| \mid y_{\boldsymbol{z}} = y_{\boldsymbol{x}}\right] + (C-1)\mathbb{E}_{\boldsymbol{z}}\left[\left|\frac{\langle \boldsymbol{x}, \boldsymbol{x}-\boldsymbol{m}_{\boldsymbol{x}}\rangle}{||\boldsymbol{x}||} + \frac{\langle \boldsymbol{z}, \boldsymbol{z}-\boldsymbol{m}_{\boldsymbol{z}}\rangle}{||\boldsymbol{z}||}\right| \mid y_{\boldsymbol{z}} \neq y_{\boldsymbol{x}}\right]}. \tag{A125}$$

*In addition, we have*

$$\log(\text{TPR}_{\text{RMIA}}(\boldsymbol{x}) - \text{FPR}_{\text{RMIA}}(\boldsymbol{x})) \approx -\frac{1}{2}\log S - \frac{1}{2}F_{|Z|}^{-1}(1-\alpha)^2 + \log\frac{\Psi(\boldsymbol{x}, C)}{\sqrt{\pi/2}}. \tag{A126}$$

*Proof.* To apply Lemma A4, we will calculate $\mathbb{E}_{\boldsymbol{z}}[|q|]$ and $\mathbb{E}_{\boldsymbol{z}}[|A|]$. Let $\mathcal{D}_{\text{target}} = \{(\boldsymbol{x}_{j,1}, j), \ldots, (\boldsymbol{x}_{j,S}, j)\}_{j=1}^{C}$. Let $s_{y_a}^{(b)}(\boldsymbol{a})$ denote the score of $(\boldsymbol{a}, y_{\boldsymbol{a}})$ when $\mathcal{D}_{\text{target}}$ contains $(\boldsymbol{b}, y_{\boldsymbol{b}})$ but not the other example. Using similar argument as in the proof of Theorem 2, we have

$$s_{y_{\boldsymbol{x}}}^{(\boldsymbol{x})}(\boldsymbol{x}) = \frac{1}{S}\langle \boldsymbol{x}, \sum_{i=1}^{S} \boldsymbol{x}_{y_{\boldsymbol{x}},i} + \boldsymbol{x} - \boldsymbol{x}_{y_{\boldsymbol{x}},S}\rangle \tag{A127}$$

$$s_{y_{\boldsymbol{x}}}^{(\boldsymbol{z})}(\boldsymbol{x}) = \begin{cases} \frac{1}{S}\langle \boldsymbol{x}, \sum_{i=1}^{S} \boldsymbol{x}_{y_{\boldsymbol{x}},i} + \boldsymbol{z} - \boldsymbol{x}_{y_{\boldsymbol{x}},S}\rangle & \text{if } y_{\boldsymbol{x}} = y_{\boldsymbol{z}} \\ \frac{1}{S}\langle \boldsymbol{x}, \sum_{i=1}^{S} \boldsymbol{x}_{y_{\boldsymbol{x}},i}\rangle & \text{if } y_{\boldsymbol{x}} \neq y_{\boldsymbol{z}} \end{cases} \tag{A128}$$

$$s_{y_{\boldsymbol{z}}}^{(\boldsymbol{x})}(\boldsymbol{z}) = \begin{cases} \frac{1}{S}\langle \boldsymbol{z}, \sum_{i=1}^{S} \boldsymbol{x}_{y_{\boldsymbol{x}},i} + \boldsymbol{x} - \boldsymbol{x}_{y_{\boldsymbol{x}},S}\rangle & \text{if } y_{\boldsymbol{x}} = y_{\boldsymbol{z}} \\ \frac{1}{S}\langle \boldsymbol{z}, \sum_{i=1}^{S} \boldsymbol{x}_{y_{\boldsymbol{z}},i}\rangle & \text{if } y_{\boldsymbol{x}} \neq y_{\boldsymbol{z}} \end{cases} \tag{A129}$$

$$s_{y_{\boldsymbol{z}}}^{(\boldsymbol{z})}(\boldsymbol{z}) = \frac{1}{S}\langle \boldsymbol{z}, \sum_{i=1}^{S} \boldsymbol{x}_{y_{\boldsymbol{z}},i} + \boldsymbol{z} - \boldsymbol{x}_{y_{\boldsymbol{z}},S}\rangle. \tag{A130}$$

Thus we obtain

$$\mu_{\boldsymbol{x},\boldsymbol{x}} = \langle \boldsymbol{x}, \boldsymbol{m}_{y_{\boldsymbol{x}}}\rangle + \frac{1}{S}\langle \boldsymbol{x}, \boldsymbol{x} - \boldsymbol{m}_{y_{\boldsymbol{x}}}\rangle \tag{A131}$$

$$\mu_{\boldsymbol{z},\boldsymbol{x}} = \begin{cases} \langle \boldsymbol{x}, \boldsymbol{m}_{y_{\boldsymbol{x}}}\rangle + \frac{1}{S}\langle \boldsymbol{x}, \boldsymbol{z} - \boldsymbol{m}_{y_{\boldsymbol{x}}}\rangle & \text{if } y_{\boldsymbol{x}} = y_{\boldsymbol{z}} \\ \langle \boldsymbol{x}, \boldsymbol{m}_{y_{\boldsymbol{x}}}\rangle & \text{if } y_{\boldsymbol{x}} \neq y_{\boldsymbol{z}} \end{cases} \tag{A132}$$

$$\mu_{\boldsymbol{x},\boldsymbol{z}} = \begin{cases} \langle \boldsymbol{z}, \boldsymbol{m}_{y_{\boldsymbol{x}}}\rangle + \frac{1}{S}\langle \boldsymbol{z}, \boldsymbol{x} - \boldsymbol{m}_{y_{\boldsymbol{x}}}\rangle & \text{if } y_{\boldsymbol{x}} = y_{\boldsymbol{z}} \\ \langle \boldsymbol{z}, \boldsymbol{m}_{y_{\boldsymbol{z}}}\rangle & \text{if } y_{\boldsymbol{x}} \neq y_{\boldsymbol{z}} \end{cases} \tag{A133}$$

$$\mu_{\boldsymbol{z},\boldsymbol{z}} = \langle \boldsymbol{z}, \boldsymbol{m}_{y_{\boldsymbol{x}}}\rangle + \frac{1}{S}\langle \boldsymbol{z}, \boldsymbol{z} - \boldsymbol{m}_{y_{\boldsymbol{x}}}\rangle \tag{A134}$$

$$\sigma_{\boldsymbol{x}} = \frac{1}{\sqrt{S}}\sqrt{\boldsymbol{x}^T \Sigma \boldsymbol{x}} \tag{A135}$$

$$\sigma_{\boldsymbol{z}} = \frac{1}{\sqrt{S}}\sqrt{\boldsymbol{z}^T \Sigma \boldsymbol{z}}, \tag{A136}$$

where $\boldsymbol{m}_{y_{\boldsymbol{z}}}$ is the true class mean of class $y_{\boldsymbol{z}}$.

Now recall that

$$q = \frac{(\mu_{\boldsymbol{x},\boldsymbol{x}} - \mu_{\boldsymbol{z},\boldsymbol{x}})(\hat{\mu}_{\boldsymbol{x},\boldsymbol{x}} - \hat{\mu}_{\boldsymbol{z},\boldsymbol{x}})}{\hat{\sigma}_{\boldsymbol{x}}^2} + \frac{(\mu_{\boldsymbol{x},\boldsymbol{z}} - \mu_{\boldsymbol{z},\boldsymbol{z}})(\hat{\mu}_{\boldsymbol{x},\boldsymbol{z}} - \hat{\mu}_{\boldsymbol{z},\boldsymbol{z}})}{\hat{\sigma}_{\boldsymbol{z}}^2} \tag{A137}$$

$$A = \frac{\sigma_{\boldsymbol{x}}}{\hat{\sigma}_{\boldsymbol{x}}^2}(\hat{\mu}_{\boldsymbol{x},\boldsymbol{x}} - \hat{\mu}_{\boldsymbol{z},\boldsymbol{x}}) + \frac{\sigma_{\boldsymbol{z}}}{\hat{\sigma}_{\boldsymbol{z}}^2}(\hat{\mu}_{\boldsymbol{x},\boldsymbol{z}} - \hat{\mu}_{\boldsymbol{z},\boldsymbol{z}}). \tag{A138}$$

Since $t_x$ and $t_z$ follow normal distributions, the location and scale parameters of the true distributions correspond to the mean and standard deviations estimated from infinitely many shadow models, respectively. Thus, we have

$$q = \left( \frac{\mu_{x,x} - \mu_{z,x}}{\sigma_x} \right)^2 + \left( \frac{\mu_{x,z} - \mu_{z,z}}{\sigma_z} \right)^2 \tag{A139}$$

$$A = \frac{\mu_{x,x} - \mu_{z,x}}{\sigma_x} + \frac{\mu_{x,z} - \mu_{z,z}}{\sigma_z}. \tag{A140}$$

Using the law of total expectation, we have

$$\mathbb{E}_z[|q|] = \Pr_z(y_z = y_x)\mathbb{E}_z[|q| \mid y_z = y_x] + \sum_{j=1, j \neq y_x}^{C} \Pr_z(y_z = j)\mathbb{E}_z[|q| \mid y_z = j] \tag{A141}$$

$$= \frac{1}{C}\mathbb{E}_z\left[ \left( \frac{\langle x, x - z \rangle}{\sqrt{S}\sqrt{x^T \Sigma x}} \right)^2 + \left( \frac{\langle z, x - z \rangle}{\sqrt{S}\sqrt{z^T \Sigma z}} \right)^2 \,\middle|\, y_z = y_x \right] \tag{A142}$$

$$+ \frac{C-1}{C}\mathbb{E}_z\left[ \left( \frac{\langle x, x - m_{y_x} \rangle}{\sqrt{S}\sqrt{x^T \Sigma x}} \right)^2 + \left( \frac{\langle z, z - m_{y_z} \rangle}{\sqrt{S}\sqrt{z^T \Sigma z}} \right)^2 \,\middle|\, y_z \neq y_x \right] \tag{A143}$$

$$= \frac{1}{CS}\mathbb{E}_z\left[ \frac{\langle x, x - z \rangle^2}{x^T \Sigma x} + \frac{\langle z, x - z \rangle^2}{z^T \Sigma z} \,\middle|\, y_z = y_x \right] \tag{A144}$$

$$+ \frac{C-1}{CS}\mathbb{E}_z\left[ \frac{\langle x, x - m_{y_x} \rangle^2}{x^T \Sigma x} + \frac{\langle z, z - m_{y_z} \rangle^2}{z^T \Sigma z} \,\middle|\, y_z \neq y_x \right], \tag{A145}$$

and

$$\mathbb{E}_z[|A|] = \Pr_z(y_z = y_x)\mathbb{E}_z[|A| \mid y_z = y_x] + \sum_{j=1, j \neq y_x}^{C} \Pr_z(y_z = j)\mathbb{E}_z[|A| \mid y_z = j] \tag{A146}$$

$$= \frac{1}{C}\mathbb{E}_z\left[ \left| \frac{\langle x, x - z \rangle}{\sqrt{S}\sqrt{x^T \Sigma x}} + \frac{\langle z, x - z \rangle}{\sqrt{S}\sqrt{z^T \Sigma z}} \right| \,\middle|\, y_z = y_x \right] \tag{A147}$$

$$+ \frac{C-1}{C}\mathbb{E}_z\left[ \left| \frac{\langle x, x - m_x \rangle}{\sqrt{S}\sqrt{x^T \Sigma x}} + \frac{\langle z, z - m_z \rangle}{\sqrt{S}\sqrt{z^T \Sigma z}} \right| \,\middle|\, y_z \neq y_x \right] \tag{A148}$$

$$= \frac{1}{C\sqrt{S}}\mathbb{E}_z\left[ \left| \frac{\langle x, x - z \rangle}{\sqrt{x^T \Sigma x}} + \frac{\langle z, x - z \rangle}{\sqrt{z^T \Sigma z}} \right| \,\middle|\, y_z = y_x \right] \tag{A149}$$

$$+ \frac{C-1}{C\sqrt{S}}\mathbb{E}_z\left[ \left| \frac{\langle x, x - m_x \rangle}{\sqrt{x^T \Sigma x}} + \frac{\langle z, z - m_z \rangle}{\sqrt{z^T \Sigma z}} \right| \,\middle|\, y_z \neq y_x \right]. \tag{A150}$$

Hence we obtain

$$\frac{\mathbb{E}_z[|q|]}{\mathbb{E}_z[|A|]}$$

$$= \frac{1}{\sqrt{S}} \cdot \frac{\mathbb{E}_z\left[ \frac{\langle x,x-z \rangle^2}{x^T \Sigma x} + \frac{\langle z,x-z \rangle^2}{z^T \Sigma z} \mid y_z = y_x \right] + (C-1)\mathbb{E}_z\left[ \frac{\langle x,x-m_x \rangle^2}{x^T \Sigma x} + \frac{\langle z,z-m_z \rangle^2}{z^T \Sigma z} \mid y_z \neq y_x \right]}{\mathbb{E}_z\left[ \left| \frac{\langle x,x-z \rangle}{\sqrt{x^T \Sigma x}} + \frac{\langle z,x-z \rangle}{\sqrt{z^T \Sigma z}} \right| \mid y_z = y_x \right] + (C-1)\mathbb{E}_z\left[ \left| \frac{\langle x,x-m_x \rangle}{\sqrt{x^T \Sigma x}} + \frac{\langle z,z-m_z \rangle}{\sqrt{z^T \Sigma z}} \right| \mid y_z \neq y_x \right]}. \tag{A151}$$

Now Lemma A4 yields

$$\mathrm{TPR}_{\mathrm{RMIA}}(x) \leq 1 - F_{|Z|}\left( F_{|Z|}^{-1}(1-\alpha) - \frac{\mathbb{E}_z[|q|]}{\mathbb{E}_z[|A|]} \right) \tag{A152}$$

$$= 1 - F_{|Z|}\left( F_{|Z|}^{-1}(1-\alpha) - \frac{\Psi(x, C)}{\sqrt{S}} \right), \tag{A153}$$

where $F_{|Z|}$ is the cdf of the folded normal distribution and

$$\Psi(x, C) = \frac{\mathbb{E}_z\left[ \frac{\langle x,x-z \rangle^2}{x^T \Sigma x} + \frac{\langle z,x-z \rangle^2}{z^T \Sigma z} \mid y_z = y_x \right] + (C-1)\mathbb{E}_z\left[ \frac{\langle x,x-m_x \rangle^2}{x^T \Sigma x} + \frac{\langle z,z-m_z \rangle^2}{z^T \Sigma z} \mid y_z \neq y_x \right]}{\mathbb{E}_z\left[ \left| \frac{\langle x,x-z \rangle}{\sqrt{x^T \Sigma x}} + \frac{\langle z,x-z \rangle}{\sqrt{z^T \Sigma z}} \right| \mid y_z = y_x \right] + (C-1)\mathbb{E}_z\left[ \left| \frac{\langle x,x-m_x \rangle}{\sqrt{x^T \Sigma x}} + \frac{\langle z,z-m_z \rangle}{\sqrt{z^T \Sigma z}} \right| \mid y_z \neq y_x \right]}. \tag{A154}$$

This completes the first half of the theorem.

Now that from Lemma A4 we have

$$\text{TPR}_{\text{RMIA}}(\boldsymbol{x}) = \Pr_Z\left(\Pr_{\boldsymbol{z}}(\lambda_{\boldsymbol{z}} + q \geq \log\gamma) \geq \tau\right) \tag{A155}$$

$$\leq \Pr_Z\left(\frac{\mathbb{E}_{\boldsymbol{z}}[|\lambda_{\boldsymbol{z}}|] + \mathbb{E}_{\boldsymbol{z}}[|q|]}{\log\gamma} \geq \tau\right) \tag{A156}$$

$$= \Pr_Z\left(|Z| \geq F_{|Z|}^{-1}(1-\alpha) - \frac{\mathbb{E}_{\boldsymbol{z}}[|q|]}{\mathbb{E}_{\boldsymbol{z}}[|A|]}\right) \tag{A157}$$

$$\text{FPR}_{\text{RMIA}}(\boldsymbol{x}) = \Pr_Z\left(\Pr_{\boldsymbol{z}}(\lambda_{\boldsymbol{z}} \geq \log\gamma) \geq \tau\right) \tag{A158}$$

$$\leq \Pr_Z\left(\frac{\mathbb{E}_{\boldsymbol{z}}[|\lambda_{\boldsymbol{z}}|]}{\log\gamma} \geq \tau\right) \tag{A159}$$

$$= \Pr_Z\left(|Z| \geq F_{|Z|}^{-1}(1-\alpha)\right). \tag{A160}$$

We claim that the bound for $\text{FPR}_{\text{RMIA}}(\boldsymbol{x})$ (Equation (A159)) is as tight as that for $\text{TPR}_{\text{RMIA}}(\boldsymbol{x})$ (Equation (A156)) for sufficiently large $S$. Let us denote

$$\kappa_{\text{TPR}}(\gamma) = \Pr_{\boldsymbol{z}}(\lambda_{\boldsymbol{z}} + q \geq \log\gamma) \tag{A161}$$

$$\kappa_{\text{FPR}}(\gamma) = \Pr_{\boldsymbol{z}}(\lambda_{\boldsymbol{z}} \geq \log\gamma). \tag{A162}$$

Since $q = O(1/S)$, for large $S$ we approximate

$$\kappa_{\text{TPR}}(\gamma) - \kappa_{\text{FPR}}(\gamma) \approx p_{\lambda_{\boldsymbol{z}}}(\log\gamma)\frac{c_0}{S}, \tag{A163}$$

for some constant $c_0$. Since $\lambda_{\boldsymbol{z}} = O(1/\sqrt{S})$, the scaled random variable $\hat{\lambda}_{\boldsymbol{z}} = \sqrt{S}\lambda_{\boldsymbol{z}}$ is almost independent of $S$. Then by the change of variables formula, we have

$$p_{\lambda_{\boldsymbol{z}}}(\log\gamma)\frac{c_0}{S} = p_{\hat{\lambda}_{\boldsymbol{z}}}(\sqrt{S}\log\gamma)\frac{c_0}{\sqrt{S}}. \tag{A164}$$

Without loss of generality we may set $\log\gamma = 1/\sqrt{S}$. Thus, we have

$$\kappa_{\text{TPR}}(1/\sqrt{S}) - \kappa_{\text{FPR}}(1/\sqrt{S}) \approx p_{\hat{\lambda}_{\boldsymbol{z}}}\left(\sqrt{S} \cdot \frac{1}{\sqrt{S}}\right)\frac{c_0}{\sqrt{S}} = p_{\hat{\lambda}_{\boldsymbol{z}}}(1)\frac{c_0}{\sqrt{S}}. \tag{A165}$$

This quantity scales as $O(1/\sqrt{S})$. Therefore,

$$\text{TPR}_{\text{RMIA}}(\boldsymbol{x}) - \text{FPR}_{\text{RMIA}}(\boldsymbol{x}) = \Pr_t(\kappa_{\text{TPR}}(1/\sqrt{S}) \geq \tau) - \Pr_t(\kappa_{\text{FPR}}(1/\sqrt{S}) \geq \tau) \tag{A166}$$

$$\approx p_{\kappa_{\text{FPR}}(1/\sqrt{S})}(\tau)(\kappa_{\text{FPR}}(1/\sqrt{S}) - \kappa_{\text{TPR}}(1/\sqrt{S})) \tag{A167}$$

$$\approx p_{\kappa_{\text{FPR}}(1/\sqrt{S})}(\tau)p_{\hat{\lambda}_{\boldsymbol{z}}}(1)\frac{c_0}{\sqrt{S}}. \tag{A168}$$

Note that $\kappa_{\text{FPR}}(1/\sqrt{S})$ and, consequently, $\tau$ are independent of $S$ for large enough $S$ since

$$\kappa_{\text{FPR}}(1/\sqrt{S}) = \Pr_{\boldsymbol{z}}\left(\lambda_{\boldsymbol{z}} \geq \frac{1}{\sqrt{S}}\right) = \Pr_{\boldsymbol{z}}\left(\frac{\hat{\lambda}_{\boldsymbol{z}}}{\sqrt{S}} \geq \frac{1}{\sqrt{S}}\right) = \Pr_{\boldsymbol{z}}(\hat{\lambda}_{\boldsymbol{z}} \geq 1). \tag{A169}$$

Hence $\text{TPR}_{\text{RMIA}}(\boldsymbol{x}) - \text{FPR}_{\text{RMIA}}(\boldsymbol{x}) = O(1/\sqrt{S})$. On the other hand, from Equations (A156) and (A159) we have for sufficiently large $S$

$$\Pr_Z\left(\frac{\mathbb{E}_{\boldsymbol{z}}[|\lambda_{\boldsymbol{z}}|]}{\log\gamma} \geq \tau\right) - \Pr_Z\left(\frac{\mathbb{E}_{\boldsymbol{z}}[|\lambda_{\boldsymbol{z}}|]}{\log\gamma} + \frac{\mathbb{E}_{\boldsymbol{z}}[|q|]}{\log\gamma} \geq \tau\right) \tag{A170}$$

$$= \Pr_Z\left(\sqrt{S}\mathbb{E}_{\boldsymbol{z}}[|\lambda_{\boldsymbol{z}}|] \geq \tau\right) - \Pr_Z\left(\sqrt{S}\mathbb{E}_{\boldsymbol{z}}[|\lambda_{\boldsymbol{z}}|] + \frac{c_1}{\sqrt{S}} \geq \tau\right) \tag{A171}$$

$$= \Pr_Z\left(\mathbb{E}_{\boldsymbol{z}}[|\hat{\lambda}_{\boldsymbol{z}}|] \geq \tau\right) - \Pr_Z\left(\mathbb{E}_{\boldsymbol{z}}[|\hat{\lambda}_{\boldsymbol{z}}|] + \frac{c_1}{\sqrt{S}} \geq \tau\right) \tag{A172}$$

$$\approx p_{\mathbb{E}_{\boldsymbol{z}}[|\hat{\lambda}_{\boldsymbol{z}}|]}(\tau)\frac{c_1}{\sqrt{S}}, \tag{A173}$$

where $c_1$ is some constant. Since for large $S$, $\mathbb{E}_{\boldsymbol{z}}[|\hat{\lambda}_{\boldsymbol{z}}|]$ and $\tau$ are independent of $S$, Equation (A173) scales as $O(1/\sqrt{S})$.

Now let

$$\text{TPR}_{\text{LiRA}}(\boldsymbol{x}) = \Pr_{Z}\left(\frac{\mathbb{E}_{\boldsymbol{z}}[|\lambda_{\boldsymbol{z}}|] + \mathbb{E}_{\boldsymbol{z}}[|q|]}{\log\gamma} \geq \tau\right) - v_{\text{TPR}} \tag{A174}$$

$$\text{FPR}_{\text{LiRA}}(\boldsymbol{x}) = \Pr_{Z}\left(\frac{\mathbb{E}_{\boldsymbol{z}}[|\lambda_{\boldsymbol{z}}|]}{\log\gamma} \geq \tau\right) - v_{\text{FPR}} \tag{A175}$$

for some $v_{\text{TPR}}, v_{\text{FPR}} \geq 0$ which evaluate the tightness of the bounds. Then we have

$$v_{\text{TPR}} - v_{\text{FPR}} \tag{A176}$$

$$= \Pr_{Z}\left(\frac{\mathbb{E}_{\boldsymbol{z}}[|\lambda_{\boldsymbol{z}}|] + \mathbb{E}_{\boldsymbol{z}}[|q|]}{\log\gamma} \geq \tau\right) - \Pr_{Z}\left(\frac{\mathbb{E}_{\boldsymbol{z}}[|\lambda_{\boldsymbol{z}}|]}{\log\gamma} \geq \tau\right) - (\text{TPR}_{\text{LiRA}}(\boldsymbol{x}) - \text{FPR}_{\text{LiRA}}(\boldsymbol{x}))$$

$$\tag{A177}$$

$$= O(1/\sqrt{S}). \tag{A178}$$

Hence we conclude that for sufficiently large $S$, the bound for $\text{FPR}_{\text{LiRA}}(\boldsymbol{x})$ is as tight as that for $\text{TPR}_{\text{LiRA}}(\boldsymbol{x})$.

Therefore, noting that $\mathbb{E}_{\boldsymbol{z}}[|q|]/\mathbb{E}_{\boldsymbol{z}}[|A|] = O(1/\sqrt{S})$, for large $S$ we obtain

$$\text{TPR}_{\text{RMIA}}(\boldsymbol{x}) - \text{FPR}_{\text{RMIA}}(\boldsymbol{x}) \tag{A179}$$

$$\approx \Pr_{Z}\left(|Z| \geq F_{|Z|}^{-1}(1-\alpha) - \frac{\mathbb{E}_{\boldsymbol{z}}[|q|]}{\mathbb{E}_{\boldsymbol{z}}[|A|]}\right) - \Pr_{Z}\left(|Z| \geq F_{|Z|}^{-1}(1-\alpha)\right) \tag{A180}$$

$$\approx p_{|Z|}\left(F_{|Z|}^{-1}(1-\alpha)\right)\frac{\mathbb{E}_{\boldsymbol{z}}[|q|]}{\mathbb{E}_{\boldsymbol{z}}[|A|]} \tag{A181}$$

$$= p_{|Z|}\left(F_{|Z|}^{-1}(1-\alpha)\right)\frac{\Psi(\boldsymbol{x}, C)}{\sqrt{S}}. \tag{A182}$$

Since $|Z|$ follows the standard folded normal distribution,

$$p_{|Z|}\left(F_{|Z|}^{-1}(1-\alpha)\right) = \frac{2}{\sqrt{2\pi}}\exp\left(-\frac{1}{2}F_{|Z|}^{-1}(1-\alpha)^2\right). \tag{A183}$$

It follows that

$$\log(\text{TPR}_{\text{RMIA}}(\boldsymbol{x}) - \text{FPR}_{\text{RMIA}}(\boldsymbol{x})) \approx -\frac{1}{2}\log S - \frac{1}{2}F_{|Z|}^{-1}(1-\alpha)^2 + \log\frac{\Psi(\boldsymbol{x}, C)}{\sqrt{\pi/2}}. \tag{A184}$$

$\square$

As for the LiRA power-law, bounding $||\boldsymbol{x} - \boldsymbol{m}_{\boldsymbol{x}}||$ and $||\boldsymbol{z} - \boldsymbol{m}_{\boldsymbol{z}}||$ will provide a worst-case upper bound for which the power-law holds.

Finally, the per-example RMIA power-law is also extended to the average-case:

**Corollary A6** (Average-case RMIA power-law). *For the simplified model with sufficiently large $S$ and infinitely many shadow models, we have*

$$\log(\overline{\text{TPR}}_{\text{RMIA}} - \overline{\text{FPR}}_{\text{RMIA}}) \approx -\frac{1}{2}\log S - \frac{1}{2}F_{|Z|}^{-1}(1-\alpha)^2 + \log\left(\mathop{\mathbb{E}}_{(\boldsymbol{x}, y_{\boldsymbol{x}})\sim\mathbb{D}}\left[\frac{\Psi(\boldsymbol{x}, C)}{\sqrt{\pi/2}}\right]\right), \tag{A185}$$

*where $\alpha \geq \overline{\text{FPR}}_{\text{RMIA}}$ and $F_{|Z|}$ is the cdf of the standard folded normal distribution.*

*Proof.* By Theorem 5 and the law of unconscious statistician, we have for large $S$

$$\overline{\text{TPR}}_{\text{RMIA}} - \overline{\text{FPR}}_{\text{RMIA}} = \int_{\mathscr{D}} p(\boldsymbol{x})(\text{TPR}_{\text{RMIA}}(\boldsymbol{x}) - \text{FPR}_{\text{RMIA}}(\boldsymbol{x}))\mathrm{d}\boldsymbol{x} \tag{A186}$$

$$\approx \int_{\mathscr{D}} p(\boldsymbol{x}) \frac{1}{\sqrt{\pi/2}} e^{-\frac{1}{2}F_{|Z|}^{-1}(1-\alpha)^2} \frac{\Psi(\boldsymbol{x},C)}{\sqrt{S}} \mathrm{d}\boldsymbol{x} \tag{A187}$$

$$= \frac{1}{\sqrt{\pi/2}} e^{-\frac{1}{2}F_{|Z|}^{-1}(1-\alpha)^2} \int_{\mathscr{D}} p(\boldsymbol{x}) \frac{\Psi(\boldsymbol{x},C)}{\sqrt{S}} \mathrm{d}\boldsymbol{x}. \tag{A188}$$

$$= \frac{1}{\sqrt{S}} e^{-\frac{1}{2}F_{|Z|}^{-1}(1-\alpha)^2} \mathop{\mathbb{E}}_{(\boldsymbol{x},y_{\boldsymbol{x}})\sim\mathbb{D}} \left[ \frac{\Psi(\boldsymbol{x},C)}{\sqrt{\pi/2}} \right], \tag{A189}$$

where $p(\boldsymbol{x})$ is the density of $\mathbb{D}$ at $(\boldsymbol{x}, y_{\boldsymbol{x}})$, and $\mathscr{D}$ is the data domain. Hence we obtain

$$\log(\overline{\text{TPR}}_{\text{RMIA}} - \overline{\text{FPR}}_{\text{RMIA}}) \approx -\frac{1}{2}\log S - \frac{1}{2}F_{|Z|}^{-1}(1-\alpha)^2 + \log\left( \mathop{\mathbb{E}}_{(\boldsymbol{x},y_{\boldsymbol{x}})\sim\mathbb{D}} \left[ \frac{\Psi(\boldsymbol{x},C)}{\sqrt{\pi/2}} \right] \right). \tag{A190}$$

$\square$

# C   Training details

## C.1   Parameterization

We utilise pre-trained feature extractors BiT-M-R50x1 (R-50) (Kolesnikov et al., 2020) with 23.5M parameters and Vision Transformer ViT-Base-16 (ViT-B) (Dosovitskiy et al., 2021) with 85.8M parameters, both pretrained on the ImageNet-21K dataset (Russakovsky et al., 2015). We download the feature extractor checkpoints from the respective repositories.

Following Tobaben et al. (2023) that show the favorable trade-off of parameter-efficient fine-tuning between computational cost, utility and privacy even for small datasets, we only consider fine-tuning subsets of all feature extractor parameters. We consider the following configurations:

- **Head:** We train a linear layer on top of the pre-trained feature extractor.
- **FiLM:** In addition to the linear layer from Head, we fine-tune parameter-efficient FiLM (Perez et al., 2018) adapters scattered throughout the network. While a diverse set of adapters has been proposed, we utilise FiLM as it has been shown to be competitive in prior work (Shysheya et al., 2023; Tobaben et al., 2023).

### C.1.1   Licenses and access

The licenses and means to access the model checkpoints can be found below.

- BiT-M-R50x1 (R-50) (Kolesnikov et al., 2020) is licensed with the Apache-2.0 license and can be obtained through the instructions on `https://github.com/google-research/big_transfer`.
- Vision Transformer ViT-Base-16 (ViT-B) (Dosovitskiy et al., 2021) is licensed with the Apache-2.0 license and can be obtained through the instructions on `https://github.com/google-research/vision_transformer`.

## C.2   Hyperparameter tuning

Our hyperparameter tuning is heavily inspired by the comprehensive few-shot experiments by Tobaben et al. (2023). We utilise their hyperparameter tuning protocol as it has been proven to yield SOTA results for (DP) few-shot models. Given the input $\mathcal{D}$ dataset we perform hyperparameter tuning by splitting the $\mathcal{D}$ into 70% train and 30% validation. We then perform the specified iterations of hyperparameter tuning using the tree-structured Parzen estimator (Bergstra et al., 2011) strategy as implemented in Optuna (Akiba et al., 2019) to derive a set of hyperparameters that yield the highest accuracy on the validation split. This set of hyperparameters is subsequently used to train all shadow models with the Adam optimizer (Kingma and Ba, 2015). Details on the set of hyperparameters that are tuned and their ranges can be found in Table A1.

Table A1: Hyperparameter ranges used for the Bayesian optimization with Optuna.

|  | lower bound | upper bound |
|---|---|---|
| batch size | 10 | $|\mathcal{D}|$ |
| clipping norm | 0.2 | 10 |
| epochs | 1 | 200 |
| learning rate | 1e-7 | 1e-2 |

## C.3   Datasets

Table A2 shows the datasets used in the paper. We base our experiments on a subset of the the few-shot benchmark VTAB (Zhai et al., 2019) that achieves a classification accuracy $> 80\%$ and thus would considered to be used by a practitioner. Additionally, we add CIFAR10 which is not part of the original VTAB benchmark.

Table A2: Used datasets in the paper, their minimum and maximum shots $S$ and maximum number of classes $C$ and their test accuracy when fine-tuning a non-DP ViT-B Head. The test accuracy for EuroSAT and Resics45 is computed on the part of the training split that is not used for training the particular model due to both datasets missing an official test split. Note that LiRA requires $2S$ for training the shadow models and thus $S$ is smaller than when only performing fine-tuning.

| dataset | (max.) $C$ | min. $S$ | max. $S$ | test accuracy (min. $S$) | test accuracy (max. $S$) |
|---|---|---|---|---|---|
| Patch Camelyon (Veeling et al., 2018) | 2 | 256 | 65536 | 82.8% | 85.6% |
| CIFAR10 (Krizhevsky, 2009) | 10 | 8 | 2048 | 92.7% | 97.7% |
| EuroSAT (Helber et al., 2019) | 10 | 8 | 512 | 80.2% | 96.7% |
| Pets (Parkhi et al., 2012) | 37 | 8 | 32 | 82.3% | 90.7% |
| Resisc45 (Cheng et al., 2017) | 45 | 32 | 256 | 83.5% | 91.6% |
| CIFAR100 (Krizhevsky, 2009) | 100 | 16 | 128 | 82.2% | 87.6% |

### C.3.1 Licenses and access

The licenses and means to access the datasets can be found below. We downloaded all datasets from TensorFlow datasets `https://www.tensorflow.org/datasets` but Resics45 which required manual download.

- Patch Camelyon (Veeling et al., 2018) is licensed with Creative Commons Zero v1.0 Universal (cc0-1.0) and we use version 2.0.0 of the dataset as specified on `https://www.tensorflow.org/datasets/catalog/patch_camelyon`.
- CIFAR10 (Krizhevsky, 2009) is licensed with an unknown license and we use version 3.0.2 of the dataset as specified on `https://www.tensorflow.org/datasets/catalog/cifar10`.
- EuroSAT (Helber et al., 2019) is licensed with MIT and we use version 2.0.0 of the dataset as specified on `https://www.tensorflow.org/datasets/catalog/eurosat`.
- Pets (Parkhi et al., 2012) is licensed with CC BY-SA 4.0 Deed and we use version 3.2.0 of the dataset as specified on `https://www.tensorflow.org/datasets/catalog/oxford_iiit_pet`.
- Resisc45 (Cheng et al., 2017) is licensed with an unknown license and we use version 3.0.0 of the dataset as specified on `https://www.tensorflow.org/datasets/catalog/resisc45`.
- CIFAR100 (Krizhevsky, 2009) is licensed with an unknown license and we use version 3.0.2 of the dataset as specified on `https://www.tensorflow.org/datasets/catalog/cifar100`.

### C.4 Compute resources

All experiments but the R-50 (FiLM) experiments are run on CPU with 8 cores and 16 GB of host memory. The training time depends on the model (ViT is cheaper than R-50), number of shots $S$ and the number of classes $C$ but ranges for the training of one model from some minutes to an hour. This assumes that the images are passed once through the pre-trained backbone and then cached as feature vectors. The provided code implements this optimization.

The R-50 (FiLM) experiments are significantly more expensive and utilise a NVIDIA V100 with 40 GB VRAM, 10 CPU cores and 64 GB of host memory. The training of 257 shadow models then does not exceed 24h for the settings that we consider.

We estimate that in total we spend around 7 days of V100 and some dozens of weeks of CPU core time but more exact measurements are hard to make.

# D  Additional results

In this section, we provide tabular results for our experiments and additional figures that did not fit into the main paper.

## D.1  Additional results for Section 4

This Section contains additional results for Section 4.

### D.1.1  Vulnerability as a function of shots

This section displays additional results to Figure 1 for FPR $\in \{0.1, 0.01, 0.001\}$ for ViT-B and R-50 in in Figure A.1 and Tables A3 and A4.

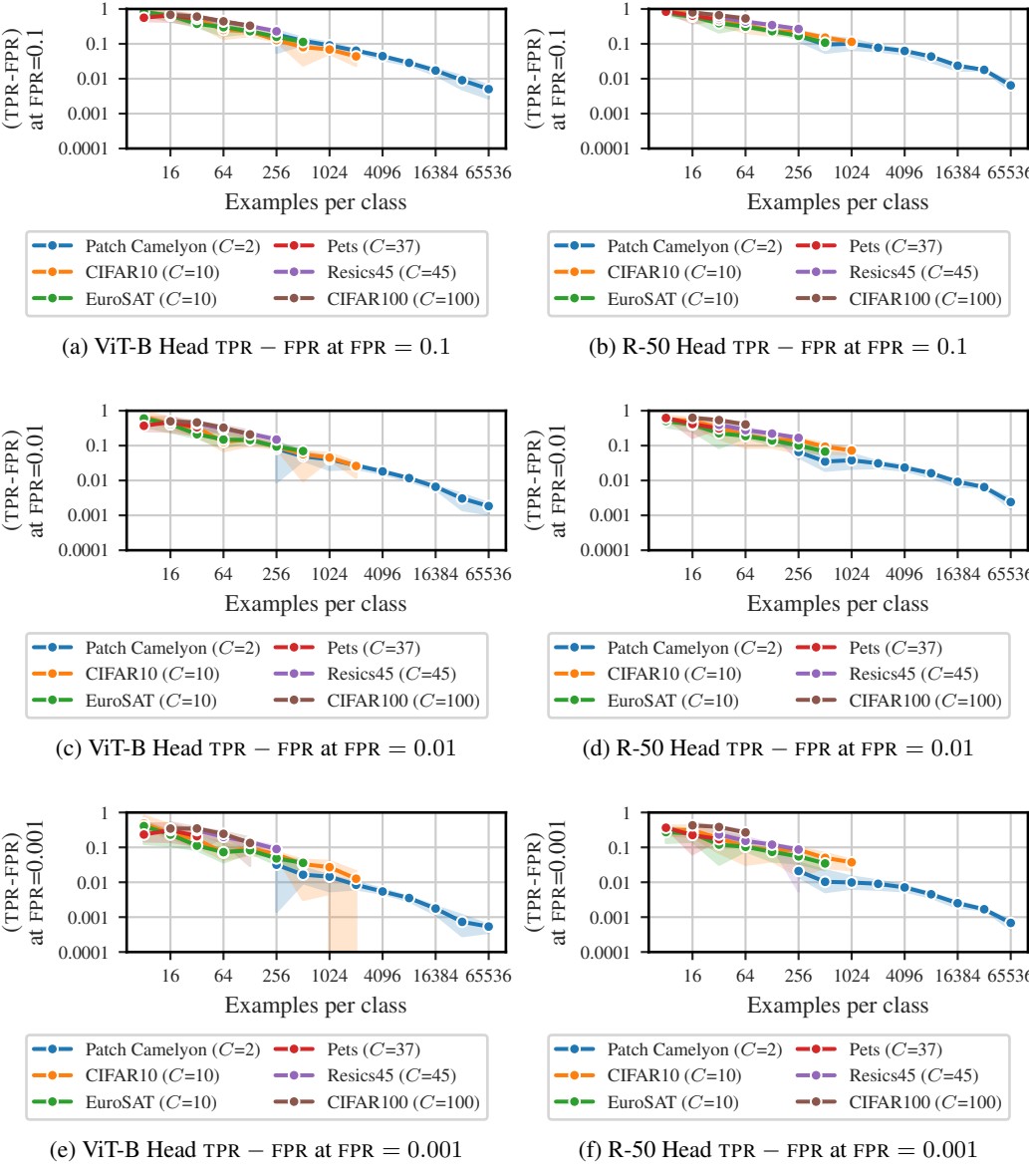

Figure A.1: MIA vulnerability as a function of shots (examples per class) when attacking a pre-trained ViT-B and R-50 Head trained without DP on different downstream datasets. The errorbars display the minimum and maximum Clopper-Pearson CIs over six seeds and the solid line the median.

Table A3: Median MIA vulnerability over six seeds as a function of $S$ (shots) when attacking a Head trained without DP on-top of a ViT-B. The ViT-B is pre-trained on ImageNet-21k.

| dataset | classes ($C$) | shots ($S$) | tpr@fpr=0.1 | tpr@fpr=0.01 | tpr@fpr=0.001 |
|---|---|---|---|---|---|
| Patch Camelyon (Veeling et al., 2018) | 2 | 256 | 0.266 | 0.086 | 0.032 |
| | | 512 | 0.223 | 0.059 | 0.018 |
| | | 1024 | 0.191 | 0.050 | 0.015 |
| | | 2048 | 0.164 | 0.037 | 0.009 |
| | | 4096 | 0.144 | 0.028 | 0.007 |
| | | 8192 | 0.128 | 0.021 | 0.005 |
| | | 16384 | 0.118 | 0.017 | 0.003 |
| | | 32768 | 0.109 | 0.014 | 0.002 |
| | | 65536 | 0.105 | 0.012 | 0.002 |
| CIFAR10 (Krizhevsky, 2009) | 10 | 8 | 0.910 | 0.660 | 0.460 |
| | | 16 | 0.717 | 0.367 | 0.201 |
| | | 32 | 0.619 | 0.306 | 0.137 |
| | | 64 | 0.345 | 0.132 | 0.067 |
| | | 128 | 0.322 | 0.151 | 0.082 |
| | | 256 | 0.227 | 0.096 | 0.054 |
| | | 512 | 0.190 | 0.068 | 0.032 |
| | | 1024 | 0.168 | 0.056 | 0.025 |
| | | 2048 | 0.148 | 0.039 | 0.013 |
| EuroSAT (Helber et al., 2019) | 10 | 8 | 0.921 | 0.609 | 0.408 |
| | | 16 | 0.738 | 0.420 | 0.234 |
| | | 32 | 0.475 | 0.222 | 0.113 |
| | | 64 | 0.400 | 0.159 | 0.074 |
| | | 128 | 0.331 | 0.155 | 0.084 |
| | | 256 | 0.259 | 0.104 | 0.049 |
| | | 512 | 0.213 | 0.080 | 0.037 |
| Pets (Parkhi et al., 2012) | 37 | 8 | 0.648 | 0.343 | 0.160 |
| | | 16 | 0.745 | 0.439 | 0.259 |
| | | 32 | 0.599 | 0.311 | 0.150 |
| Resics45 (Cheng et al., 2017) | 45 | 32 | 0.672 | 0.425 | 0.267 |
| | | 64 | 0.531 | 0.295 | 0.168 |
| | | 128 | 0.419 | 0.212 | 0.115 |
| | | 256 | 0.323 | 0.146 | 0.072 |
| CIFAR100 (Krizhevsky, 2009) | 100 | 16 | 0.814 | 0.508 | 0.324 |
| | | 32 | 0.683 | 0.445 | 0.290 |
| | | 64 | 0.538 | 0.302 | 0.193 |
| | | 128 | 0.433 | 0.208 | 0.114 |

Table A4: Median MIA vulnerability over six seeds as a function of $S$ (shots) when attacking a Head trained without DP on-top of a R-50. The R-50 is pre-trained on ImageNet-21k.

| dataset | classes ($C$) | shots ($S$) | tpr@fpr=0.1 | tpr@fpr=0.01 | tpr@fpr=0.001 |
|---|---|---|---|---|---|
| Patch Camelyon (Veeling et al., 2018) | 2 | 256 | 0.272 | 0.076 | 0.022 |
| | | 512 | 0.195 | 0.045 | 0.011 |
| | | 1024 | 0.201 | 0.048 | 0.011 |
| | | 2048 | 0.178 | 0.041 | 0.010 |
| | | 4096 | 0.163 | 0.033 | 0.008 |
| | | 8192 | 0.143 | 0.026 | 0.006 |
| | | 16384 | 0.124 | 0.019 | 0.004 |
| | | 32768 | 0.118 | 0.016 | 0.003 |
| | | 65536 | 0.106 | 0.012 | 0.002 |
| CIFAR10 (Krizhevsky, 2009) | 10 | 8 | 0.911 | 0.574 | 0.324 |
| | | 16 | 0.844 | 0.526 | 0.312 |
| | | 32 | 0.617 | 0.334 | 0.183 |
| | | 64 | 0.444 | 0.208 | 0.106 |
| | | 128 | 0.334 | 0.159 | 0.084 |
| | | 256 | 0.313 | 0.154 | 0.086 |
| | | 512 | 0.251 | 0.103 | 0.051 |
| | | 1024 | 0.214 | 0.082 | 0.038 |
| EuroSAT (Helber et al., 2019) | 10 | 8 | 0.846 | 0.517 | 0.275 |
| | | 16 | 0.699 | 0.408 | 0.250 |
| | | 32 | 0.490 | 0.236 | 0.121 |
| | | 64 | 0.410 | 0.198 | 0.105 |
| | | 128 | 0.332 | 0.151 | 0.075 |
| | | 256 | 0.269 | 0.111 | 0.056 |
| | | 512 | 0.208 | 0.077 | 0.036 |
| Pets (Parkhi et al., 2012) | 37 | 8 | 0.937 | 0.631 | 0.366 |
| | | 16 | 0.745 | 0.427 | 0.227 |
| | | 32 | 0.588 | 0.321 | 0.173 |
| Resics45 (Cheng et al., 2017) | 45 | 32 | 0.671 | 0.405 | 0.235 |
| | | 64 | 0.534 | 0.289 | 0.155 |
| | | 128 | 0.445 | 0.231 | 0.121 |
| | | 256 | 0.367 | 0.177 | 0.088 |
| CIFAR100 (Krizhevsky, 2009) | 100 | 16 | 0.897 | 0.638 | 0.429 |
| | | 32 | 0.763 | 0.549 | 0.384 |
| | | 64 | 0.634 | 0.414 | 0.269 |

### D.1.2 Vulnerability as a function of the number of classes

This section displays additional results to Figure 2 for FPR $\in \{0.1, 0.01, 0.001\}$ for ViT-B and R-50 in in Figure A.2 and Tables A5 and A6.

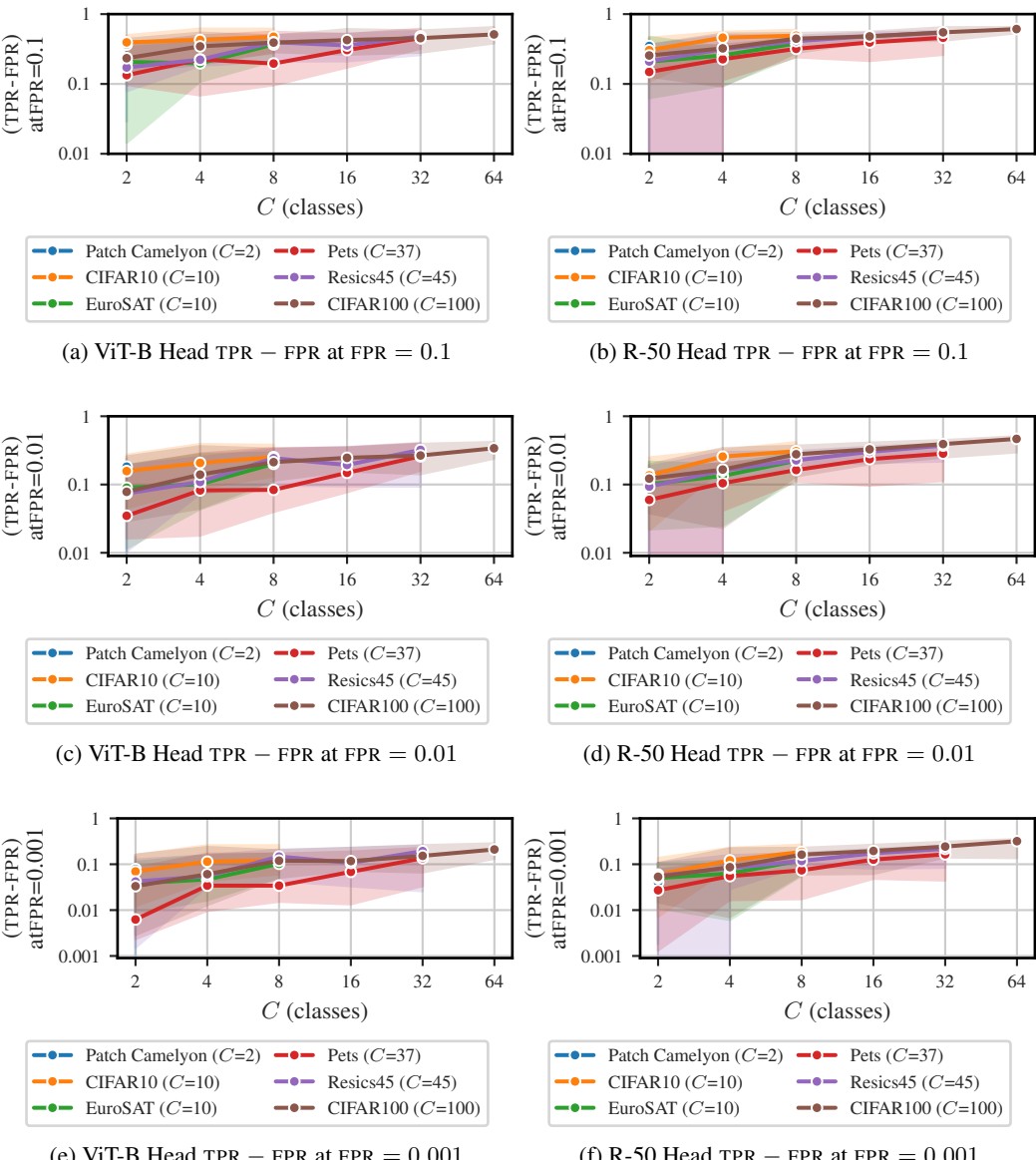

Figure A.2: MIA vulnerability as a function of $C$ (classes) when attacking a ViT-B and R-50 Head fine-tuned without DP on different datasets where the classes are randomly sub-sampled and $S = 32$. The solid line displays the median and the errorbars the min and max clopper-pearson CIs over 12 seeds.

Table A5: Median MIA vulnerability over 12 seeds as a function of $C$ (classes) when attacking a Head trained without DP on-top of a ViT-B. The Vit-B is pre-trained on ImageNet-21k.

| dataset | shots ($S$) | classes ($C$) | tpr@fpr=0.1 | tpr@fpr=0.01 | tpr@fpr=0.001 |
|---|---|---|---|---|---|
| Patch Camelyon (Veeling et al., 2018) | 32 | 2 | 0.467 | 0.192 | 0.080 |
| CIFAR10 (Krizhevsky, 2009) | 32 | 2 | 0.494 | 0.167 | 0.071 |
| | | 4 | 0.527 | 0.217 | 0.115 |
| | | 8 | 0.574 | 0.262 | 0.123 |
| EuroSAT (Helber et al., 2019) | 32 | 2 | 0.306 | 0.100 | 0.039 |
| | | 4 | 0.298 | 0.111 | 0.047 |
| | | 8 | 0.468 | 0.211 | 0.103 |
| Pets (Parkhi et al., 2012) | 32 | 2 | 0.232 | 0.045 | 0.007 |
| | | 4 | 0.324 | 0.092 | 0.035 |
| | | 8 | 0.296 | 0.094 | 0.035 |
| | | 16 | 0.406 | 0.158 | 0.069 |
| | | 32 | 0.553 | 0.269 | 0.136 |
| Resics45 (Cheng et al., 2017) | 32 | 2 | 0.272 | 0.084 | 0.043 |
| | | 4 | 0.322 | 0.119 | 0.056 |
| | | 8 | 0.496 | 0.253 | 0.148 |
| | | 16 | 0.456 | 0.204 | 0.108 |
| | | 32 | 0.580 | 0.332 | 0.195 |
| CIFAR100 (Krizhevsky, 2009) | 32 | 2 | 0.334 | 0.088 | 0.035 |
| | | 4 | 0.445 | 0.150 | 0.061 |
| | | 8 | 0.491 | 0.223 | 0.121 |
| | | 16 | 0.525 | 0.256 | 0.118 |
| | | 32 | 0.553 | 0.276 | 0.153 |
| | | 64 | 0.612 | 0.350 | 0.211 |

Table A6: Median MIA vulnerability over 12 seeds as a function of $C$ (classes) when attacking a Head trained without DP on-top of a R-50. The R-50 is pre-trained on ImageNet-21k.

| dataset | shots ($S$) | classes ($C$) | tpr@fpr=0.1 | tpr@fpr=0.01 | tpr@fpr=0.001 |
|---|---|---|---|---|---|
| Patch Camelyon (Veeling et al., 2018) | 32 | 2 | 0.452 | 0.151 | 0.041 |
| CIFAR10 (Krizhevsky, 2009) | 32 | 2 | 0.404 | 0.146 | 0.060 |
| | | 4 | 0.560 | 0.266 | 0.123 |
| | | 8 | 0.591 | 0.318 | 0.187 |
| EuroSAT (Helber et al., 2019) | 32 | 2 | 0.309 | 0.111 | 0.050 |
| | | 4 | 0.356 | 0.144 | 0.064 |
| | | 8 | 0.480 | 0.233 | 0.123 |
| Pets (Parkhi et al., 2012) | 32 | 2 | 0.249 | 0.068 | 0.029 |
| | | 4 | 0.326 | 0.115 | 0.056 |
| | | 8 | 0.419 | 0.173 | 0.075 |
| | | 16 | 0.493 | 0.245 | 0.127 |
| | | 32 | 0.559 | 0.294 | 0.166 |
| Resics45 (Cheng et al., 2017) | 32 | 2 | 0.310 | 0.103 | 0.059 |
| | | 4 | 0.415 | 0.170 | 0.083 |
| | | 8 | 0.510 | 0.236 | 0.119 |
| | | 16 | 0.585 | 0.311 | 0.174 |
| | | 32 | 0.644 | 0.382 | 0.218 |
| CIFAR100 (Krizhevsky, 2009) | 32 | 2 | 0.356 | 0.132 | 0.054 |
| | | 4 | 0.423 | 0.176 | 0.087 |
| | | 8 | 0.545 | 0.288 | 0.163 |
| | | 16 | 0.580 | 0.338 | 0.196 |
| | | 32 | 0.648 | 0.402 | 0.244 |
| | | 64 | 0.711 | 0.476 | 0.320 |

### D.1.3 Data for FiLM and from scratch training

Table A7: MIA vulnerability data used in Figure 5b. Note that the data from Carlini et al. (2022) is only partially tabular, thus we estimated the TPR at FPR from the plots in the Appendix of their paper.

| model | dataset | classes (C) | shots (S) | source | tpr@ fpr=0.1 | tpr@ fpr=0.01 | tpr@ fpr=0.001 |
|---|---|---|---|---|---|---|---|
| R-50 FiLM | CIFAR10 (Krizhevsky, 2009) | 10 | 50 | This work | 0.482 | 0.275 | 0.165 |
| | CIFAR100 (Krizhevsky, 2009) | 100 | 10 | Tobaben et al. (2023) | 0.933 | 0.788 | 0.525 |
| | | | 25 | Tobaben et al. (2023) | 0.766 | 0.576 | 0.449 |
| | | | 50 | Tobaben et al. (2023) | 0.586 | 0.388 | 0.227 |
| | | | 100 | Tobaben et al. (2023) | 0.448 | 0.202 | 0.077 |
| | EuroSAT (Helber et al., 2019) | 10 | 8 | This work | 0.791 | 0.388 | 0.144 |
| | Patch Camelyon (Veeling et al., 2018) | 2 | 256 | This work | 0.379 | 0.164 | 0.076 |
| | Pets (Parkhi et al., 2012) | 37 | 8 | This work | 0.956 | 0.665 | 0.378 |
| | Resics45 (Cheng et al., 2017) | 45 | 32 | This work | 0.632 | 0.379 | 0.217 |
| from scratch | CIFAR10 (Krizhevsky, 2009) | 10 | 2500 | Carlini et al. (2022) | 0.300 | 0.110 | 0.084 |
| (wide ResNet) | CIFAR100 (Krizhevsky, 2009) | 100 | 250 | Carlini et al. (2022) | 0.700 | 0.400 | 0.276 |

### D.1.4 Predicting dataset vulnerability as function of $S$ and $C$

This section provides additional results for the model based on Equation (13)

Table A8: Results for fitting Equation (13) with statsmodels Seabold and Perktold (2010) to ViT Head data at FPR $\in \{0.1, 0.01, 0.001\}$. We utilize an ordinary least squares. The test $R^2$ assesses the fit to the data of R-50 Head.

| coeff. | FPR | $R^2$ | test $R^2$ | coeff. value | std. error | $t$ | $p > |z|$ | coeff. [0.025 | coeff. 0.975] |
|---|---|---|---|---|---|---|---|---|---|
| $\beta_S$ (for $S$) | 0.1 | 0.952 | 0.907 | -0.506 | 0.011 | -44.936 | 0.000 | -0.529 | -0.484 |
| | 0.01 | 0.946 | 0.854 | -0.555 | 0.014 | -39.788 | 0.000 | -0.582 | -0.527 |
| | 0.001 | 0.930 | 0.790 | -0.627 | 0.019 | -32.722 | 0.000 | -0.664 | -0.589 |
| $\beta_C$ (for $C$) | 0.1 | 0.952 | 0.907 | 0.090 | 0.021 | 4.231 | 0.000 | 0.048 | 0.131 |
| | 0.01 | 0.946 | 0.854 | 0.182 | 0.026 | 6.960 | 0.000 | 0.131 | 0.234 |
| | 0.001 | 0.930 | 0.790 | 0.300 | 0.036 | 8.335 | 0.000 | 0.229 | 0.371 |
| $\beta_0$ (intercept) | 0.1 | 0.952 | 0.907 | 0.314 | 0.045 | 6.953 | 0.000 | 0.225 | 0.402 |
| | 0.01 | 0.946 | 0.854 | 0.083 | 0.056 | 1.491 | 0.137 | -0.027 | 0.193 |
| | 0.001 | 0.930 | 0.790 | -0.173 | 0.077 | -2.261 | 0.025 | -0.324 | -0.022 |

Figure A.3 shows the performance for all considered FPR.

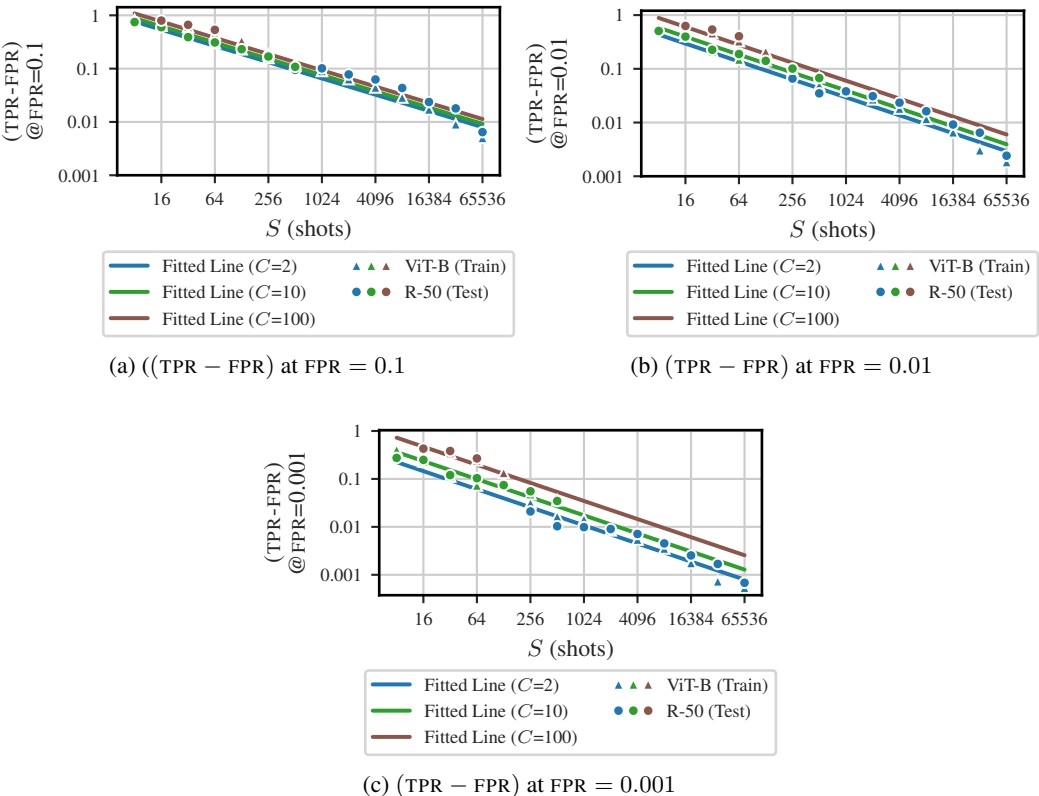

(a) $((\mathrm{TPR} - \mathrm{FPR})$ at FPR $= 0.1$

(b) $(\mathrm{TPR} - \mathrm{FPR})$ at FPR $= 0.01$

(c) $(\mathrm{TPR} - \mathrm{FPR})$ at FPR $= 0.001$

Figure A.3: Predicted MIA vulnerability as a function of $S$ (shots) using a model based on Equation (13) fitted Table A3 (ViT-B). The triangles show the median $\mathrm{TPR} - \mathrm{FPR}$ for the train set (ViT-B; Table A3) and circle the test set (R-50; Table A4) over six seeds. Note that the triangles and dots for $C = 10$ are for EuroSAT.

## D.2 Simpler variant of the prediction model

The prediction model in the main text (Equation (13)) avoids predicting TPR < FPR in the tail when $S$ is very large. In this section, we analyze a variation of the regression model that is simpler and predicts $\log_{10}(\text{TPR})$ instead of $\log_{10}(\text{TPR} - \text{FPR})$. This variation fits worse to the empirical data and will predict TPR < FPR for high $S$.

The general form this variant can be found in Equation (A191), where $\beta_S$, $\beta_C$ and $\beta_0$ are the learnable regression parameters.

$$\log_{10}(\text{TPR}) = \beta_S \log_{10}(S) + \beta_C \log_{10}(C) + \beta_0 \qquad (A191)$$

Table A9 provides tabular results on the performance of the variant.

Table A9: Results for fitting Equation (A191) with statsmodels Seabold and Perktold (2010) to ViT Head data at FPR $\in \{0.1, 0.01, 0.001\}$. We utilize an ordinary least squares. The test $R^2$ assesses the fit to the data of R-50 Head.

| coeff. | FPR | $R^2$ | test $R^2$ | coeff. value | std. error | $t$ | $p > |z|$ | coeff. [0.025 | coeff. 0.975] |
|---|---|---|---|---|---|---|---|---|---|
| $\beta_S$ (for $S$) | 0.1 | 0.908 | 0.764 | -0.248 | 0.008 | -30.976 | 0.000 | -0.264 | -0.233 |
| | 0.01 | 0.940 | 0.761 | -0.416 | 0.011 | -36.706 | 0.000 | -0.438 | -0.393 |
| | 0.001 | 0.931 | 0.782 | -0.553 | 0.017 | -32.507 | 0.000 | -0.586 | -0.519 |
| $\beta_C$ (for $C$) | 0.1 | 0.908 | 0.764 | 0.060 | 0.015 | 3.955 | 0.000 | 0.030 | 0.089 |
| | 0.01 | 0.940 | 0.761 | 0.169 | 0.021 | 7.941 | 0.000 | 0.127 | 0.211 |
| | 0.001 | 0.931 | 0.782 | 0.297 | 0.032 | 9.303 | 0.000 | 0.234 | 0.360 |
| $\beta_0$ (intercept) | 0.1 | 0.908 | 0.764 | 0.029 | 0.032 | 0.913 | 0.362 | -0.034 | 0.093 |
| | 0.01 | 0.940 | 0.761 | -0.118 | 0.045 | -2.613 | 0.010 | -0.208 | -0.029 |
| | 0.001 | 0.931 | 0.782 | -0.295 | 0.068 | -4.345 | 0.000 | -0.429 | -0.161 |

Figure A.4 plots the performance of the variant similar to Figure 5a in the main text.

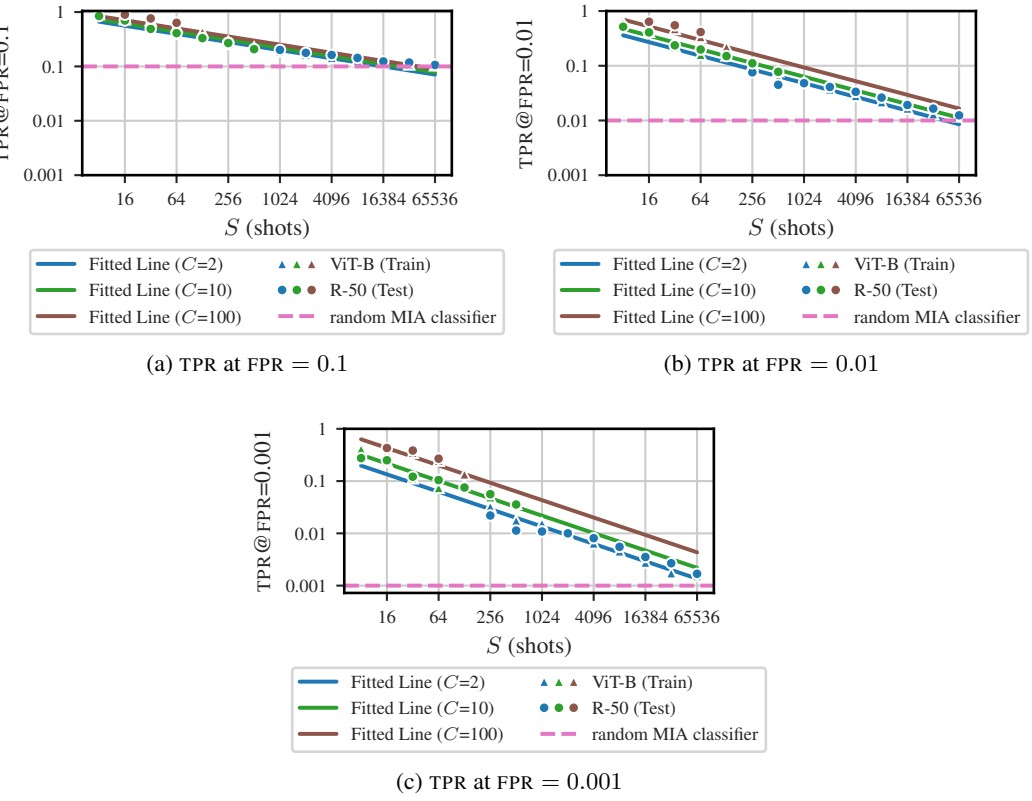

(a) TPR at FPR $= 0.1$

(b) TPR at FPR $= 0.01$

(c) TPR at FPR $= 0.001$

Figure A.4: Predicted MIA vulnerability as a function of $S$ (shots) using a model based on Equation (A191) fitted Table A3 (ViT-B). The triangles show the median TPR for the train set (ViT-B; Table A3) and circle the test set (R-50; Table A4) over six seeds. Note that the triangles and dots for $C = 10$ are for EuroSAT.

## D.3 Empirical results for RMIA

Figures A.5 to A.7 report additional results for RMIA Zarifzadeh et al. (2024).

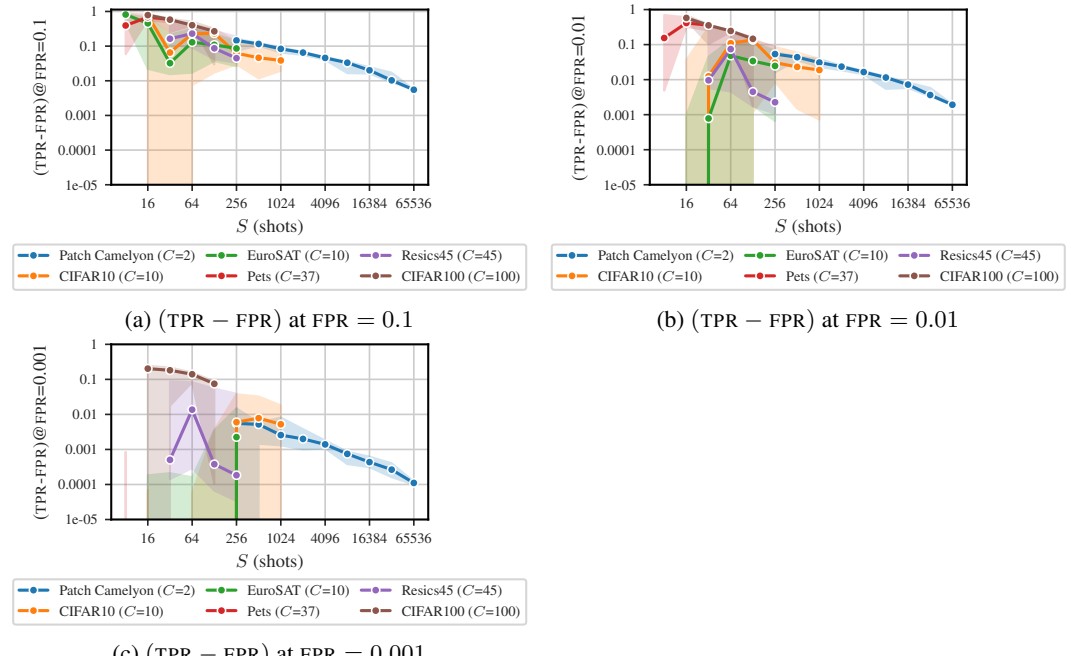

(a) $(\mathrm{TPR} - \mathrm{FPR})$ at $\mathrm{FPR} = 0.1$

(b) $(\mathrm{TPR} - \mathrm{FPR})$ at $\mathrm{FPR} = 0.01$

(c) $(\mathrm{TPR} - \mathrm{FPR})$ at $\mathrm{FPR} = 0.001$

Figure A.5: RMIA (Zarifzadeh et al., 2024) vulnerability ($\mathrm{TPR} - \mathrm{FPR}$ at fixed $\mathrm{FPR}$) as a function of $S$ (shots) when attacking a ViT-B Head fine-tuned without DP on different datasets. We observe at power-law relationship but especially at low $\mathrm{FPR}$ the relationship is not as clear as with LiRA (compare to Figure A.1). The solid line displays the median and the error bars the minimum of the lower bounds and maximum of the upper bounds for the Clopper-Pearson CIs over six seeds.

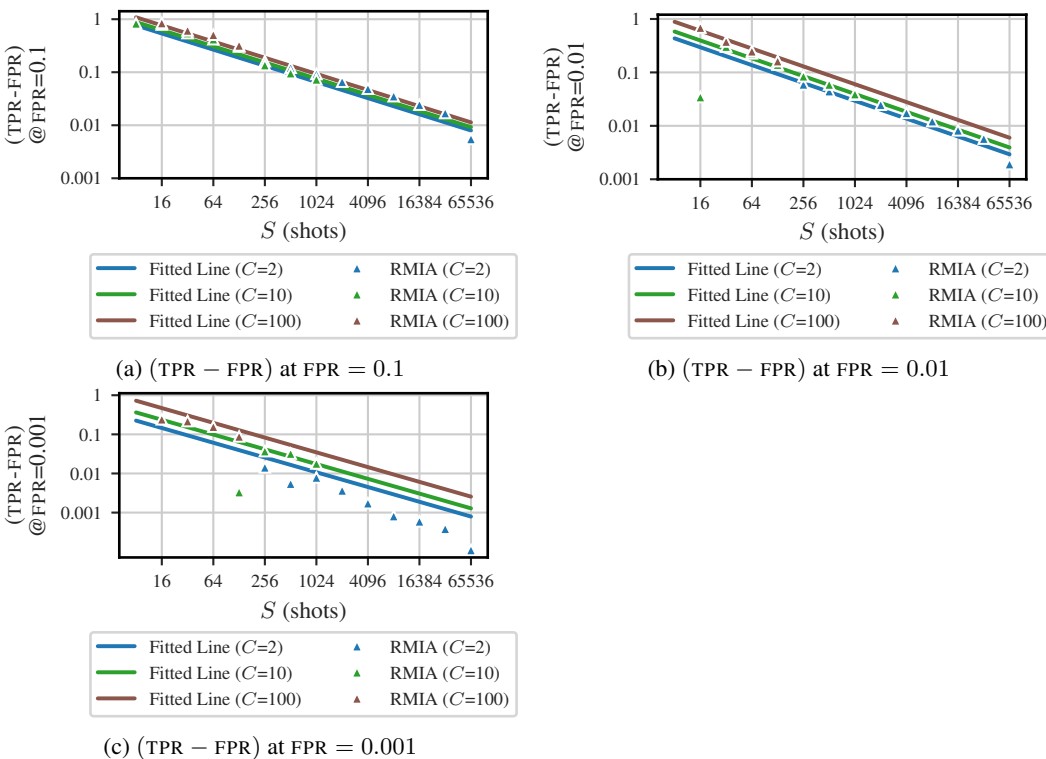

(a) $(\text{TPR} - \text{FPR})$ at $\text{FPR} = 0.1$

(b) $(\text{TPR} - \text{FPR})$ at $\text{FPR} = 0.01$

(c) $(\text{TPR} - \text{FPR})$ at $\text{FPR} = 0.001$

Figure A.6: Predicted MIA vulnerability $((\text{TPR} - \text{FPR})$ at $\text{FPR})$ based on LiRA vulnerability data as a function of $S$ (shots) in comparison to observed RMIA (Zarifzadeh et al., 2024) vulnerability on the same settings. The triangles show the highest TPR when attacking (ViT-B Head) with RMIA over six seeds (datasets: Patch Camelyon, EuroSAT and CIFAR100). Especially at $\text{FPR} = 0.1$ the relationship behaves very similar for both MIAs, but RMIA shows more noisy behavior at lower FPR.

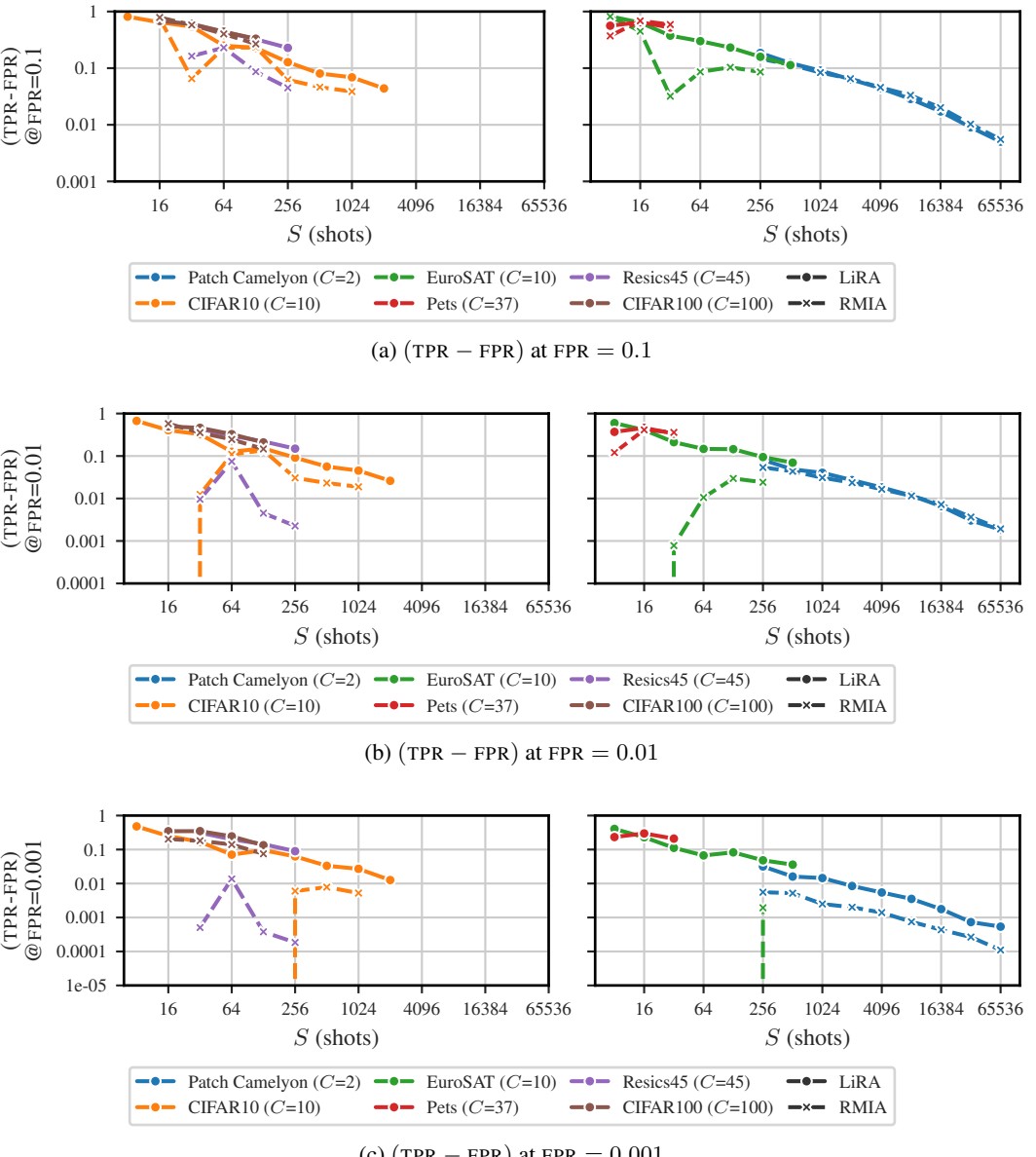

(a) $\left(\mathrm{TPR} - \mathrm{FPR}\right)$ at $\mathrm{FPR} = 0.1$

(b) $\left(\mathrm{TPR} - \mathrm{FPR}\right)$ at $\mathrm{FPR} = 0.01$

(c) $\left(\mathrm{TPR} - \mathrm{FPR}\right)$ at $\mathrm{FPR} = 0.001$

Figure A.7: LiRA and RMIA vulnerability ($\left(\mathrm{TPR} - \mathrm{FPR}\right)$) as a function of shots ($S$) when attacking a ViT-B Head fine-tuned without DP on different datasets. For better visibility, we split the datasets into two panels. We observe the power-law for both attacks, but the RMIA is more unstable than LiRA. The lines display the median over six seeds.

## D.4 Tabular results for Section 4.4

Table A10 displays the tabular results for Figure 6 in Section 4.4.

Table A10: Tabular results for Figure 6 on when attacking a ViT-B (Head) fine-tuned on PatchCamelyon. We display the median over six seeds at FPR $= 0.1$.

| $S$ | Max TPR | TPR of 0.999 Quantile | TPR of 0.99 Quantile | TPR of 0.95 Quantile |
|---|---|---|---|---|
| 16384 | 0.77 | 0.43 | 0.22 | 0.11 |
| 23170 | 0.73 | 0.37 | 0.19 | 0.10 |
| 32768 | 0.65 | 0.30 | 0.15 | 0.08 |
| 49152 | 0.54 | 0.23 | 0.13 | 0.07 |
| 65536 | 0.49 | 0.19 | 0.11 | 0.07 |

# E    Details on Section 4.5

In Section 4.5, we compare the results of our empirical models of vulnerability (Sections 4.3 and 4.4) to DP bounds. Below we explain how we make the connection between both.

First, we compute the upper bound on the TPR for a given $(\epsilon, \delta)$-DP privacy budget at a given FPR using Theorem 7 reformulated from Kairouz et al. (2015) below:

**Theorem 7** (Kairouz et al. (2015)). *A mechanism $\mathcal{M} : \mathcal{X} \to \mathcal{Y}$ is $(\epsilon, \delta)$-DP if and only if for all adjacent $\mathcal{D} \sim \mathcal{D}'$*

$$\text{TPR} \leq \min\{e^\epsilon \text{FPR} + \delta, 1 - e^{-\epsilon}(1 - \delta - \text{FPR})\} . \tag{A192}$$

For a given $(\epsilon, \delta)$ and FPR we then obtain a value for the TPR.

Next, we use the linear model from Section 4.3 to solve for the minimum $S$ predicted to be required given $C = 2$ classes in our example. The coefficients can be found in Table A8 for the average case and Section 4.4 for the worst-case. We solve the TPR from the linear model as

$$\log_{10}(\text{TPR} - \text{FPR}) = \beta_S \log_{10}(S) + \beta_C \log_{10}(C) + \beta_0 \tag{A193}$$

$$\Leftrightarrow \text{TPR} = S^{\beta_S} C^{\beta_C} 10^{\beta_0} + \text{FPR}. \tag{A194}$$

Now, we find the minimum $S$ that the TPR from Equation (A194) upper bounds the TPR of Equation (A192) as

$$S^{\beta_S} C^{\beta_C} 10^{\beta_0} + \text{FPR} = \min\{e^\epsilon \text{FPR} + \delta, 1 - e^{-\epsilon}(1 - \delta - \text{FPR})\} \tag{A195}$$

$$\Rightarrow S = \left( \frac{\min\{e^\epsilon \text{FPR} + \delta, 1 - e^{-\epsilon}(1 - \delta - \text{FPR})\} - \text{FPR}}{C^{\beta_C} 10^{\beta_0}} \right)^{1/\beta_S} \tag{A196}$$

