# OpenReview forum: "Impact of Dataset Properties on Membership Inference Vulnerability of Deep Transfer Learning"
_NeurIPS.cc/2025/Conference — NeurIPS 2025 poster_

### Official Review · Reviewer_AdF2 · 2025-06-21

**Clarity:** 3
**Significance:** 2
**Originality:** 2
**Rating:** 5
**Confidence:** 3

**Summary:**

This paper both theoretically and empirically establishes a quantitative relationship between black-box membership inference attack (MIA) success and the number of examples per class for fine-tuned image classification models. Leveraging this relationship, the authors propose a method to predict the MIA vulnerability of datasets under transfer learning.

**Questions:**

What about other data properties, e.g., variability within or between classes?

I suggest adding a short explanation of how RMIA improves performance even with a limited number of shadow models (Line 122). Do you know why RMIA is more unstable than LiRA?

**Ethical Concerns:**

["NO or VERY MINOR ethics concerns only"]

**Final Justification:**

The work is thorough and clearly written. The authors addressed the questions of all reviewers satisfactorily. I believe the work provides interesting insights but is not groundbreaking.

**Limitations:**

Yes, but I would like to see a more detailed discussion on the assumptions made on the underlying distributions and how these assumptions influence the conclusions.

**Paper Formatting Concerns:**

-

**Quality:**

3

**Strengths And Weaknesses:**

Strengths:
- Theoretical analyses: The authors establish a power law relationship between per-example vulnerability and examples per class for a simplified model of membership inference. They then generalize it for average vulnerability.
- Thorough empirical validation: The empirical findings confirm the theoretical relationship, demonstrating that average MIA vulnerability can be predicted using the number of classes and examples per class. The authors show that this prediction generalizes to different target model and fine-tuning methods, though it tends to underestimate vulnerability when training from scratch.
- Discussion of how this observation connects to DP guarantees, concluding that the number of examples per class required for protecting the most vulnerable examples is impractically large
- clearly written, well organized

Weaknesses:
- only dataset properties discussed are the number of classes and examples per class (see questions)
- would benefit from more discussion around the underlying data assumptions (see limitations)
- As stated, has limited practical relevance (but I still believe the work provides an interesting perspective that emphasizes the necessity of noise addition in transfer learning to mitigate privacy risks)

---

> ### Author Rebuttal · Authors · 2025-07-30
>
> Thanks for your review.
>
> > I would like to see a more detailed discussion on the assumptions made on the underlying distributions and how these assumptions influence the conclusions.
>
> In the simplified model of MIA, we assume that data is normally distributed around the class centres. As our empirical findings suggest, we believe that this assumption reflects most of typical well-behaved real-world data. We leave it to future work to study other distributions like heavy-tailed distributions. . We will change the limitations as follows:
>
> *”Furthermore, our simplified model assumes well-behaved underlying distributions, meaning that the data is normally distributed around the class centres. We leave the analysis of other data distributions (e.g. heavy-tailed distributions) to future work.”*
>
> > What about other data properties, e.g., variability within or between classes?
>
> Thanks for the suggestion, we believe that exploring other data properties is an interesting direction for future work as there is indeed a difference between classes as can be seen by the errorbars when classes are subsampled in Fig. 2.
>
> Additionally, we focus on settings in Section 4 where the test accuracy is $\ge80\%$ and not on datasets that have classes that are not separable, which could be a further dataset property worth exploring.
>
> We will add both as a suggestions to future work in our discussion by writing:
>
> *”Our experiments show that there is a difference between classes in terms of vulnerability and an interesting direction for future work is to understand the properties of classes that influence this vulnerability, e.g., variability within or between classes or their separability."*
>
> > I suggest adding a short explanation of how RMIA improves performance even with a limited number of shadow models (Line 122).
>
> We are rephrasing the claims of RMIA paper (their Fig. 3) in our background but do not have any data on this explicit claim as we always use many shadow models for our experiments.
>
> > Do you know why RMIA is more unstable than LiRA?
>
> First, we would like to say that RMIA is generally stronger than LiRA when only a few shadow models are used, but we use many shadow models where LiRA is strong as can be seen in Fig. 3 of Zarifzadeh et al., 2024. In that regime LiRA seems more stable than RMIA in our experiments.
>
> We use the implementation of the original authors of RMIA (in their ml_privacy_meter GitHub repository) and believe that we use it correctly as the majority of seeds is performing well (see Figure 3 in our paper). We are unsure why certain seeds degrade heavily with RMIA, but our setting is different from the RMIA paper:
> - We use fine-tuning while they train from scratch
> - We evaluate small datasets while they use large datasets.

---

> > ### Comment · Reviewer_AdF2 · 2025-08-06
> >
> > My questions have been addressed. I'll keep my positive score.

---

### Official Review · Reviewer_Hpo1 · 2025-06-23

**Clarity:** 2
**Significance:** 3
**Originality:** 3
**Rating:** 4
**Confidence:** 3

**Summary:**

This paper explores how class size affects data points' vulnerability to MIA in a transfer learning setting. Building on an experimentally observed power law relationship between class size and MIA vulnerability, the authors theoretically derive and experimentally verify this relationship, specifically for fine-tuned vision transformers and ResNet. The paper works to connect MIA vulnerability under these conditions to DP guarantees.

**Questions:**

Why do you think your prediction model ``underestimates the vulnerability of the from-scratch trained target models'' (lines 295-296)? What could be modified in your theory to better account for this, given that from-scratch training remains an important learning paradigm.

How is the regime explored here "between DP and typical MIA evaluations"? This claim is made in intro, but practically it just seems like this work derives theory for and experiments with non-DP fine-tuned classifiers. DP is only mentioned in lines 316-327 and some results shown in Figure 1. Given the strong claims in the intro about this work bridging MIA and DP, I would have expected to see much more substantial discussion of the interplay between these two methods.

More work is needed to distinguish the contribution of this paper from other works that have previously identified small class size as a risk factor for MIA (e.g. Kulynych et al, Chang and Shokri, etc.) Can the authors clarify where others' work ends and theirs begins?

**Ethical Concerns:**

["NO or VERY MINOR ethics concerns only"]

**Final Justification:**

The authors' rebuttal sufficiently addressed my concerns. Assuming the promised changes are made to the paper, I am now in favor of acceptance.

**Limitations:**

Yes

**Quality:**

3

**Strengths And Weaknesses:**

Strengths:
- Well developed theoretical explanations of how dataset properties relate to MIA vulnerability. This is an important contribution as the community actively works to understand what makes certain points more vulnerable to MIAs.
- Thorough experimental exploration of power law results in the transfer learning setting with ViTs and ResNet.

Weaknesses:
- Claims about DP in abstract are confusing/unnecessary given that the majority of the paper proceeds to ignore DP.
- Unclear how this result generalizes. The power law demonstrated in Figure 1 is derived in a very particular setting (e.g. fine-tuning on a ViT), and it's unclear if this result generalizes to other architectures and/or learning paradigms such as learning from scratch. Theory suggests it will, but it would be nice to see this backed up with experiments and/or argument.
- The run-on sentence at the end of the abstract was very hard to parse. There are several other run on-sentences in the paper (e.g. last sentence of 1st intro para). Please fix for readability.
- Figure 1 needs more explanation. What is C? Are all the different colors representing datasets? Why do you use the term "shots"? This may all be explained later in the paper, but the first figure should be interpretable on its own. Add content to the legend accordingly.

---

> ### Author Rebuttal · Authors · 2025-07-30
>
> Thanks for your review.
>
> We will address the points listed in your questions and weaknesses below:
> - From scratch training: We discuss why it is different based on Feldman and Zhang (2020)
> - Link to DP: Will be made clearer in updated manuscript
> - Novelty over other work: See the response below.
> - Claims about DP in abstract: We mention DP to motivate MIA, rather than make claims.
> - Generalization of power-law: We show it for ResNet and other fine-tuning methods, but believe it will not generalize to “from scratch training”
> - Readability of sentences in abstract, intro and caption of Fig. 1: Will be addressed
>
> See more details below:
>
> # Questions
> > Why do you think your prediction model ``underestimates the vulnerability of the from-scratch trained target models'' (lines 295-296)? What could be modified in your theory to better account for this, given that from-scratch training remains an important learning paradigm.
>
> We refer in our related work to Feldman and Zhang (2020) who show that pre-training (=from scratch training) with high utility requires memorizing a lot of training data while fine-tuning appears less vulnerable to memorization. It seems likely that pre-training will keep memorising outliers regardless of dataset size, so unlikely to have a power law. One argument for this is the capacity as fine-tuning has limited parameters while pre-training neural networks usually grow with the amount of pre-training data (as recommended by scaling laws).
>
> We will amend the related work sentence discussing Feldman and Zhang (2020) (in line 59f) as follows to be more clear about this:
>
> *”Feldman and Zhang (2020) showed that neural networks trained from scratch are required to memorize a significant fraction of their training data to obtain high utility, while the memorisation is greatly reduced for fine-tuning.”*
>
> About theory, according to Lemma 1 and the proof of Theorem 2, necessary conditions for the power-law is that $(\mu_\mathrm{in} - \mu_\mathrm{out}) / \sigma$ converges to zero at rate $O(1/S^\alpha)$ with $\alpha > 0$. In our simplified model, this holds with $\alpha=½$. However, $\mu_\mathrm{in} - \mu_\mathrm{out} \rightarrow 0$ would not always be the case for larger neural networks trained from scratch. Therefore, we do not expect similar power-laws in more general training algorithms. We will add this point after Theorem 2 and the from-scratch training part.
>
> > How is the regime explored here "between DP and typical MIA evaluations"? This claim is made in intro, but practically it just seems like this work derives theory for and experiments with non-DP fine-tuned classifiers. DP is only mentioned in lines 316-327 and some results shown in Figure 1. Given the strong claims in the intro about this work bridging MIA and DP, I would have expected to see much more substantial discussion of the interplay between these two methods.
>
> While DP gives a universal upper bound of privacy loss in a machine learning algorithm regardless of data, MIA, in contrast, provides a statistical lower bound that is often more practically realistic and depends on the training dataset. In our paper, we empirically and theoretically establish the power-law of MIA vulnerability, which bridges this gap by illustrating how the TPR for MIA of a non-DP algorithm decreases to match the TPR guaranteed by strong DP bounds. Obviously the guarantees are not of the same nature and we are not claiming that our approach can provide similar firm guarantees as DP.
>
> We reformulate that claim to be more precise as follows (line 19-20):
>
> *”In this paper, we seek to explore MIA vulnerability to extrapolate this gap”*
>
> > More work is needed to distinguish the contribution of this paper from other works that have previously identified small class size as a risk factor for MIA (e.g. Kulynych et al, Chang and Shokri, etc.) Can the authors clarify where others' work ends and theirs begins?
>
> Our novelty compared to prior works is the extrapolation of MIA vulnerability to DP guarantee, enabled by the power-law and the worst-case MIA analysis, both evaluated at a low FPR.
>
> Although previous studies including Kulynych et al. and Chang and Shokri demonstrated that classes with fewer examples are more vulnerable to MIA, they did not explore the **rate** of change in MIA vulnerability evaluated at a law FPR as the number of shots per class changes. In contrast, we explore this rate, namely the power-law, which helps estimate how much data you would need to obtain a desired MIA vulnerability. In addition, while prior research predominantly focuses on average-case vulnerability, we also show this power-law holds for worst-case vulnerability. The worst-case MIA is more important in that different samples exhibit different degrees of vulnerability.
>
> We will reformulate the last sentence of the related work part (line 67-68) as follows:
>
> *“Our work significantly expands on these works by a) explicitly identifying a quantitative relationship between dataset properties and MIA vulnerability, i.e., the power-law in Equation (1) and b) focusing on the worst-case vulnerability, both evaluated at a low FPR. This in turn allows us to extrapolate MIA vulnerability to DP guarantee.”*
>
> # Weaknesses
> > Claims about DP in abstract are confusing/unnecessary given that the majority of the paper proceeds to ignore DP.
>
> We use the sentence “MIAs complement formal guarantees from differential privacy (DP) under a more realistic adversary model.” to motivate the need for MIA evaluations.
>
> > Unclear how this result generalizes. The power law demonstrated in Figure 1 is derived in a very particular setting (e.g. fine-tuning on a ViT), and it's unclear if this result generalizes to other architectures and/or learning paradigms such as learning from scratch. Theory suggests it will, but it would be nice to see this backed up with experiments and/or argument.
>
> In Fig. 5 we show that our empirical finding generalizes to a ResNet (5a) as well as fine-tuning with adapters (5b, left). We believe that it does not generalize to “learning from scratch” (see your other question above).
>
> > The run-on sentence at the end of the abstract was very hard to parse. There are several other run on-sentences in the paper (e.g. last sentence of 1st intro para). Please fix for readability.
>
> Thanks for your suggestions. We will change the last sentence for the abstract as follows:
>
> *”We show that the vulnerability of non-DP models when measured as the attacker advantage at fixed false positive rate reduces according to a simple power law as the number of examples per class increases. A similar power-law applies even for the most vulnerable points, but the dataset size needed for adequate protection of the most vulnerable points is very large.”*
>
> We will fix the last sentence of 1st intro para:
>
> *”MIAs assume an often more realistic adversary model with access to just the data distribution and the unknown training data becoming latent variables that introduce stochasticity into the attack. However, the practical evaluation is statistical and cannot provide universal guarantees.”*
>
> We will fix other run on-sentences in the updated version of the paper.
>
> > Figure 1 needs more explanation. What is C? Are all the different colors representing datasets? Why do you use the term "shots"? This may all be explained later in the paper, but the first figure should be interpretable on its own. Add content to the legend accordingly.
>
> With C we denote the number of classes and the different colors are representing different fine-tuning datasets. We use the term shots as it is commonly used to denote examples per class, e.g., in the computer vision domain. We will update the x-axis of Figure to use examples per class.
>
> We will amend the caption as follows:
>
> *“We observe a power-law relation between MIA vulnerability and examples per class (denoted as $S$ or shots) when attacking a fine-tuned ViT-B Head using LiRA. Each colored line denotes a different fine-tuning dataset where $C$ specifies the number of classes. The solid line is median and the error bars the min/max bounds for the Clopper-Pearson CIs over six seeds.”

---

> > ### Comment · Reviewer_Hpo1 · 2025-08-01
> > **Response to rebuttal**
> >
> > Thanks to the authors for their thoughtful and thorough response to my concerns. Assuming the promised changes will be made to the paper, I am updating my score to a 4.

---

### Official Review · Reviewer_7Yjp · 2025-06-28

**Clarity:** 3
**Significance:** 3
**Originality:** 3
**Rating:** 5
**Confidence:** 4

**Summary:**

This paper provides a theoretical analysis of the impact of dataset size and numbers of classes on the performance of MIA attacks. At a high level, it derives relationships between the number of classes, datapoints per class, and the MIA vulnerability of a fine-tuned model when subject to a LIRA or RIMA attacks.

**Questions:**

See Weaknesses for areas which could be improved.

**Ethical Concerns:**

["NO or VERY MINOR ethics concerns only"]

**Final Justification:**

No major discussion with authors - keeping score unchanged

**Limitations:**

Yes

**Paper Formatting Concerns:**

No formatting concerns

**Quality:**

3

**Strengths And Weaknesses:**

Strengths

+ This is a large area with lots of empirical results - although the general intuitions exist that 1) larger the dataset (i.e. more examples per class) makes MIA attacks harder and 2) the more complex it is in terms of number of classes the MIA performance improves; this paper formalises this with a theoretical analysis to give a more principled grounding and better predictive ability for pre-deployment useage of models.

+ There is a good selection of different models and datasets (albeit, in the image only domain) showing results across a wide cross section.

Weakness

+ The theoretical analysis holds for fine-tuning, although diverges significantly for models trained from scratch in Fig. 5b, with from scratch models being significantly more vulnerable. Tentative explanations are given, though they are not very rigorous.

+ In particular, this can point to the pre-training dataset having a strong influence on the results - which deviate from the theory. If the theory only follows if the last layer, and last layer only, is fine tuned then this further limits the scope of the results.

+ The analysis focuses heavily on two attacks. The proposed theory may not hold in a general sense to future (potentially stronger) attacks that rely on a different attack approach or principle. It is however, worth noting that the paper highlights that a LiRA based approach is the optimal strategy to carry out MIA via Neyman–Pearson lemma, so a strong attack class is considered here.

Overall

On the whole, even if just relevant to fine-tuning, it is is a very common deployment scenario and, for now, the analysed attacks are some of the strongest MIA attacks to date. Thus, adding theory to practical MIA effectiveness is a useful contribution.

---

> ### Author Rebuttal · Authors · 2025-07-30
>
> Thanks for your review.
>
> We thank you for the interesting points you raised for future work and we will add them to the discussion section.

---

### Official Review · Reviewer_c8jK · 2025-07-03

**Clarity:** 3
**Significance:** 2
**Originality:** 3
**Rating:** 5
**Confidence:** 2

**Summary:**

This paper explores membership inference attacks (MIAs) with respect to dataset properties under the non-differential privacy (non-DP) setting for fine-tuned models. Specifically, it establishes a power-law relationship indicating reduced MIA vulnerability as the number of examples per class increases, based on a simplified regression model. The employed MIA methods include LiRA and RMIA. This is a technically solid paper, supported by multiple experiments that demonstrate the effectiveness of the findings.

**Questions:**

1. Regarding the scope of the research, what is the rationale for studying the power law of MIA on fine-tuned models? Would this relationship be more difficult to observe if the model were trained from scratch?
2. For LiRA, does the offline version still exhibit the observed power-law behavior?
3. In Figure 3, could the authors explain why the power-law trend for RMIA appears less stable compared to that of LiRA?

**Ethical Concerns:**

["NO or VERY MINOR ethics concerns only"]

**Final Justification:**

The authors’ response has addressed my concerns; accordingly, I have raised my score.

**Limitations:**

yes

**Quality:**

3

**Strengths And Weaknesses:**

**Strengths**
1.  The methodology is technically sound, and the presentation is clear and well-structured.

2. The employed MIA methods, LiRA and RMIA, are among the current state-of-the-art.

3.  The paper studies both average and individual-level vulnerabilities, providing a more complete picture.



**Weaknesses**

1. The motivation for focusing on fine-tuned models is unclear. Please refer to Question 1.

2. The theoretical power law is derived from a simplified model and does not directly apply to deep learning models, where the relationship between training data and MIA vulnerability has already been extensively studied. This limits the overall contribution.

---

> ### Author Rebuttal · Authors · 2025-07-30
>
> Thanks for your review.
>
> > Regarding the scope of the research, what is the rationale for studying the power law of MIA on fine-tuned models? Would this relationship be more difficult to observe if the model were trained from scratch?
>
> We focus on fine-tuned models for the following reasons:
> - By far the most practical way to get practically usable models
> - Lower bound on vulnerability
> - Allows connection to theory (simplified model is last layer fine-tuning)
> - Fine-tuning is computationally much cheaper, enabling large-scale experiments with a reasonable budget.
>
> > For LiRA, does the offline version still exhibit the observed power-law behavior?
>
> Thank you for raising an interesting point. We conducted additional theoretical analysis on offline LiRA, and found that the power-law holds in our simplified model. This result will be added to Appendix and mentioned in the first paragraph of Sec 3. In short, if we assume that offline LiRA scores $t_x$ follow Gaussian distributions (unlike the general location-scale family for online LiRA), we obtain a similar closed-form vulnerability like Lemma 1. This in turn implies the same power-law for offline LiRA.
>
> Empirically for the deep learning models, the LiRA offline attack is less robust and can be much weaker than the online attack. See Fig. 6 of Carlini et al., 2022 and Fig. 3 of Zarifzadeh et al., 2024.
>
> > In Figure 3, could the authors explain why the power-law trend for RMIA appears less stable compared to that of LiRA?
>
> First, we would like to say that RMIA is generally stronger than LiRA when only a few shadow models are used, but we use many shadow models where LiRA is strong as can be seen in Fig. 3 of Zarifzadeh et al., 2024. In that regime LiRA seems more stable than RMIA in our experiments.
>
> We use the implementation of the original authors of RMIA (in their ml_privacy_meter GitHub repository) and believe that we use it correctly as the majority of seeds is performing well (see Figure 3 in our paper). We are unsure why certain seeds degrade heavily with RMIA, but our setting is different from the RMIA paper:
> - Fine-tuning while they train from scratch
> - We evaluate small datasets

---

> > ### Comment · Reviewer_c8jK · 2025-08-03
> > **Thank You for Your Response**
> >
> > Thank you for the clarification. Your response addressed my concerns, and I have raised my score accordingly.

---

### Comment · Area_Chair_pSne · 2025-08-01

Dear Reviewers,

The author-reviewer discussion phase has started. If you want to discuss with the authors about more concerns and questions, please post your thoughts by adding official comments as soon as possible.

Thanks for your efforts and contributions to NeurIPS 2025.

Best regards,

Your Area Chair

---

### Decision · Program_Chairs · 2025-09-17

**Decision:**

Accept (poster)

**Comment:**

This paper did an interesting exploration of MIA in the transfer learning and fine-tuning scenarios. Both scenarios are quite important in the current practice of machine learning algorithms and methods. Thus, the studied topic is important and timely. Some presentation issues were pointed out in the initial reviews, and then the authors addressed them well during the rebuttal. In the updated version of this paper, the authors should focus on the main message demonstration and remove some redundant parts, as reviewers pointed out.

After the rebuttal, all reviewers find the merits of this paper and supported the acceptance.